# TimeRecipe: A Time-Series Forecasting Recipe via Benchmarking Module Level Effectiveness

**Zhiyuan Zhao**
Georgia Institute of Technology
leozhao1997@gatech.edu

**Juntong Ni**
Emory University
juntong.ni@emory.edu

**Shangqing Xu**
Georgia Institute of Technology
sxu452@gatech.edu

**Haoxin Liu**
Georgia Institute of Technology
hliu763@gatech.edu

**Wei Jin**
Emory University
wei.jin@emory.edu

**B. Aditya Prakash**
Georgia Institute of Technology
badityap@cc.gatech.edu

## ABSTRACT

Time-series forecasting is an essential task with wide real-world applications across domains. While recent advances in deep learning have enabled time-series forecasting models with accurate predictions, there remains considerable debate over which architectures and design components, such as series decomposition or normalization, are most effective under varying conditions. Existing benchmarks primarily evaluate models at a high level, offering limited insight into why certain designs work better. To mitigate this gap, we propose TimeRecipe, a unified benchmarking framework that systematically evaluates time-series forecasting methods at the module level. TimeRecipe conducts over 10,000 experiments to assess the effectiveness of individual components across a diverse range of datasets, forecasting horizons, and task settings. Our results reveal that exhaustive exploration of the design space can yield models that outperform existing state-of-the-art methods and uncover meaningful intuitions linking specific design choices to forecasting scenarios. Furthermore, we release a practical toolkit within TimeRecipe that recommends suitable model architectures based on these empirical insights.

## 1 INTRODUCTION

Time-series forecasting plays a critical role in a wide range of real-world domains, including economics, urban computing, and epidemiology (Zhu & Shasha, 2002; Zheng et al., 2014; Deb et al., 2017; Mathis et al., 2024). These applications focus on predicting future trends or events based on patterns observed in historical time-series data. Recently, the emergence of deep learning has significantly advanced the field, leading to the development of numerous forecasting models (Lai et al., 2018; Torres et al., 2021; Salinas et al., 2020; Nie et al., 2023; Zhou et al., 2021; Liu & Kamarthi, 2024; Liu et al., 2024b; Ni et al., 2025b; Zhao et al., 2025c). These models have demonstrated strong predictive accuracy and generalization capability across diverse datasets and domains, particularly within the supervised time-series forecasting paradigm.

Despite recent successes, particularly at the model level (i.e., end-to-end forecasting architectures and pipelines), ongoing debates persist regarding the most effective deep learning strategies at the module level, referring to the internal design components within forecasting models. For example, while Transformer-based architectures are known for their ability to capture long-range temporal dependencies, they tend to struggle to generalize well on highly irregular time-series patterns, such as ETT time series. This has led to the reflection in reconsidering the effectiveness of MLP-based designs (Zeng et al., 2023; Xu et al., 2023; Wang et al., 2024a; Ni et al., 2025a). Beyond architectural debates, further divergence arises in the design of specific modules and components. For instance, (Wang et al., 2024a) highlights the importance of separately modeling trend and seasonal components

via series decomposition, which is overlooked in other contemporary works (Shi et al., 2024). Additionally, (Zhao et al., 2025b) raises concerns for the effectiveness of instance normalization in short-term forecasting tasks when capturing rapid trend changes, despite it being a commonly effective practice primarily associated with long-term forecasting scenarios.

However, despite ongoing debates surrounding divergent design choices at module level, existing benchmarks only emphasize evaluation at the model level (Qiu et al., 2024; Aksu et al., 2024; Li et al., 2024; Du et al., 2024), overlooking the importance of benchmarking the effectiveness of specific module-level components. These studies typically conclude with case-specific best-performing models, which can be less practical when applying benchmarked results to real-world forecasting scenarios that fall outside the benchmarked settings (Brigato et al., 2025). Moreover, while these benchmarks are comprehensive and empirically informative, they offer limited intuitive insights into why certain models outperform others in specific forecasting scenarios, leaving important questions about model behavior and design choices underexplored.

To bridge this gap and gain a deeper understanding of the underlying factors driving model performance, we propose TIMERECIPE. While existing benchmarks focus primarily on model-level evaluation and offer limited interpretability, TIMERECIPE takes a step further by benchmarking time-series forecasting methods at a finer-grained, module level. Specifically, TIMERECIPE aims to assess the effectiveness of individual components commonly used in state-of-the-art forecasting models. This enables us to answer a key research question: **Which modules and model designs are most effective under specific time-series forecasting scenarios?** Guided by this question, we summarize the main contributions of this work as follows:

- **Novel Benchmarking Scope:** Unlike existing time-series forecasting benchmarks that focus primarily on holistic evaluation of entire state-of-the-art (SOTA) models, TIMERECIPE focuses on assessing the effectiveness of individual modules commonly used in model construction. To the best of our knowledge, TIMERECIPE is the first benchmark to systematically explore the design space of forecasting models for supervised time-series forecasting tasks at the modular level.

- **Comprehensive Evaluations:** TIMERECIPE evaluates hundreds of module combinations across diverse forecasting scenarios, spanning univariate and multivariate settings, as well as short- and long-term horizons. The benchmark encompasses dozens of datasets and involves over 10,000 distinct experiments, offering a robust and exhaustive evaluation framework.

- **Insightful Findings:** The TIMERECIPE benchmark reveals that exhaustive exploration of the modular design space can yield forecasting models that outperform existing SOTA approaches. Moreover, it uncovers meaningful correlations between module effectiveness and specific characteristics of time-series data and forecasting tasks, offering insights beyond raw accuracy.

- **Actionable Toolkit:** Building on the above insights, we develop a training-free toolkit that makes model architecture selections within TIMERECIPE. We demonstrate its effectiveness by comparing the selected architectures against those discovered via exhaustive search.

**Problem Formulation.** We define the time-series forecasting problem following popular existing formulations (Zhou et al., 2021; Liu et al., 2023a; 2024a): Given historical observations $\mathrm{X}_t = \{\boldsymbol{x}_{t-L}, \ldots, \boldsymbol{x}_t\} \in \mathbb{R}^{L \times d_{\mathrm{X}}}$ consisting of $L$ past time steps and $d_{\mathrm{X}}$ variables at time step $t$, the goal is to predict the future $H$ steps $\mathrm{Y}_t = \{\boldsymbol{x}_{t+1}, \ldots, \boldsymbol{x}_{t+H}\} \in \mathbb{R}^{H \times d_{\mathrm{X}}}$. For convenience, we denote X as the collection of all $\mathrm{X}_t$ over the full time series of length $T$, and similarly, Y as the collection of all corresponding $\mathrm{Y}_t$. The time-series forecasting task aims to learn a model parameterized by $\theta$ through empirical risk minimization (ERM) to obtain $f_\theta : \mathrm{X} \to \mathrm{Y}$ for all time steps $t \in T$.

**Additional Related Work.** Due to page limitations, we provide an extended discussion of related works in Appendix A, including reviews of model designs for supervised learning and foundation models in time-series forecasting, as well as existing time-series forecasting benchmarks.

## 2 TIMERECIPE FRAMEWORK

Since the introduction of Transformer-based architectures to time-series forecasting, particularly following the release of Informer (Zhou et al., 2021), most modern approaches have converged toward a common design paradigm, referred to here as the ***Canonical Architecture***, as illustrated in Figure 1.

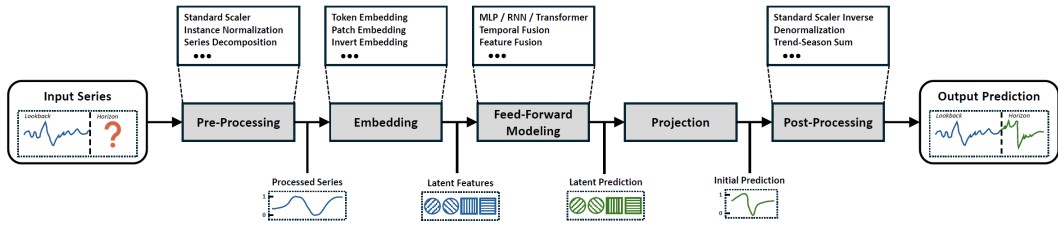

Figure 1: The proposed canonical architecture in TIMERECIPE for constructing general time-series forecasting models. The canonical architecture comprises five key components: pre-processing, embedding, feed-forward modeling, projection, and post-processing.

This canonical architecture consists of five major components: pre-processing, embedding, feed-forward modeling, projection, and post-processing (which is paired with pre-processing operations). The canonical architecture serves as the foundation of TIMERECIPE and captures the typical structure adopted by many state-of-the-art (SOTA) forecasting methods.

Following this structure, TIMERECIPE is designed to offer a systematic evaluation with in-depth intuitions of alternative module choices across key components of the canonical architecture. Specifically, TIMERECIPE benchmarks common module designs for pre-processing, embedding, and feed-forward modeling, with the aim of understanding their relative effectiveness across diverse forecasting scenarios and in conjunction with other design choices. For example, one example question to answer by our benchmark could be: *how do different embedding strategies perform under varying task settings, and how do they interact with other designs from other components of the canonical architecture?*

We omit the benchmarking of the projection component, as it typically consists of a simple linear layer and has attracted less interest regarding its design effectiveness. Similarly, the post-processing component is inherently paired with pre-processing operations; thus, evaluating the effectiveness of pre-processing modules simultaneously assesses the corresponding post-processing steps. We detail all benchmarked module designs included by TIMERECIPE in the subsequent sections.

## 2.1 PRE-PROCESSING

In TIMERECIPE, we consider two types of popular time series pre-processing approaches that are widely used in time-series forecasting tasks: Instance Normalization (Kim et al., 2021; Liu et al., 2022c; Fan et al., 2023; Liu et al., 2023b; Han et al., 2024b) and Series Decomposition (Wu et al., 2021; Wang et al., 2024a; 2025).

**Instance Normalization.** Instance normalization normalizes each input sample independently to a standard 0–1 distribution, regardless of its original distribution. This process enables the model to learn translations from historical lookback to horizon predictions more stably, as all samples are mapped into a consistent distributional space. Instance normalization is paired with a denormalization step applied after the model output, projecting the predictions from the normalized space back to the original feature space. The full process is as follows:

$$\text{Norm}: X_t^{\text{Norm}} = \frac{X_t - \mu(X_t)}{\sqrt{\sigma^2(X_t) + \epsilon}}, \quad \text{Denorm}: \hat{Y}_t = \hat{Y}_t^{\text{Norm}}\sqrt{\sigma^2(X_t) + \epsilon} + \mu(X_t) \quad (1)$$

**Series Decomposition.** Series decomposition in time-series forecasting aims to disentangle the seasonal and trend components within input instances using simple moving average operations. The trend component, obtained via moving average, captures the overall directional changes of the time series and primarily consists of low-frequency variations. The seasonal component is defined as the residual between the original time series and the extracted trend, typically reflecting higher-frequency, periodic patterns. We formulate this process as follows.

$$X_t^{\text{Trend}} = \text{AvgPool}(\text{Padding}(X_t)), \quad X_t^{\text{Season}} = X_t - X_t^{\text{Trend}} \quad (2)$$

For post-processing, the predictions of the trend and seasonal components from the feed-forward model are typically summed to reconstruct the final predictions (i.e., $\hat{Y} = \hat{Y}^{\text{Trend}} + \hat{Y}^{\text{Season}}$).

## 2.2 EMBEDDING

In TIMERECIPE, we consider four popular embedding approaches widely adopted in time-series forecasting tasks: Token (Zhou et al., 2021), Patch (Nie et al., 2023), Invert (Liu et al., 2023a), and Frequency (Xu et al., 2023) Embedding. Additionally, we include a no-embedding variant as a controlled baseline for completeness, although it typically yields inferior performance, e.g., applying feed-forward modeling directly on the raw feature space. In short, the mathematical formulations of these embeddings are summarized in Equation 3.

$$
\begin{aligned}
X_t \in [B, L, D], \quad X_t^{\text{Token}} &= \text{Conv}(\text{Padding}(X_t^T))^T & &\in [B, L, H] \\
X_t^{\text{Patch}} &= \text{Conv}_{k=\text{PatchLen},s=\text{Stride}}(\text{Padding}(X_t^T)) & &\in [B \times D, H, \lfloor \frac{L - \text{PatchLen}}{\text{Stride}} \rfloor + 2] \\
X_t^{\text{Invert}} &= \text{Linear}(X_t^T) & &\in [B, H, D] \\
X_t^{\text{Freq}} &= \text{rFFT}(X_t) & &\in [B, \lfloor \frac{L}{2} \rfloor + 1, D]
\end{aligned}
\tag{3}
$$

Here, the superscript $T$ denotes the transpose operation between the second and third dimensions. D is the number of raw features (i.e., $d_X$), and H represents the hidden dimension. We detail the specifics of each embedding strategy in the following subsections.

**Token Embedding.** Token embedding applies convolutional operations along the temporal axis of the input time series to project each timestamp into a higher-dimensional embedding space, analogous to word embeddings in natural language processing (Devlin et al., 2019). Specifically, each timestamp, comprising all features and neighboring time steps (depending on the kernel size), is treated as an individual token. This approach preserves temporal order while enabling the model to learn contextualized representations from sequences of embedded tokens.

**Patch Embedding.** Patch embedding segments the input time series into patches every several timestamps along the temporal dimension using convolution, similar to the patching mechanism in vision tasks (Dosovitskiy et al., 2020). In the time-series context, patch embedding operates on each feature (channel) independently. The feed-forward model then treats the collection of patches as a batch, resulting in a channel-independent processing scheme, which is a design introduced alongside the patch embedding paradigm for time-series forecasting.

**Invert Embedding.** Invert embedding performs a linear projection across the temporal dimension for each feature independently, treating the entire lookback window of a single variable as a token. This representation is typically followed by a feed-forward model that focuses on learning inter-feature (token-wise) dependencies. It enables the model to capture relationships across different variables while maintaining the temporal integrity of each.

**Frequency Embedding.** Frequency embedding applies the RealFFT (rFFT) (Brigham & Morrow, 2009) to transform time-series sequences from the time domain into the frequency domain, enabling models to operate on spectral components rather than raw temporal signals. The resulting representation typically requires subsequent feed-forward models capable of handling complex-valued inputs, currently limited to MLPs in PyTorch implementations. To reconstruct the original temporal features, an inverse rFFT (irFFT) is employed in place of standard linear projections. Notably, frequency embedding is a non-parametric operation, distinguishing it from other learnable embedding methods.

## 2.3 FEED-FORWARD MODELING

In TIMERECIPE, we consider various feed-forward modeling approaches from both a model architecture perspective (FF-type) and a model fusion perspective. Specifically, for model architectures, we include MLPs, Transformers, and RNNs. For model fusion strategies, we distinguish between temporal fusion, which aims to capture temporal dependencies, and feature fusion, which focuses on modeling correlations among features. We provide detailed descriptions of these feed-forward modeling approaches below.

**Model Architectures.** While Transformer-based models have achieved notable success in time-series forecasting, largely due to their ability to capture long-range dependencies across time steps or features, recent studies have also highlighted the effectiveness of MLP-based models for this task (Xu et al., 2023; Zeng et al., 2023; Ni et al., 2025a). Motivated by these findings, TIMERECIPE aims

to provide a deeper understanding of the relative effectiveness of different model architectures. For completeness, we also include RNNs as a baseline, representing a classical class of autoregressive models. RNN-based methods preceded the widespread adoption of Transformers, showing competitive performance in earlier work (Salinas et al., 2020; Lai et al., 2018) and continue to demonstrate potential in more recent studies (Kong et al., 2025).

**Modeling Fusion.** Existing time-series forecasting approaches predict future values by either modeling temporal dependencies or capturing feature correlations. When considering fusion types across different model architectures, we highlight that the processing differs even under the same fusion type, depending on the underlying architecture, as shown below:

$$
\begin{aligned}
\text{Temporal:} \quad &\text{MLP}(X_t^{\text{Emb}} \in [B, D, L]), &\quad \text{Feature:} \quad &\text{MLP}(X_t^{\text{Emb}} \in [B, L, D]) \\
&\text{RNN}(X_t^{\text{Emb}} \in [B, L, D]), &\quad &\text{RNN}(X_t^{\text{Emb}} \in [B, D, L]) \\
&\text{Transformer}(X_t^{\text{Emb}} \in [B, L, D]), &\quad &\text{Transformer}(X_t^{\text{Emb}} \in [B, D, L])
\end{aligned}
\tag{4}
$$

Here, we slightly abuse the notations: B is the batchsize, L is the temporal dimension, and D is the feature dimension, which are the 1st, 2nd, and 3rd dimension after the embedding, respectively, as to generalize the different shapes produced by various embedding strategies.

## 2.4 TIMERECIPE BENCHMARK

With the component designs introduced above, we now present TIMERECIPE, **a unified time-series forecasting framework** that (i) implements the discussed module designs from each component of the canonical architecture, and (ii) automatically adjusts inter-module connections. TIMERECIPE controls the use of different designs through hyperparameters, including whether to apply instance normalization and series decomposition, the choice of embedding type, model architecture, and fusion type. Accordingly, TIMERECIPE constructs the full model pipeline by automatically adjusting hidden dimensions, initializing appropriate module connections, and applying proper tensor operations during the forward pass. Beyond providing a comprehensive benchmarking and intuition-driven study as in this work, TIMERECIPE also offers practical benefits: it enables time-series researchers to conveniently build and evaluate models on their own data, and provides flexibility for exploring novel designs across different components within the canonical architecture framework.

**Benchmark Scope and Coverage.** Through exhaustive design space exploration enabled by TIMERECIPE, the framework covers over 100 types of model architectures via module-level combinations. By adjusting the component configurations, TIMERECIPE is able to encompass many popular time-series forecasting models, as partially illustrated in Table 1. Consequently, comprehensive benchmarking of TIMERECIPE not only evaluates existing forecasting models, but also uncovers the potential of alternative combinations of model designs.

| TIMERECIPE Component | | | | | Published Work |
|---|---|---|---|---|---|
| IN | SD | Fusion | Embedding | FF-Type | Model |
| ✓ | ✗ | Feature | Invert | Trans. | iTransformer (Liu et al., 2023a) |
| ✓ | ✓ | Temporal | Freq. | MLP | FITS (Xu et al., 2023) |
| ✓ | ✗ | Temporal | Patch | Trans. | PatchTST (Nie et al., 2023), PAttn (Tan et al., 2024) |
| ✗ | ✓ | Temporal | None | MLP | DLinear (Zeng et al., 2023) |
| ✗ | ✓ | Feature | Token | Trans. | Autoformer (Wu et al., 2021) |
| ✗ | ✗ | Temporal | Token | Trans. | Informer (Zhou et al., 2021) |

Table 1: Examples of existing methods that are covered by TIMERECIPE benchmark.

**Remark 1.** *Channel-independence (Nie et al., 2023; Han et al., 2024a) is an important property in time-series forecasting models, referring to the ability to forecast multiple time series independently while sharing model parameters. TIMERECIPE implicitly incorporates channel-independence in its evaluations by combining specific embedding types and model architectures with temporal modeling fusion. For example, using invert embedding with an MLP model architecture alongside temporal fusion serves as an instance of channel-independent modeling.*

**Remark 2.** *New modules can be readily incorporated into TIMERECIPE. We focus on representative modules that are widely adopted and influential design choices, while omitting others to avoid the combinatorial explosion of possible configurations. The uncovered designs generally fall into two categories. First, highly specialized designs, such as tangled temporal and feature fusion with*

*delicate design used specifically in Crossformer (Zhang & Yan, 2023). Second, designs that are of less concern on module effectiveness, such as data augmentation based on down-sampling (Wang et al., 2024a; 2025) For instance, TimeMixer (Wang et al., 2024a) can be viewed as IN+SD+Temporal-Fusion+Non-Embedding+MLP (thus falling within* TIMERECIPE*), with an additional augmentation applied at the pre-processing stage. Augmentation approaches exhibit general benefits for predictive models with fewer concerns; therefore, they are currently omitted from our benchmark studies.*

**Remark 3.** *While many current SOTA forecasting methods focus on foundation and prompt-based methods,* TIMERECIPE *targets supervised learning and already includes the leading supervised approaches with existing module-level combinations. Foundation and prompt-based methods are typically built as end-to-end pre-trained frameworks, where altering a single module can disrupt the effectiveness of other pre-trained components. Furthermore, time-series foundation models remain at an early and fragmented stage, with directions ranging from scaling Transformers (e.g., Time-MOE (Shi et al., 2024), TimesFM (Das et al.)) to adapting LLMs (e.g., Chronos (Ansari et al., 2024)) or repurposing tabular foundation models (e.g., TabPFN (Hollmann et al., 2022)). Rather than benchmarking these diverse paradigms, which have already been explored in works such as GiftEval (Aksu et al., 2024), our focus is on delivering actionable insights into supervised models, which we believe can also inform the principled development of future time-series foundation models.*

## 3    EVALUATION PROTOCOL

### 3.1    EVALUATION TASKS

We distinguish time-series forecasting tasks from two perspectives. First, from a feature perspective, we categorize tasks into *multivariate forecasting*, where $d_X > 1$ and the goal is to predict the future values of all feature dimensions simultaneously, and *univariate forecasting*, where $d_X = 1$ and the task focuses on forecasting future values of a single dimension based solely on its own historical observations without exogenous features. Second, from a horizon length perspective, we classify tasks into *short-term* and *long-term* forecasting. We define short-term forecasting as cases where the lookback window is longer than the forecasting horizon (i.e., $L > H$), and long-term forecasting as cases where the lookback window is shorter than the forecasting horizon (i.e., $L \leq H$).

Following this distinction, TIMERECIPE aims to comprehensively benchmark the effectiveness of the modules described in Section 2 across all time-series forecasting tasks, including short-term univariate, short-term multivariate, long-term univariate, and long-term multivariate forecasting tasks. These tasks cover a wide range of time-series forecasting scenarios, where the driven intuition on module effectivness can offer practical insights for real-world forecasting applications.

### 3.2    EVALUATION DATASET

**LTSF.** The Long-Term Time-Series Forecasting (LTSF) datasets (Zhou et al., 2021) are widely adopted benchmarks for evaluating both short-term and long-term forecasting tasks. They test a model's ability to generalize across diverse domains, including Electricity Transformer Temperature (**ETT**; ETTh1/2, ETTm1/2) (Zhou et al., 2021), Influenza-like Illness statistics (**ILI**), Electricity Consumption Load (**ECL**) (Asuncion et al., 2007), meteorological data from the National Renewable Energy Laboratory (**Weather**), and foreign exchange rates across various countries (**Exchange**).

**PEMS.** The Performance Measurement System (PEMS, extended Traffic dataset in LTSF) datasets (Li et al., 2017) are standard benchmarks for time-series forecasting, commonly used in traffic prediction research. These datasets contain road occupancy or flow measurements collected by loop detectors on highways across different districts in California. We include **PEMS03**, **PEMS04**, **PEMS07**, and **PEMS08**, each varying in geographic scope, number of sensors, and data volume.

**M4.** The M4 dataset (Makridakis et al., 2018) is a large-scale benchmark for evaluating forecasting models across diverse real-world time series. It includes 100,000 series from domains such as macroeconomics, microeconomics, finance, industry, and demography. Each time series varies in length and frequency, spanning yearly, quarterly, monthly, weekly, daily, and hourly settings.

We include additional details and the rationale behind the selection of these datasets in Appendix B.1.

### 3.3 EVALUATION METRIC

We adopt common evaluation metrics on each dataset. Specifically, we use mean squared error (MSE) and mean absolute error (MAE) for evaluations on LTSF and PEMS datasets, and symmetric mean absolute percentage error (SMAPE), mean absolute scaled error (MASE), and overall weighted average (OWA) for M4 dataset. The formula of these metrics is detailed in Appendix B.2.

Since error scales vary significantly across datasets, we use averaged rank scores to enable fair comparisons of module effectiveness. For example, if a module combination ranks 1st in MSE and 2nd in MAE, its average rank score is calculated as 1.5. This unified ranking metric facilitates consistent evaluation across datasets and forms the basis of our analysis in Section 4.

**Implementation.** The implementation of TIMERECIPE follows the widely-used Time-Series Library [1] (Wang et al., 2024b). For usability, we introduce hyperparameters that control the inclusion of each module, allowing users to specify their usage of each module while TIMERECIPE automatically adjusts the internal architecture and dimensions accordingly. Detailed hyperparameter settings and tuning strategies are provided in Appendix B.4.

**Reproducibility.** For reproducibility, we provide all data, code, and scripts, publicly available at `https://github.com/AdityaLab/TimeRecipe`. All experiments are conducted on 32GB V100 GPUs. For statistical robustness, all results are based on the average over four random seeds.

## 4 RESULTS AND DISCUSSION

In this section, we present the benchmark results along with key insights and discussions derived from them. Due to space constraints, we include partial results in Appendix C. and full results (both raw and processed) are available at: `https://github.com/AdityaLab/TimeRecipeResults`.

### 4.1 INSIGHTS DERIVED FROM BENCHMARK RESULTS

#### 4.1.1 TIMERECIPE YIELDS MODEL ARCHITECTURES OUTPERFORMING SOTA

A key observation from our evaluations is that exhaustive exploration of the design space across diverse datasets can identify model configurations that outperform existing SOTA approaches that are themselves included within the search space. For example, in the short-term multivariate forecasting on the PEMS03 dataset with a horizon of 12, the top-ranked configuration achieves an MSE of 0.714, outperforming iTransformer, one of the best existing forecasting models also covered by TIMERECIPE, which attains an MSE of 0.739 and ranks only 7th among all evaluated design combinations.

More importantly, this phenomenon is not a rare case: it holds in 92 out of 102 evaluated scenarios, detailed in Appendix C.3, suggesting that in over 90% of cases, TIMERECIPE can identify a model architecture that surpasses the SOTA. On average, the best existing approaches lag behind the top-performing configurations identified by TIMERECIPE by 13.66 ranking positions. By exhaustively exploring the design space, TIMERECIPE further achieves an average forecasting error reduction of **5.4%** compared to the best existing approaches (std.$= 2.88\%$, t-test $p$-value$= 0.0069$).

This observation underscores the importance of exhaustive design space search, as by TIMERECIPE, for achieving superior forecasting performance across different scenarios. Moreover, TIMERECIPE offers practical utility in the convenience of the construction of end-to-end forecasting pipelines. For instance, when introducing a novel embedding technique for time-series forecasting, researchers can leverage the canonical architecture of TIMERECIPE to identify the optimal configuration that best complements the proposed component. This facilitates fair and effective evaluations across a wide range of scenarios and supports the development of new state-of-the-art solutions.

#### 4.1.2 TIMERECIPE UNCOVERS MODULES EFFECTIVENESS LINKING TO DATA PROPERTIES

The major motivation of TIMERECIPE is to systematically investigate the effectiveness of various architectural modules across diverse time-series forecasting scenarios, especially in relation to the intrinsic characteristics of different datasets. This motivation stems from the variation observed in

---

[1]`https://github.com/thuml/Time-Series-Library`

optimal architectural configurations across datasets. For instance, the top-performing models on multivariate forecasting tasks using the ETT datasets often combine patch embeddings with MLP or RNN-based feed-forward networks. In contrast, on the Electricity dataset, models that with invert embeddings with Transformer architectures yield the best results. Another noteworthy finding is that while instance normalization generally improves performance across many LTSF benchmarks, its performance on the PEMS datasets is degraded. These trends highlight the potential for uncovering meaningful interactions between model architecture choices and dataset properties.

To address this gap and answer our central research question: **Which modules and model designs are most effective under specific time-series forecasting scenarios?** As a first step, we establish a taxonomy of time-series data characteristics. This taxonomy includes key properties such as seasonality, trend, stationarity, transition, shifting, and correlation (Qiu et al., 2024), as detailed in Appendix B.3. In addition to these intrinsic data properties, we further categorize forecasting tasks based on input structure, namely, whether the data is univariate or multivariate, as well as the number of input features (N-Feature) and the horizon-to-lookback ratio (HL-Ratio), which may also influence the effectiveness of specific module configurations.

Building on these data characteristics, we conduct a correlation analysis to explore the relationship between dataset properties and the effectiveness of various module designs. Through the analysis, we observe patterns among the statistically significant module-level effectiveness, as shown in Table 2. Specifically, we employ statistical hypothesis testing via t-tests to assess whether particular module configurations yield significantly improved performance under specific data conditions. This analysis directly supports our primary research goal of identifying the most appropriate design choices for diverse forecasting scenarios. A complete list of all statistically significant correlations (i.e., $p$-value $\leq 0.05$) is provided in Table 4, Appendix C.1.

| Setting | Choice | | Condition | | |
|---|---|---|---|---|---|
| Multivariate | Instance Norm. | *when* | n-feature is low
shifting is high | hl-ratio is high
trend is high | correlation is high
seasonality is low |
| | Series Decomp. | *when* | shifting is low | | |
| | Temporal Fusion | *when* | n-feature is low
transition is high
seasonality is low | hl-ratio is high
shifting is high | correlation is low
trend is high |
| | Invert Embed. | *when* | n-feature is high
shifting is low | correlation is high | transition is low |
| | Token Embed. | *when* | hl-ratio is low
stationarity is low | trend is low | seasonality is high |
| | Patch Embed. | *when* | trend is high | | |
| | RNN Arch. | *when* | n-feature is high
trend is low | hl-ratio is low
stationarity is low | shifting is low |
| | Transformer Arch. | *when* | correlation is high | transition is low | seasonality is high |
| Univariate | Series Decomp. | *when* | transition is high
stationarity is high | shifting is high | trend is high |
| | Temporal Fusion | *when* | hl-ratio is high
seasonality is low | shifting is high | trend is high |
| | Non-Embed. | *when* | trend is high | | |
| | Transformer Arch. | *when* | seasonality is high | | |

Table 2: Module choices under specific data property conditions.

These findings provide a more granular view of module-level effectiveness, some of which align with well-established knowledge. For example, instance normalization, which is specifically designed to handle distribution shifts, is most beneficial when shifts are large or seasonality is low. Similarly, RNN architectures are more flexible when the HL-ratio is low, as errors tend to accumulate over longer horizons. Furthermore, our results highlight directions for future in-depth studies. For instance, while this work provides empirical insights, further investigation is warranted to understand how time series decomposition interacts with shifting patterns: being more effective under less shifting in multivariate setups and under more shifting in univariate setups. Overall, these results suggest that architectural choices should be carefully aligned with the characteristics of the underlying data, with certain configurations consistently outperforming others under specific conditions.

## 4.2 PRACTICAL IMPLICATIONS FROM THE INSIGHTS

Given the insights discussed above, a natural question arises: how can these findings be leveraged to inform better model architecture design for a given time-series dataset? To address this, TIMERECIPE integrates a training-free model selection mechanism based on a LightGBM (Ke et al., 2017) regression model. Specifically, we train a regression model that maps both dataset characteristics and model configurations to their associated rank scores benchmarked by TIMERECIPE. For a new forecasting task, we first compute the relevant data characteristics and then estimate the rank scores for a range of candidate configurations. The configuration with the lowest predicted rank score will be selected.

We evaluate the model selection toolkit on two forecasting scenarios: (i) an in-distribution case involving short-term multivariate forecasting on the ETTh1 dataset (which is used in the TIMERECIPE, though this specific setting is not benchmarked), and (ii) three out-of-distribution cases using a new univariate unemployment forecasting dataset introduced in Time-MMD (Liu et al., 2024a), where the dataset details are provided in Appendix B.1, which is not part of the original TIMERECIPE benchmark. The results, presented in Table 3, and Table 5 in Appendix C.2, demonstrate that even with a simple tree-based approach, TIMERECIPE is capable of selecting models that are closer to the globally optimal architecture than the existing best-performing model. These findings underscore the potential of deploying TIMERECIPE in real-world model selection scenarios, where it is able to make model selections that can surpass the SOTA performance without the need for training.

|  |  |  | | Social_12_S | | | | | | |
| --- | --- | --- | --- | --- | --- | --- | --- | --- | --- | --- |
|  | IN | SD | Fusion | Embed | FF-Type | Rank | MSE | | MAE | |
| TIMERECIPE Best | ✓ | ✗ | Feature | Patch | MLP | 1.0 | 0.0854 | - | 0.2072 | - |
| Existing Best (PatchTST) | ✓ | ✗ | Temporal | Patch | Trans. | 25.5 | 0.0994 | -16.4% | 0.2256 | -8.9% |
| Top-3 Selection | ✓ | ✓ | Feature | Patch | RNN | 14.0 | 0.0950 | -11.2% | 0.2202 | -6.3% |
|  | ✓ | ✓ | Temporal | Invert | RNN | 5.5 | 0.0897 | -5.0% | 0.2123 | -2.4% |
|  | ✗ | ✗ | Temporal | Token | MLP | 70.0 | 0.1310 | -53.4% | 0.2689 | -29.7% |
| **Selection better than existing best: Yes** (Selection 1&2) | | | | | | | | | | |

Table 3: Comparison of model configurations selected by our method against the best-performing existing model and the global best results found through exhaustive design space search. At least one of our top-3 selections consistently outperforms the existing best and closely approaches the optimal configuration on Social_12_S (see Table 5, Appendix C.2, for more and similar observations). This demonstrates the value of TIMERECIPE, even when using a simple tree-based approach.

## 4.3 IMPLICATIONS FOR FUTURE RESEARCH

We identify key benefits of TIMERECIPE that can support future research in time-series forecasting.

**Convenient Framework for Time-Series Module Design and Evaluation.** The TIMERECIPE's canonical architecture provides a practical and extensible framework for testing new component designs in time-series forecasting. A key benefit is the ease with which novel modules can be integrated and evaluated in a standardized pipeline. For example, researchers can introduce new embedding strategies beyond those currently benchmarked and seamlessly insert them into the TIMERECIPE framework. The system can then be used to automatically explore optimal configurations of the remaining components, such as pre-processing or feed-forward types, and thereby facilitate efficient and systematic evaluation of new design choices within a comprehensive forecasting pipeline.

**Implications for Designing Advanced Time-Series Forecasting Models.** TIMERECIPE indicates that no single existing architecture consistently outperforms others across all time-series forecasting scenarios, even with taking recent SOTA into consideration (Shi et al., 2024; Wang et al., 2025; Zhong et al., 2025; Kong et al., 2025). However, rather than interpreting this as a limitation that precludes the development of broadly effective models, we argue that this insight points to the potential of hybrid architectures. Future SOTA models may benefit from integrating multiple architectural components tailored to different data properties. For example, (Ni et al., 2025a) has shown that some temporal patterns are better captured by Transformer-based models, while others are more effectively modeled by MLPs. Extending this intuition, TIMERECIPE aims to reveal that similar complementarities exist across a wide range of design dimensions, such as normalization strategies, embedding schemes (e.g., patch vs. token), or architectural combinations. These findings suggest that next-generation forecasting systems, including foundation models, which are often derived from supervised learning

approaches, could possibly achieve broader robustness and improved generalization by dynamically leveraging hybrid designs tailored to specific forecasting scenarios and pattern types.

**Advancing AutoML for Time-Series Forecasting.** Another important contribution of TIMERECIPE lies in its potential to advance AutoML studies in time-series forecasting. Earlier studies have mainly focused on model-level searches combined with hyperparameter and learning scheme optimization (Alsharef et al., 2022b;a; Shchur et al., 2023; Westergaard et al., 2024). With the canonical architecture introduced by TIMERECIPE, future AutoML for time-series forecasting may enable finer-grained, module-level search across different components of the forecasting pipeline. This approach allows AutoML systems to explore a significantly broader and more flexible design space, in conjunction with traditional hyperparameter and learning scheme tuning. As a result, TIMERECIPE potentially benefits more granular and effective AutoML in time-series applications.

## 5 CONCLUSION

In this work, we introduced TIMERECIPE, a benchmarking framework that systematically evaluates the effectiveness of individual modules for time-series forecasting. By decomposing forecast architectures into modular components, TIMERECIPE allows a fine-grained analysis of design choices across a wide range of datasets and task settings. Our study presents comprehensive benchmark evaluations through more than 10,000 experiments to date, as well as the novel evaluation scope of module-level effectiveness to our best knowledge. We further include the limitation discussion in Appendix D.

Our results highlight two core insights: first, that exhaustive design space search results in model designs that can outperform SOTA; and second, that module effectiveness is highly data-dependent, with no single design universally superior across all scenarios. These findings not only challenge the notion of one-size-fits-all solutions but also motivate future research into adaptive and hybrid model architectures. By releasing our benchmark and toolkit, we aim to support the community in both evaluating existing designs and exploring new ones under a unified, interpretable framework.

## 6 ACKNOWLEDGMENT

The work from Georgia Institute of Technology was partly supported by the NSF (Expeditions CCF-1918770, CAREER IIS-2028586, Medium IIS-1955883, Medium IIS-2403240, Medium IIS-2106961), NIH (1R01HL184139), CDC MInD program, Meta, and Dolby faculty gifts. The work from Emory University was partly supported by NSF CNS-2437345 and by the National Institute of Allergy and Infectious Diseases of the NIH under Award No. 1R01AI197111. The content is the sole responsibility of the authors and does not necessarily represent the views of the NIH.

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

## A  RELATED WORKS

**Supervised Time-Series Forecasting.** Time-series forecasting, predicting future values based on historical observations, is a long-standing challenge across numerous domains. Early approaches have been dominated by statistical methods. Autoregressive Integrated Moving Average (ARIMA) models (Box, 2013) and Exponential Smoothing (ETS) techniques (Hyndman et al., 2008) have become foundational, modeling temporal dependencies and trends through established statistical principles. These methods often assume linearity or specific data structures and can struggle with complex, non-linear patterns present in many real-world datasets.

As a transformative architecture from deep learning, the introduction of the Transformer architecture (Vaswani et al., 2017), with its powerful self-attention mechanism, has marked a pivotal moment for time-series forecasting. Early efforts have focused on enhancing the efficiency and effectiveness of Transformers for long sequential data, a common characteristic of time series. Models like Reformer (Kitaev et al., 2020) have introduced techniques to reduce computational complexity, while Informer (Zhou et al., 2021), Autoformer (Wu et al., 2021), Pyraformer (Liu et al., 2022b), and FEDformer (Zhou et al., 2022) have explored specialized attention mechanisms (e.g., sparse attention, auto-correlation, pyramidal attention) and decomposition strategies (e.g., trend-seasonal decomposition, frequency domain processing) to better capture temporal dependencies in long sequences. Further refinements have addressed specific challenges inherent in time-series data (Liu et al., 2024a; Zhao et al., 2025a). For instance, Non-stationary Transformers (Liu et al., 2022c) have incorporated mechanisms to handle distribution shifts over time, ETSformer (Woo et al., 2022) has integrated principles from classical exponential smoothing, and Crossformer (Zhang & Yan, 2023) has focused on modeling dependencies across different variates in multivariate settings. PatchTST (Nie et al.) has proposed segmenting time series into patches, treating them as tokens, which proves highly effective for long-term forecasting. More recently, iTransformer (Liu et al.) has inverted the roles of embedding and attention layers, achieving strong results.

Despite the success of Transformer variants, recent research has spurred a debate regarding their necessity, demonstrating that simpler architectures, particularly those based on Multi-Layer Perceptrons

(MLPs) or even linear layers, can achieve competitive or superior performance on many benchmarks. DLinear (Zeng et al., 2023) has proposed a simple linear model with decomposition, challenging the complexity of contemporary Transformers. Subsequently, various MLP-based models have emerged, often emphasizing efficiency and specialized designs. LightTS (Zhang & Yan, 2022) has utilized sampling-oriented MLP structures, TSMixer (Chen et al., 2023) has employed an all-MLP architecture with mixing across time steps and features. TimeMixer (Wang et al., 2024a) and TimeMixer++ (Wang et al., 2025) have further reflected on the competent duties of Transformer components and have repurposed the Transformer architecture.

Beyond Transformers and MLPs, other architectural paradigms continue to be explored. Convolutional approaches, such as MICN (Wang et al., 2023) using multi-scale local and global context modeling, SCINet (Liu et al., 2022a) employing sample convolution and interaction and TimesNet (Wu et al.) modeling temporal 2D variations, offer alternative ways to capture temporal features. Recurrent architectures are also being revisited, exhibiting strong performance for forecasting (Kong et al., 2025). Recent explorations also include leveraging pretrained language models for forecasting tasks (Jin et al.; Tan et al., 2024; Liu et al., 2025a). This diverse landscape, i.e., various model architectures, highlights the necessity and urgency of this time-series forecasting recipe at the model level, which is developed in this paper.

**Time-Series Foundation Models.** Time-series foundation models, pre-trained on vast amounts of diverse time-series data, aim to zero-shot adapt to unseen time-series datasets. Existing methods largely follow the architectural design of foundation language models. For instance, Chronos (Ansari et al.) is based on the T5 (Raffel et al., 2020) architecture, TimesFM (Das et al.) employs the decoder-only Transformer architecture, Lag-Llama (Rasul et al.) explicitly leverages the Llama architecture. Recent advancements (Liu et al., 2024c; Shi et al., 2024) address computational scaling while enhancing capacity by introducing mixture of experts (MoE) (Cai et al., 2025) techniques from foundation language models. We envision that the model-level time-series forecasting recipe proposed in this paper provides key insights for the architectural design of the next generation of foundation time-series models.

**Time-Series Benchmarks.** Robust and standardized benchmarks are essential for evaluating the performance of time-series forecasting models, enabling fair comparison and reproducible research. Historically, the M4 and M5 competitions (Makridakis et al., 2020; 2022) have served as crucial benchmarks, providing large collections of diverse time series primarily in business, demographic, and economic domains, along with rigorous evaluation protocols (Makridakis et al., 2020; 2022). Recent efforts focus on providing unified benchmarking frameworks and toolkits. TFB (Qiu et al., 2024), for instance, introduces a scalable suite covering multiple domains and methods, emphasizing reproducibility and systematic, aspect-based analysis. Furthermore, BasicTS (Liang et al., 2022) focuses on benchmarking multivariate time series forecasting, ProbTS (Zhang et al., 2024) benchmarks both point and distributional forecasting, and GIFT-Eval (Aksu et al., 2024) is designed for benchmarking foundation time-series models. Recently, ReC4TS (Liu et al., 2025b) benchmarks how different reasoning strategies enhance zero-shot time-series forecasting. In addition, Time-MMD (Liu et al., 2024a) and CiK (Ashok et al., 2024) are designed for benchmarking multimodal time-series forecasting. Different from all existing time-series benchmarks, our work uniquely focuses on evaluating the effectiveness of individual module-level design choices for time-series forecasting. To the best of our knowledge, this is the first benchmark to systematically address this underexplored yet critical aspect of model design.

**Auto-ML in Time-Series.** Automated Machine Learning (AutoML) aims to automate the end-to-end process of applying machine learning, enhancing accessibility and efficiency. AutoML research in time series can be categorized as: (1) automated feature engineering, which generates relevant temporal features like lags and rolling statistics (Cerqueira et al., 2021; Costa, 2021); (2) automated model selection, searching across diverse model families from statistical methods (e.g., ARIMA) to machine learning (e.g., boosted trees) and deep learning architectures (e.g., LSTMs, Transformers) (Ying et al., 2020; Abdallah et al., 2022; Shchur et al., 2023); (3) hyperparameter optimization, using techniques like Bayesian optimization or evolutionary algorithms to tune model configurations (Wu et al., 2022; Fristiana et al., 2024); (4) neural architecture search, which specifically automates the design of deep learning model structures suitable for capturing complex temporal patterns (Rakhshani et al., 2020; Wu et al., 2023).

## B  DETAILED EVALUATION SETUP

### B.1  ADDITIONAL DATASET EXPLANATION

In this section, we will show more details of the datasets and our selection.

**ETT.** The ETT datset, alias Electricity Transformer Temperature (Zhou et al., 2021), contains collected oil temperature and electricity load data per minute of two Chinese stations between 2016/07 to 2018/07. The original data is then aggregated every one hour (ETTh1/2) or every 15 minutes (ETTm1/2).

**ILI.** The ILI dataset, alias Influenza-like Illness statistics, contains weekly influenza surveillance collected and released by the CDC [2] starting from 1997-98 influenza season.

**ECL.** The ECL dataset, alias Electricity Consumption Load (Asuncion et al., 2007), contains the hourly electricity consumption history of 321 clients covering two years.

**Weather.** The Weather dataset contains hourly meteorological data from 1,600 locations across the U.S. between 2010 and 2013.

**Exchange.** The Exchange Rate dataset(Lai et al., 2018) contains daily exchange rates of U.S. dollars in eight foreign countries, including Australia, Britain, Canada, Switzerland, China, Japan, New Zealand, and Singapore, between 1990 and 2016.

**PEMS.** The PEMS dataset, alias Performance Measurement System, contains traffic data per minute from 325 sensors in the Bay Area collected by California Transportation Agencies (CalTrans) between January 2017 and May 2017. We adopted data from 4 sparsely located sensors in our evaluations.

**M4.** The M4 dataset contains 100,000 domain-specific series from domains such as macroeconomics, microeconomics, finance, industry, and demography. Each time series is aggregated in various granularities, from hourly to yearly.

Our collected data covers multiple domains, multiple granularities, and multiple time ranges, ensuring plenty of diversity like the corresponding series. By evaluating such datasets, we guarantee that the outputted recipe from our method maintains considerable coverage in real-world time-series tasks.

**Time-MMD.** The Time-MMD dataset (Liu et al., 2024a) is a multimodal time-series benchmark covering nine domains, such as economics, agriculture, and security. It consists of temporally aligned numerical time-series data and corresponding textual information collected from news articles and reports.

In this work, we use only the numerical time-series component for evaluation. Since Time-MMD is not included in the original TIMERECIPE benchmark, it serves as an out-of-distribution testbed to assess whether the architectural insights derived from Table 2 and our model selection strategy generalize beyond the benchmark datasets.

### B.2  EVALUATION METRIC FORMULATION

We follow the common evaluation metrics across diverse forecasting datasets (Oreshkin et al., 2019; Liu et al., 2022a; Wang et al., 2024a). We evaluate the LTSF dataset and PEMS dataset by two metrics: mean squared error (MSE) and mean absolute error (MAE), aiming to evaluate the errors under the same scale. Given a predicted sequence $\hat{\mathbf{Y}} = \{\hat{x}_{t+1}, \ldots, \hat{x}_{t+H}\}$ and corresponding ground truth $\mathbf{Y} = \{x_{t+1}, \ldots, x_{t+H}\}$ where $\mathbf{Y}, \hat{\mathbf{Y}} \in \mathbb{R}^{H \times d_X}$, these two metrics are calculated by:

$$\text{MSE} = \sum_{i=1}^{H} \frac{(x_{t+i} - \hat{x}_{t+i})^2}{H} \tag{5}$$

$$\text{MAE} = \sum_{i=1}^{H} \frac{|x_{t+i} - \hat{x}_{t+i}|}{H} \tag{6}$$

---

[2] https://gis.cdc.gov/grasp/fluview/fluportaldashboard.html

Meanwhile, we evaluate the M4 dataset by three metrics: Symmetric Mean Absolute Percentage Error (SMAPE), Mean Absolute Scaled Error (MASE), and Overall Weighted Average (OWA), to normalize the errors across vast scales inside the dataset:

$$\text{SMAPE} = \frac{200}{H} \sum_{i=1}^{H} \frac{|x_{t+i} - \hat{x}_{t+i}|}{|x_{t+i}| + |\hat{x}_{t+i}|} \tag{7}$$

$$\text{MASE} = \frac{1}{H} \sum_{i=1}^{H} \frac{|x_{t+i} - \hat{x}_{t+i}}{\frac{1}{H-S} \sum_{j=s+1}^{H} |x_{t+j} - x_{t+j-s}|} \tag{8}$$

$$\text{OWA} = \frac{1}{2} \left[ \frac{\text{SMAPE}}{\text{SMAPE}_{\text{Naive2}}} + \frac{\text{MASE}}{\text{MASE}_{\text{Naive2}}} \right] \tag{9}$$

### B.3 DATASET PROPERTIES MEASUREMENT

We define the data properties involved in TIMERECIPE following TFB (Qiu et al., 2024).

**Trend.** refers to the long-term changes of time-series. As shown in Algorithm 1, we involve Seasonal and Trend decomposition using Loess (STL) (Cleveland et al., 1990) as a decomposition function to calculate the trending values.

**Seasonality.** refers to the changes in time-series that repeat every certain period. Similar to the calculation of Trend, we involve STL to calculate the seasonality, which is shown in Algorithm 2.

**Stationarity.** refers to the indicator of whether a time-series approximately satisfies all of the following: 1) the mean of any observation inside the series is constant, 2) the variance of any observation is finite, 3) the covariance between any two observations depends only on their distance. The calculation is shown in Algorithm 3, where ADF refers to the Augmented Dickey-Fuller (ADF) test.

**Shifting.** refers to time-series changes upon a certain direction over time. Given a threshold $m$, we calculate the shifting indicators in Algorithm 4.

**Transition** refers to the covariance of the transition matrix across symbols from 3-value windows. We calculate the transition values as shown in Algorithm 5,

**Correlation.** refers to the possibility that different channels of a multivariate sequence share a similar distribution. As shown in algorithm 6, we calculate the correlation referring to Catch22 (Lubba et al., 2019) library, which is designed to extract 22 features from time series.

---

**Algorithm 1:** Calculating Trend Strength of Time-Series

**Input** : Time-Series $\mathbf{X} \in \mathbb{R}^{T \times 1}$
**Output :** Trend strength $\beta$ of $\mathbf{X}$
1  S, T, R $\leftarrow$ STL(X);
2  **return** $\beta \leftarrow \max(0, 1 - \frac{\text{var}(R)}{\text{var}(T+R)})$;

---

**Algorithm 2:** Calculating Seasonality Strength of Time-Series

**Input** : Time-Series $\mathbf{X} \in \mathbb{R}^{T \times 1}$
**Output :** Seasonality strength $\zeta$ of $\mathbf{X}$
1  S, T, R $\leftarrow$ STL(X);
2  **return** $\zeta \leftarrow \max(0, 1 - \frac{\text{var}(R)}{\text{var}(S+R)})$;

---

### B.4 HYPERPARAMETER TUNING

For the modules involved in our evaluation, several components (e.g., series decomposition) are inherently non-parametric and do not introduce additional hyperparameters. For modules that do

---

**Algorithm 3:** Calculating Stationarity Value of Time-Series

---

**Input** : Time-Series $\mathbf{X} \in \mathbb{R}^{T \times 1}$
**Output** : Stationarity value $\gamma$ of $\mathbf{X}$
1   $s \leftarrow \text{ADF}(\mathbf{X})$;
2   **return** $\gamma \leftarrow (s \leq 0.05)$;

---

**Algorithm 4:** Calculating Shifting Values of Time-Series

---

**Input** : Time-Series $\mathbf{X} \in \mathbb{R}^{T \times 1}$, number of thresholds $m$
**Output** : Shifting value $\delta$ of $\mathbf{X}$
1   $Z \leftarrow Z - \text{normalize}(\mathbf{X})$;
2   $Z_{min} \leftarrow \min(Z),\ Z_{max} \leftarrow \max(Z)$;
3   $S \leftarrow \{s_i | s_i \leftarrow Z_{min} + (i-1)\frac{Z_{max} - Z_{min}}{m}, i \in [1, m]\}$;
4   **for** $i = 1$ **to** $m$ **do**
5     $K_i \leftarrow \{j \mid Z_j > s_i, j \in [1, T]\}$;
6     $M_i \leftarrow \text{median}(K_i)$
7   **end**
8   $M' \leftarrow \text{Min} - \text{Max Normalization}(M)$;
9   **return** $\delta \leftarrow \text{median}(M')$;

---

involve architectural hyperparameters, we tune key parameters such as hidden dimension, number of layers, and method-specific configurations (e.g., patch length). This design ensures that each module combination is sufficiently expressive without being over-parameterized, thereby avoiding excessive training cost while maintaining fair architectural comparisons across different module combinations.

For optimization-related hyperparameters, such as learning rate, we follow the standard training protocols provided by the Time-Series Library. In particular, we fix these training hyperparameters, such as using a unified learning rate across most experiments, which is a common practice in neural architecture search studies Ying et al. (2019); Qin et al. (2022). This strategy isolates the impact of architectural variations and ensures that performance differences are primarily attributable to module design rather than training configurations.

## C  DETAILED RESULTS

### C.1  COMPREHENSIVE CLAIMS (SECTION 4.1.2 CONTD.)

Here we present all the claims on the correlations between the effectiveness of the module and the data characteristics that are statistically significant (e.g., p-value$\leq$0.05). The comprehensive results, including the t-test $p$-value, are shown in Table 4.

### C.2  MODEL SELECTION TOOLKIT EVALUATION (SECTION 4.2 CONTD.)

Here, we present additional evaluation results for our training-free model selection approach on `ETTh1_24_M`, `Environment_48_S`, and `Security_12_S`. As shown in Table 5, these results are consistent with the conclusions drawn from Table 3.

### C.3  PERFORMANCE RANKING DETAILS (SECTION 4.1.1 CONTD.)

To present a clearer comparison of the performance across different module combinations for each dataset and forecasting horizon, we present the top-10 ranked results. The ranking is based on the average of the ranks obtained from two metrics: MSE and MAE. Specifically, if a module combination ranks 1st in MSE and 2nd in MAE, its final rank score is computed as the average, i.e., 1.5. The detailed rankings are summarized in the following tables.

For clarity, we denote each experimental setup using the format "{dataset_horizon_feature}". For instance, a multivariate forecasting task on the ETTh1 dataset with a forecasting horizon of 96 is

---

**Algorithm 5:** Calculating Transition Values of Time-Series

---

**Input** : Time-Series $\mathbf{X} \in \mathbb{R}^{T \times 1}$
**Output** : Transition value $\Delta \in (0, \frac{1}{3})$ of $\mathbf{X}$

1   $\tau \leftarrow$ First zero crossing of $X$'s autocorrelation;
2   $Y \leftarrow$ Downsampling $X$ with stride $\tau$;
3   $I \leftarrow \text{argsort}(Y)$;
4   $T' \leftarrow \text{length}(Y)$ **for** $j \in [0 : T']$ **do**
5   |   $Z_j \leftarrow \lfloor \frac{3I_j}{T'} \rfloor$;
6   **end**
7   $M \leftarrow [0]^{3 \times 3}$ **for** $j \in [0 : T']$ **do**
8   |   $M[Z_j - 1][Z_{j+1} - 1] \leftarrow M[Z_j - 1][Z_{j+1} - 1] + 1$
9   **end**
10   $M' \leftarrow \frac{1}{T'} M$;
11   $C \leftarrow$ covariance matrix between the columns of $M'$;
12   **return** $\Delta \leftarrow tr(C)$

---

**Algorithm 6:** Calculating Correlation Values of Time-Series

---

**Input** : Multivariate Time-Series $\mathbf{X} \in \mathbb{R}^{T \times N}$
**Output** : Correlation value $\eta \in (0, 1)$ of $\mathbf{X}$

1   $F = \langle F^1, \ldots, F^N \rangle \in \mathcal{R}^{22 \times N} \leftarrow \text{Catch22}(\mathbf{X})$;
2   $P = \{ r_{Pearson}(F^i, F^j) \mid i \in [1, N], j \in [i+1, N], i, j \in N^* \}$;
3   **return** $\eta \leftarrow \text{mean}(P) + \frac{1}{1 + \text{var}(P)}$

---

represented as "{ETTh1_96_M}". In particular, we present the results on multivariate forecasting on LTSF datasets through Table 6∼ 13, univariate forecasting results on LTSF datasets through Table 19∼ 26, multivariate forecasting results on PEMS datasets through Table 27∼ 30, univariate forecasting results on PEMS datasets through Table 19∼ 26, and univariate forecasting results on M4 datasets on Table 18.

## D   LIMITATION DISCUSSION

In this work, we break down time-series models into five key design modules and benchmark a wide range of their combinations. Our framework identifies model configurations that can perform significantly better than existing standalone models. More importantly, our findings offer practitioners clear, understandable, and direct guidance. This ensures that the field advances with a deeper understanding and a more structured methodological approach, rather than through unsystematic exploration. However, we acknowledge that this work has the following limitations:

**Not all time-series designs are covered.** We recognize that TIMERECIPE does not include every existing module design in time-series forecasting. The designs not covered generally fall into two categories. First, there are highly specialized designs, such as those in Crossformer (Zhang & Yan, 2023). These are specifically created for the Transformer architecture to capture dependencies across time and dimensions but are difficult to apply to other model architectures. Second, some model architectures are not yet widely adopted, such as models based on large language models (LLMs), for example, Time-LLM (Jin et al.). Such models often have high computational costs, and their effectiveness is still a subject of ongoing discussion (Tan et al., 2024).

Nevertheless, as the first-of-its-kind benchmark, TIMERECIPE has covered a significant and representative portion of the design space for supervised time-series forecasting at the modular level.

**The findings are primarily empirical.** The insights from TIMERECIPE are based on extensive empirical evaluations. While these experiments provide robust and thorough evaluations, and the toolkit recommends model architectures based on these empirical insights, a deeper theoretical analysis of why certain module designs are most effective under specific time-series forecasting scenarios is beyond the current scope of this benchmark study.

| Setting | Property | Direction | Choice | $p$-value |
|---------|----------|-----------|--------|-----------|
| Multivariate | N-Feature | ↘ | IN=True | 0.000 |
| | | ↘ | Fusion=Temporal | 0.000 |
| | | ↗ | Embed=Invert | 0.000 |
| | | ↗ | FF=RNN | 0.000 |
| | HL-Ratio | ↘ | Embed=Token | 0.002 |
| | | ↘ | FF=RNN | 0.000 |
| | | ↗ | IN=True | 0.000 |
| | | ↗ | Fusion=Temporal | 0.014 |
| | Correlation | ↘ | IN=True | 0.000 |
| | | ↘ | Fusion=Temporal | 0.000 |
| | | ↗ | Embed=Invert | 0.000 |
| | | ↗ | FF=Transformer | 0.000 |
| | Transition | ↘ | Embed=Invert | 0.012 |
| | | ↘ | FF=Transformer | 0.000 |
| | | ↗ | IN=True | 0.000 |
| | | ↗ | Fusion=Temporal | 0.000 |
| | Shifting | ↘ | SD=True | 0.034 |
| | | ↘ | Embed=Invert | 0.000 |
| | | ↘ | FF=RNN | 0.000 |
| | | ↗ | IN=True | 0.000 |
| | | ↗ | Fusion=Temporal | 0.000 |
| | Seasonality | ↘ | IN=True | 0.000 |
| | | ↘ | Fusion=Temporal | 0.000 |
| | | ↗ | Embed=Token | 0.000 |
| | | ↗ | FF=Transformer | 0.000 |
| | Trend | ↘ | Embed=Token | 0.006 |
| | | ↘ | FF=RNN | 0.000 |
| | | ↗ | IN=True | 0.000 |
| | | ↗ | Fusion=Temporal | 0.001 |
| | | ↗ | Embed=Patch | 0.000 |
| | Stationarity | ↘ | Embed=Token | 0.021 |
| | | ↘ | FF=RNN | 0.000 |
| | | ↗ | Fusion=Temporal | 0.026 |
| Univariate | HL-Ratio | ↗ | Fusion=Temporal | 0.022 |
| | Transition | ↗ | SD=True | 0.000 |
| | Shifting | ↗ | SD=True | 0.000 |
| | | ↗ | Fusion=Temporal | 0.011 |
| | Seasonality | ↘ | Fusion=Temporal | 0.008 |
| | | ↗ | FF=Transformer | 0.000 |
| | Trend | ↗ | SD=True | 0.000 |
| | | ↗ | Fusion=Temporal | 0.000 |
| | | ↗ | Embed=None | 0.044 |
| | Stationarity | ↗ | SD=True | 0.002 |

Table 4: Statistical correlation between data/task properties and the effectiveness of specific architectural module choices in improving ranking performance (lower is better). Each row indicates whether a particular module configuration (e.g., input normalization, fusion type, embedding, feedforward block) is associated with a significantly lower or higher rank under certain data characteristics. The direction arrows (↘ or ↗) denote whether the choice is favored when the property is low or high, respectively, and the $p$-value quantifies the strength of the statistical association. Results are grouped by multivariate and univariate forecasting settings.

Nonetheless, our claims are supported by comprehensive benchmarking, over 10,000 experiments across diverse datasets, forecasting horizons, and task settings. The results are also statistically meaningful, with reported outcomes averaged over multiple random seeds for statistical robustness. Furthermore, statistical hypothesis testing validates the correlation findings.

| | IN | SD | Fusion | Embed | FF-Type | Rank | MSE | | MAE | |
|---|---|---|---|---|---|---|---|---|---|---|
| | | | | | ETTh1_24_M | | | | | |
| TIMERECIPE Best | ✓ | ✗ | Feature | Patch | MLP | 1.5 | 0.2963 | - | 0.3467 | - |
| Existing Best (PatchTST) | ✓ | ✗ | Temporal | Patch | Trans. | 4.0 | 0.2988 | -0.9% | 0.3520 | -1.5% |
| | ✓ | ✗ | Feature | Patch | MLP | 1.5 | 0.2963 | 0.0% | 0.3467 | 0.0% |
| Top-3 Selection | ✓ | ✗ | Temporal | Patch | MLP | 20.5 | 0.3096 | -4.3% | 0.3586 | -3.4% |
| | ✓ | ✗ | Temporal | Patch | Trans. | 4.0 | 0.2988 | -0.9% | 0.3520 | -1.5% |
| **Selection better than existing best: Yes** (Selection 1&3) | | | | | | | | | | |
| | | | | | Environment_48_S | | | | | |
| | IN | SD | Fusion | Embed | FF-Type | Rank | MSE | | MAE | |
| TIMERECIPE Best | ✓ | ✓ | Temporal | Patch | RNN | 2.0 | 0.2912 | - | 0.3786 | - |
| Existing Best (DLinear) | ✓ | ✓ | Temporal | None | MLP | 16.0 | 0.2950 | -1.3% | 0.3825 | -1.0% |
| | ✓ | ✗ | Feature | Patch | RNN | 21.5 | 0.2987 | -2.5% | 0.3824 | -1.0% |
| Top-3 Selection | ✓ | ✓ | Feature | Patch | RNN | 36.5 | 0.3068 | -5.3% | 0.3936 | -4.0% |
| | ✓ | ✓ | Feature | Token | RNN | 2.5 | 0.2920 | -0.2% | 0.3781 | +0.1% |
| **Selection better than existing best: Yes** (Selection 3) | | | | | | | | | | |
| | | | | | Security_12_S | | | | | |
| | IN | SD | Fusion | Embed | FF-Type | Rank | MSE | | MAE | |
| TIMERECIPE Best | ✓ | ✗ | Feature | None | RNN | 2.5 | 74.2170 | - | 4.0465 | - |
| Existing Best (DLinear) | ✓ | ✓ | Temporal | None | MLP | 26.0 | 75.2914 | -1.4% | 4.3136 | -6.6% |
| | ✓ | ✗ | Feature | Invert | Trans. | 56.0 | 86.9113 | -17.1% | 5.1710 | -27.8% |
| Top-3 Selection | ✓ | ✓ | Temporal | Patch | RNN | 19.0 | 75.2749 | -1.4% | 4.2357 | -4.7% |
| | ✓ | ✓ | Temporal | Patch | MLP | 14.5 | 74.9425 | -0.9% | 4.2124 | -0.4% |
| **Selection better than existing best: Yes** (Selection 2&3) | | | | | | | | | | |

Table 5: Comparison of model configurations selected by our method against the best-performing existing model and the global best results found through exhaustive design space search. At least one of our top-3 selections consistently outperforms the existing best and closely approaches the optimal configuration on ETTh1_24_M, Environment_48_S, and Security_12_S. The results demonstrate the value of TIMERECIPE, even when using a simple tree-based approach.

| Setup | IN | SD | Fusion | Embed | FF-Type | MSE | Rank | MAE | Rank | Total |
|---|---|---|---|---|---|---|---|---|---|---|
| | ✓ | ✗ | Feature | Patch | MLP | **0.3761** | **1** | 0.3920 | 3 | **2.0** |
| | ✓ | ✓ | Feature | Patch | MLP | 0.3783 | 3 | **0.3916** | **2** | **2.5** |
| | ✗ | ✗ | Temporal | Patch | RNN | 0.3799 | 5 | 0.3952 | 5 | 5.0 |
| | ✗ | ✓ | Feature | Patch | MLP | 0.3794 | 4 | 0.4011 | 8 | 6.0 |
| ETTh1_96_M | ✗ | ✓ | Temporal | Patch | RNN | 0.3803 | 7 | 0.3971 | 6 | 6.5 |
| | ✓ | ✗ | Temporal | Frequency | MLP | **0.3776** | **2** | 0.4019 | 11 | 6.5 |
| | ✓ | ✓ | Temporal | Patch | RNN | 0.3828 | 13 | **0.3915** | **1** | 7.0 |
| | ✓ | ✗ | Temporal | None | MLP | 0.3801 | 6 | 0.4019 | 10 | 8.0 |
| | ✓ | ✗ | Temporal | Patch | RNN | 0.3837 | 14 | 0.3930 | 4 | 9.0 |
| | ✓ | ✗ | Feature | None | Trans. | 0.3811 | 9 | 0.4018 | 9 | 9.0 |
| | ✓ | ✗ | Feature | Patch | MLP | 0.4328 | 6 | **0.4238** | **2** | **4.0** |
| | ✓ | ✓ | Temporal | Patch | RNN | 0.4314 | 4 | 0.4265 | 5 | **4.5** |
| | ✗ | ✗ | Temporal | Patch | RNN | 0.4314 | 3 | 0.4285 | 6 | **4.5** |
| | ✗ | ✓ | Temporal | Patch | RNN | 0.4339 | 9 | **0.4216** | **1** | 5.0 |
| ETTh1_192_M | ✓ | ✗ | Temporal | Frequency | MLP | **0.4306** | **2** | 0.4314 | 9 | 5.5 |
| | ✓ | ✗ | Temporal | None | MLP | **0.4292** | **1** | 0.4318 | 10 | 5.5 |
| | ✗ | ✗ | Feature | Patch | MLP | 0.4325 | 5 | 0.4310 | 8 | 6.5 |
| | ✓ | ✓ | Feature | Patch | MLP | 0.4347 | 12 | 0.4260 | 4 | 8.0 |
| | ✓ | ✗ | Feature | None | Trans. | 0.4336 | 8 | 0.4322 | 12 | 10.0 |
| | ✓ | ✗ | Temporal | Patch | MLP | 0.4334 | 7 | 0.4343 | 17 | 12.0 |
| | ✓ | ✗ | Temporal | Invert | MLP | **0.4626** | **1** | **0.4462** | **2** | **1.5** |
| | ✓ | ✗ | Temporal | Patch | RNN | 0.4766 | 7 | **0.4433** | **1** | **4.0** |
| | ✓ | ✗ | Temporal | Patch | Trans. | 0.4713 | 3 | 0.4510 | 6 | 4.5 |
| | ✓ | ✗ | Temporal | None | MLP | **0.4703** | **2** | 0.4526 | 10 | 6.0 |
| ETTh1_336_M | ✓ | ✗ | Feature | Patch | MLP | 0.4799 | 12 | 0.4467 | 3 | 7.5 |
| | ✓ | ✓ | Temporal | Invert | MLP | 0.4774 | 8 | 0.4518 | 7 | 7.5 |
| | ✗ | ✓ | Temporal | Patch | RNN | 0.4746 | 5 | 0.4541 | 14 | 9.5 |
| | ✓ | ✓ | Temporal | Patch | RNN | 0.4835 | 16 | 0.4482 | 4 | 10.0 |
| | ✓ | ✗ | Temporal | Frequency | MLP | 0.4744 | 4 | 0.4554 | 17 | 10.5 |
| | ✓ | ✓ | Feature | None | Trans. | 0.4790 | 9 | 0.4538 | 13 | 11.0 |
| | ✓ | ✗ | Temporal | Invert | MLP | **0.4716** | **1** | **0.4698** | **1** | **1.0** |
| | ✓ | ✗ | Temporal | Patch | Trans. | **0.4741** | **2** | 0.4722 | 3 | **2.5** |
| | ✓ | ✓ | Temporal | Invert | MLP | 0.4818 | 4 | 0.4730 | 4 | 4.0 |
| | ✓ | ✗ | Temporal | Patch | RNN | 0.4855 | 7 | **0.4719** | **2** | 4.5 |
| ETTh1_720_M | ✓ | ✗ | Feature | None | Trans. | 0.4792 | 3 | 0.4792 | 7 | 5.0 |
| | ✓ | ✗ | Temporal | None | MLP | 0.4842 | 5 | 0.4787 | 6 | 5.5 |
| | ✓ | ✓ | Temporal | Patch | RNN | 0.4853 | 6 | 0.4730 | 5 | 5.5 |
| | ✓ | ✓ | Feature | Invert | Trans. | 0.4979 | 8 | 0.4825 | 9 | 8.5 |
| | ✓ | ✓ | Temporal | None | MLP | 0.4982 | 10 | 0.4819 | 8 | 9.0 |
| | ✓ | ✓ | Temporal | Frequency | MLP | 0.5009 | 12 | 0.4840 | 11 | 11.5 |

Table 6: Top-10 configurations for the ETTh1 dataset multivariate forecasting. IN: Instance Norm, SD: Series Decomposition. ✓ indicates module used, ✗ indicates not used. Red/blue highlights indicate best and second-best performances.

| Setup | IN | SD | Fusion | Embed | FF-Type | MSE | Rank | MAE | Rank | Total |
|---|---|---|---|---|---|---|---|---|---|---|
| | ✓ | ✓ | Feature | Patch | MLP | **0.2888** | **2** | **0.3392** | **1** | **1.5** |
| | ✓ | ✓ | Temporal | Patch | RNN | 0.2895 | 3 | **0.3397** | **2** | **2.5** |
| | ✗ | ✗ | Temporal | Patch | RNN | **0.2877** | **1** | 0.3422 | 5 | 3.0 |
| | ✓ | ✗ | Feature | Patch | MLP | 0.2904 | 4 | 0.3406 | 3 | 3.5 |
| ETTh2_96_M | ✓ | ✗ | Temporal | Patch | RNN | 0.2917 | 7 | 0.3410 | 4 | 5.5 |
| | ✓ | ✓ | Temporal | Invert | MLP | 0.2913 | 5 | 0.3432 | 9 | 7.0 |
| | ✓ | ✓ | Temporal | Frequency | MLP | 0.2920 | 8 | 0.3426 | 6 | 7.0 |
| | ✓ | ✓ | Temporal | None | MLP | 0.2915 | 6 | 0.3430 | 8 | 7.0 |
| | ✓ | ✗ | Feature | Patch | RNN | 0.2944 | 11 | 0.3427 | 7 | 9.0 |
| | ✓ | ✗ | Temporal | Frequency | MLP | 0.2927 | 9 | 0.3440 | 11 | 10.0 |
| | ✓ | ✓ | Temporal | Patch | RNN | **0.3684** | **2** | **0.3891** | **1** | **1.5** |
| | ✓ | ✗ | Feature | Patch | MLP | **0.3677** | **1** | 0.3923 | 4 | **2.5** |
| | ✓ | ✓ | Feature | Patch | MLP | 0.3705 | 3 | **0.3904** | **2** | **2.5** |
| | ✓ | ✓ | Temporal | Patch | MLP | 0.3722 | 6 | 0.3917 | 3 | 4.5 |
| ETTh2_192_M | ✓ | ✗ | Temporal | Patch | RNN | 0.3718 | 5 | 0.3934 | 5 | 5.0 |
| | ✓ | ✓ | Temporal | None | MLP | 0.3714 | 4 | 0.3952 | 11 | 7.5 |
| | ✓ | ✓ | Temporal | Frequency | MLP | 0.3734 | 10 | 0.3937 | 6 | 8.0 |
| | ✓ | ✗ | Temporal | None | MLP | 0.3723 | 7 | 0.3947 | 10 | 8.5 |
| | ✓ | ✗ | Feature | Patch | RNN | 0.3731 | 8 | 0.3953 | 12 | 10.0 |
| | ✓ | ✗ | Temporal | Invert | MLP | 0.3745 | 13 | 0.3943 | 8 | 10.5 |
| | ✓ | ✓ | Temporal | Patch | RNN | 0.4142 | 3 | **0.4269** | **1** | **2.0** |
| | ✓ | ✗ | Feature | Invert | Trans. | **0.4131** | **2** | 0.4292 | 6 | **4.0** |
| | ✓ | ✓ | Temporal | Invert | MLP | 0.4159 | 5 | 0.4286 | 3 | **4.0** |
| | ✓ | ✗ | Temporal | Patch | RNN | 0.4164 | 6 | 0.4289 | 5 | 5.5 |
| ETTh2_336_M | ✓ | ✗ | Feature | None | Trans. | 0.4173 | 8 | 0.4286 | 4 | 6.0 |
| | ✓ | ✓ | Temporal | Patch | MLP | 0.4158 | 4 | 0.4293 | 9 | 6.5 |
| | ✓ | ✗ | Feature | Patch | RNN | 0.4166 | 7 | 0.4294 | 10 | 8.5 |
| | ✓ | ✗ | Temporal | Patch | MLP | 0.4196 | 11 | 0.4292 | 7 | 9.0 |
| | ✓ | ✗ | Temporal | Frequency | MLP | 0.4187 | 9 | 0.4306 | 11 | 10.0 |
| | ✓ | ✓ | Feature | Patch | MLP | 0.4206 | 13 | 0.4292 | 8 | 10.5 |
| | ✓ | ✓ | Temporal | Invert | RNN | 0.4269 | 21 | **0.4286** | **2** | 11.5 |
| | ✓ | ✗ | Feature | Invert | Trans. | **0.4218** | **1** | **0.4426** | **2** | **1.5** |
| | ✓ | ✓ | Temporal | Invert | MLP | 0.4262 | 4 | **0.4426** | **1** | **2.5** |
| | ✓ | ✗ | Feature | None | Trans. | 0.4257 | 3 | 0.4443 | 3 | 3.0 |
| | ✓ | ✓ | Temporal | Patch | MLP | **0.4244** | **2** | 0.4446 | 5 | 3.5 |
| ETTh2_720_M | ✓ | ✗ | Temporal | Patch | RNN | 0.4265 | 5 | 0.4445 | 4 | 4.5 |
| | ✓ | ✗ | Temporal | None | MLP | 0.4312 | 7 | 0.4455 | 6 | 6.5 |
| | ✓ | ✓ | Temporal | None | MLP | 0.4290 | 6 | 0.4456 | 7 | 6.5 |
| | ✓ | ✗ | Temporal | Patch | MLP | 0.4314 | 8 | 0.4483 | 10 | 9.0 |
| | ✓ | ✗ | Temporal | Invert | MLP | 0.4320 | 10 | 0.4458 | 8 | 9.0 |
| | ✓ | ✓ | Temporal | Invert | RNN | 0.4326 | 11 | 0.4464 | 9 | 10.0 |

Table 7: Top-10 configurations for the ETTh2 dataset multivariate forecasting. IN: Instance Norm, SD: Series Decomposition. ✓ indicates module used, ✗ indicates not used. Red/blue highlights indicate best and second-best performances.

| Setup | IN | SD | Fusion | Embed | FF-Type | MSE | Rank | MAE | Rank | Total |
|---|---|---|---|---|---|---|---|---|---|---|
| | ✓ | ✓ | Feature | Patch | MLP | **0.3274** | **1** | 0.3647 | 3 | **2.0** |
| | ✓ | ✗ | Temporal | Patch | RNN | 0.3281 | 3 | **0.3640** | **2** | **2.5** |
| | ✓ | ✓ | Feature | Patch | RNN | **0.3280** | **2** | 0.3654 | 4 | 3.0 |
| | ✓ | ✓ | Temporal | Patch | RNN | 0.3308 | 8 | **0.3628** | **1** | 4.5 |
| ETTm1_96_M | ✓ | ✗ | Temporal | Patch | MLP | 0.3290 | 4 | 0.3661 | 7 | 5.5 |
| | ✓ | ✗ | Temporal | None | MLP | 0.3294 | 6 | 0.3655 | 5 | 5.5 |
| | ✓ | ✓ | Temporal | Invert | RNN | 0.3292 | 5 | 0.3669 | 9 | 7.0 |
| | ✓ | ✗ | Feature | None | Trans | 0.3311 | 10 | 0.3659 | 6 | 8.0 |
| | ✓ | ✗ | Feature | Patch | MLP | 0.3308 | 9 | 0.3662 | 8 | 8.5 |
| | ✓ | ✓ | Temporal | Patch | MLP | 0.3299 | 7 | 0.3682 | 11 | 9.0 |
| | ✓ | ✓ | Temporal | Patch | RNN | 0.3681 | 3 | **0.3830** | **1** | **2.0** |
| | ✓ | ✓ | Feature | Patch | MLP | **0.3669** | **2** | 0.3851 | 3 | **2.5** |
| | ✓ | ✗ | Temporal | Patch | RNN | 0.3706 | 6 | **0.3836** | **2** | 4.0 |
| | ✓ | ✗ | Temporal | Patch | MLP | **0.3667** | **1** | 0.3881 | 8 | 4.5 |
| ETTm1_192_M | ✓ | ✓ | Temporal | Patch | MLP | 0.3688 | 4 | 0.3872 | 6 | 5.0 |
| | ✓ | ✓ | Feature | Patch | RNN | 0.3694 | 5 | 0.3862 | 5 | 5.0 |
| | ✓ | ✗ | Feature | Patch | MLP | 0.3710 | 7 | 0.3855 | 4 | 5.5 |
| | ✓ | ✓ | Temporal | None | MLP | 0.3722 | 8 | 0.3890 | 9 | 8.5 |
| | ✓ | ✗ | Temporal | None | MLP | 0.3729 | 12 | 0.3878 | 7 | 9.5 |
| | ✓ | ✓ | Temporal | Freq | MLP | 0.3724 | 9 | 0.3893 | 10 | 9.5 |
| | ✓ | ✓ | Temporal | Patch | RNN | 0.4001 | 3 | **0.4065** | **1** | **2.0** |
| | ✓ | ✓ | Feature | Patch | MLP | 0.4003 | 4 | 0.4082 | 4 | **4.0** |
| | ✓ | ✗ | Temporal | Patch | MLP | **0.4000** | **2** | 0.4100 | 7 | 4.5 |
| | ✓ | ✗ | Temporal | Patch | RNN | 0.4025 | 7 | **0.4069** | **2** | 4.5 |
| ETTm1_336_M | ✓ | ✓ | Temporal | Patch | MLP | **0.3979** | **1** | 0.4109 | 9 | 5.0 |
| | ✓ | ✓ | Feature | Patch | RNN | 0.4003 | 5 | 0.4093 | 5 | 5.0 |
| | ✓ | ✗ | Feature | Patch | MLP | 0.4032 | 9 | 0.4073 | 3 | 6.0 |
| | ✓ | ✓ | Temporal | Invert | MLP | 0.4030 | 8 | 0.4097 | 6 | 7.0 |
| | ✓ | ✓ | Temporal | Freq | MLP | 0.4022 | 6 | 0.4107 | 8 | 7.0 |
| | ✓ | ✓ | Temporal | None | MLP | 0.4058 | 12 | 0.4110 | 10 | 11.0 |
| | ✓ | ✗ | Temporal | Patch | MLP | **0.4577** | **2** | **0.4424** | **2** | **2.0** |
| | ✓ | ✓ | Temporal | Patch | MLP | **0.4558** | **1** | 0.4441 | 3 | **2.0** |
| | ✓ | ✓ | Temporal | Patch | RNN | 0.4628 | 4 | **0.4393** | **1** | **2.5** |
| | ✗ | ✓ | Temporal | Patch | MLP | 0.4579 | 3 | 0.4486 | 10 | 6.5 |
| ETTm1_720_M | ✓ | ✓ | Temporal | Invert | MLP | 0.4653 | 7 | 0.4467 | 6 | 6.5 |
| | ✓ | ✗ | Temporal | Patch | RNN | 0.4682 | 11 | 0.4462 | 4 | 7.5 |
| | ✓ | ✗ | Feature | Patch | MLP | 0.4679 | 10 | 0.4463 | 5 | 7.5 |
| | ✓ | ✓ | Feature | Patch | RNN | 0.4650 | 6 | 0.4485 | 9 | 7.5 |
| | ✓ | ✓ | Temporal | Freq | MLP | 0.4648 | 5 | 0.4487 | 11 | 8.0 |
| | ✓ | ✓ | Temporal | None | MLP | 0.4678 | 9 | 0.4484 | 8 | 8.5 |

Table 8: Top-10 configurations for the ETTm1 dataset multivariate forecasting. IN: Instance Norm, SD: Series Decomposition. ✓ indicates module used, ✗ indicates not used. Red/blue highlights indicate best and second-best performances.

| Setup | IN | SD | Fusion | Embed | FF-Type | MSE | Rank | MAE | Rank | Total |
|---|---|---|---|---|---|---|---|---|---|---|
| | ✓ | ✓ | Temporal | Invert | RNN | 0.1766 | 4 | **0.2576** | **1** | **2.5** |
| | ✓ | ✓ | Feature | Patch | MLP | 0.1763 | 3 | 0.2581 | 3 | **3.0** |
| | ✓ | ✓ | Feature | Patch | RNN | **0.1761** | **2** | 0.2590 | 4 | **3.0** |
| | ✓ | ✗ | Feature | Patch | MLP | 0.1768 | 5 | **0.2577** | **2** | 3.5 |
| ETTm2_96_M | ✓ | ✗ | Feature | Patch | RNN | **0.1761** | **1** | 0.2600 | 9 | 5.0 |
| | ✓ | ✓ | Temporal | Patch | MLP | 0.1771 | 7 | 0.2598 | 7 | 7.0 |
| | ✓ | ✓ | Temporal | Freq | MLP | 0.1772 | 8 | 0.2596 | 6 | 7.0 |
| | ✓ | ✓ | Feature | Token | Trans. | 0.1770 | 6 | 0.2599 | 8 | 7.0 |
| | ✓ | ✗ | Temporal | Freq | MLP | 0.1795 | 16 | 0.2595 | 5 | 10.5 |
| | ✓ | ✗ | Feature | Token | Trans. | 0.1783 | 11 | 0.2607 | 13 | 12.0 |
| | ✓ | ✗ | Feature | Patch | MLP | **0.2403** | **1** | **0.2996** | **1** | **1.0** |
| | ✓ | ✓ | Feature | Patch | MLP | 0.2405 | 3 | **0.2997** | **2** | **2.5** |
| | ✓ | ✓ | Temporal | Patch | MLP | **0.2405** | **2** | 0.3008 | 4 | 3.0 |
| | ✓ | ✓ | Temporal | Invert | RNN | 0.2422 | 6 | 0.3008 | 5 | 5.5 |
| ETTm2_192_M | ✓ | ✓ | Temporal | Freq | MLP | 0.2406 | 4 | 0.3017 | 7 | 5.5 |
| | ✓ | ✓ | Temporal | Patch | RNN | 0.2427 | 8 | 0.3008 | 6 | 7.0 |
| | ✓ | ✓ | Feature | Patch | RNN | 0.2417 | 5 | 0.3024 | 9 | 7.0 |
| | ✓ | ✗ | Temporal | Patch | RNN | 0.2432 | 11 | 0.3007 | 3 | 7.0 |
| | ✓ | ✓ | Temporal | None | MLP | 0.2426 | 7 | 0.3023 | 8 | 7.5 |
| | ✓ | ✓ | Temporal | Invert | MLP | 0.2428 | 9 | 0.3031 | 10 | 9.5 |
| | ✓ | ✓ | Feature | Patch | MLP | **0.2980** | **1** | **0.3378** | **2** | **1.5** |
| | ✓ | ✗ | Temporal | Patch | RNN | 0.3009 | 5 | **0.3369** | **1** | **3.0** |
| | ✓ | ✗ | Feature | Patch | MLP | 0.3002 | 3 | 0.3389 | 3 | **3.0** |
| | ✓ | ✓ | Temporal | Freq | MLP | **0.2982** | **2** | 0.3393 | 5 | 3.5 |
| ETTm2_336_M | ✓ | ✓ | Temporal | Patch | RNN | 0.3017 | 8 | 0.3390 | 4 | 6.0 |
| | ✓ | ✓ | Temporal | None | MLP | 0.3003 | 4 | 0.3405 | 8 | 6.0 |
| | ✓ | ✓ | Feature | Patch | RNN | 0.3010 | 6 | 0.3398 | 6 | 6.0 |
| | ✓ | ✓ | Temporal | Patch | MLP | 0.3013 | 7 | 0.3398 | 7 | 7.0 |
| | ✓ | ✗ | Temporal | Freq | MLP | 0.3018 | 9 | 0.3405 | 9 | 9.0 |
| | ✓ | ✗ | Temporal | Patch | MLP | 0.3024 | 10 | 0.3409 | 10 | 10.0 |
| | ✓ | ✓ | Temporal | Patch | RNN | **0.3942** | **1** | **0.3924** | **1** | **1.0** |
| | ✓ | ✗ | Feature | Patch | MLP | 0.3981 | 3 | 0.3960 | 4 | **3.5** |
| | ✓ | ✓ | Temporal | Invert | MLP | **0.3965** | **2** | 0.3961 | 5 | **3.5** |
| | ✓ | ✓ | Feature | Patch | MLP | 0.3983 | 4 | 0.3958 | 3 | **3.5** |
| ETTm2_720_M | ✓ | ✗ | Temporal | Patch | RNN | 0.3991 | 8 | **0.3951** | **2** | 5.0 |
| | ✓ | ✓ | Temporal | Patch | MLP | 0.3984 | 5 | 0.3963 | 6 | 5.5 |
| | ✓ | ✗ | Temporal | Invert | MLP | 0.3990 | 6 | 0.3966 | 7 | 6.5 |
| | ✓ | ✓ | Temporal | None | MLP | 0.3991 | 7 | 0.3968 | 8 | 7.5 |
| | ✓ | ✗ | Temporal | Freq | MLP | 0.3992 | 9 | 0.3980 | 10 | 9.5 |
| | ✓ | ✗ | Temporal | None | MLP | 0.4006 | 10 | 0.3975 | 9 | 9.5 |

Table 9: Top-10 configurations for the ETTm2 dataset multivariate forecasting. IN: Instance Norm, SD: Series Decomposition. ✓ indicates module used, ✗ indicates not used. Red/blue highlights indicate best and second-best performances.

| Setup | IN | SD | Fusion | Embed | FF-Type | MSE | Rank | MAE | Rank | Total |
|---|---|---|---|---|---|---|---|---|---|---|
| | ✓ | ✓ | Feature | Patch | MLP | **0.0821** | **1** | **0.1974** | **1** | **1.0** |
| | ✓ | ✓ | Temporal | Patch | RNN | 0.0823 | 3 | **0.1978** | **2** | **2.5** |
| | ✓ | ✗ | Temporal | Patch | MLP | **0.0823** | **2** | 0.1988 | 4 | 3.0 |
| | ✓ | ✗ | Temporal | Patch | RNN | 0.0828 | 5 | 0.1987 | 3 | 4.0 |
| | ✓ | ✗ | Temporal | Invert | MLP | 0.0829 | 6 | 0.1994 | 6 | 6.0 |
| Exchange_96_M | ✓ | ✗ | Feature | Patch | MLP | 0.0833 | 7 | 0.1988 | 5 | 6.0 |
| | ✓ | ✗ | Temporal | Patch | Trans | 0.0827 | 4 | 0.2002 | 8 | 6.0 |
| | ✓ | ✓ | Temporal | Patch | MLP | 0.0836 | 9 | 0.2000 | 7 | 8.0 |
| | ✓ | ✓ | Temporal | Invert | MLP | 0.0839 | 10 | 0.2004 | 9 | 9.5 |
| | ✓ | ✗ | Temporal | None | MLP | 0.0845 | 11 | 0.2017 | 10 | 10.5 |
| | ✓ | ✗ | Temporal | Patch | MLP | **0.1724** | **1** | **0.2939** | **1** | **1.0** |
| | ✓ | ✗ | Temporal | None | MLP | **0.1734** | **2** | 0.2954 | 4 | **3.0** |
| | ✓ | ✗ | Feature | Patch | MLP | 0.1742 | 4 | **0.2945** | **2** | **3.0** |
| | ✓ | ✓ | Temporal | Patch | RNN | 0.1744 | 5 | 0.2948 | 3 | 4.0 |
| | ✓ | ✓ | Temporal | Invert | MLP | 0.1741 | 3 | 0.2954 | 5 | 4.0 |
| Exchange_192_M | ✓ | ✗ | Temporal | Patch | RNN | 0.1754 | 7 | 0.2958 | 6 | 6.5 |
| | ✓ | ✗ | Temporal | Invert | MLP | 0.1751 | 6 | 0.2965 | 8 | 7.0 |
| | ✓ | ✓ | Feature | Patch | MLP | 0.1757 | 9 | 0.2962 | 7 | 8.0 |
| | ✓ | ✓ | Feature | Patch | RNN | 0.1757 | 8 | 0.2972 | 9 | 8.5 |
| | ✓ | ✓ | Temporal | None | MLP | 0.1761 | 11 | 0.2979 | 11 | 11.0 |
| | ✓ | ✗ | Temporal | Patch | MLP | **0.3192** | **1** | **0.4075** | **1** | **1.0** |
| | ✓ | ✗ | Temporal | Invert | MLP | **0.3246** | **2** | **0.4121** | **2** | **2.0** |
| | ✓ | ✓ | Temporal | Patch | RNN | 0.3278 | 5 | 0.4127 | 3 | 4.0 |
| | ✓ | ✗ | Feature | Patch | RNN | 0.3278 | 3 | 0.4143 | 7 | 5.0 |
| | ✓ | ✓ | Feature | Patch | MLP | 0.3279 | 6 | 0.4134 | 4 | 5.0 |
| Exchange_336_M | ✓ | ✗ | Temporal | Patch | RNN | 0.3284 | 7 | 0.4135 | 5 | 6.0 |
| | ✓ | ✗ | Temporal | None | MLP | 0.3278 | 4 | 0.4148 | 8 | 6.0 |
| | ✓ | ✗ | Feature | Patch | MLP | 0.3296 | 8 | 0.4139 | 6 | 7.0 |
| | ✓ | ✓ | Feature | Invert | Trans. | 0.3312 | 10 | 0.4165 | 9 | 9.5 |
| | ✓ | ✓ | Feature | None | Trans. | 0.3307 | 9 | 0.4172 | 10 | 9.5 |
| | ✗ | ✗ | Temporal | Freq | MLP | **0.8183** | **1** | **0.6837** | **1** | **1.0** |
| | ✓ | ✓ | Temporal | None | MLP | 0.8337 | 3 | **0.6871** | **2** | **2.5** |
| | ✓ | ✗ | Temporal | Patch | MLP | 0.8378 | 4 | 0.6872 | 3 | 3.5 |
| | ✗ | ✓ | Temporal | Freq | MLP | **0.8205** | **2** | 0.6950 | 7 | 4.5 |
| | ✓ | ✗ | Temporal | Patch | Trans. | 0.8411 | 5 | 0.6895 | 4 | 4.5 |
| Exchange_720_M | ✓ | ✗ | Temporal | None | MLP | 0.8490 | 8 | 0.6955 | 9 | 8.5 |
| | ✓ | ✓ | Temporal | Patch | RNN | 0.8544 | 11 | 0.6948 | 6 | 8.5 |
| | ✓ | ✓ | Feature | Patch | MLP | 0.8545 | 12 | 0.6947 | 5 | 8.5 |
| | ✓ | ✗ | Feature | Patch | MLP | 0.8542 | 10 | 0.6954 | 8 | 9.0 |
| | ✓ | ✗ | Temporal | Invert | MLP | 0.8493 | 9 | 0.6958 | 10 | 9.5 |

Table 10: Top-10 configurations for the Exchange Rate dataset multivariate forecasting. IN: Instance Norm, SD: Series Decomposition. ✓ indicates module used, ✗ indicates not used. Red/blue highlights indicate best and second-best performances.

| Setup | IN | SD | Fusion | Embed | FF-Type | MSE | Rank | MAE | Rank | Total |
|---|---|---|---|---|---|---|---|---|---|---|
| | ✓ | ✗ | Feature | Patch | Trans. | **2.0338** | **2** | **0.8776** | **1** | **1.5** |
| | ✓ | ✓ | Feature | Patch | Trans. | **2.0034** | **1** | 0.9064 | 3 | **2.0** |
| | ✓ | ✗ | Temporal | Freq | MLP | 2.1586 | 3 | **0.8874** | **2** | 2.5 |
| | ✓ | ✗ | Temporal | Invert | MLP | 2.1982 | 4 | 0.9084 | 4 | 4.0 |
| ILI_24_M | ✓ | ✗ | Temporal | None | MLP | 2.2416 | 5 | 0.9231 | 5 | 5.0 |
| | ✓ | ✗ | Temporal | Token | MLP | 2.3280 | 9 | 0.9492 | 7 | 8.0 |
| | ✓ | ✓ | Temporal | Token | MLP | 2.3096 | 7 | 0.9640 | 11 | 9.0 |
| | ✓ | ✓ | Temporal | Freq | MLP | 2.4091 | 13 | 0.9563 | 9 | 11.0 |
| | ✓ | ✗ | Feature | Invert | RNN | 2.4471 | 17 | 0.9480 | 6 | 11.5 |
| | ✓ | ✗ | Temporal | Token | Trans. | 2.3071 | 6 | 1.0035 | 20 | 13.0 |
| | ✓ | ✗ | Temporal | Token | MLP | **1.9663** | **2** | **0.9067** | **1** | **1.5** |
| | ✓ | ✗ | Feature | Invert | RNN | **1.9444** | **1** | **0.9104** | **2** | **1.5** |
| | ✓ | ✓ | Feature | Invert | Trans. | 2.1458 | 3 | 0.9381 | 7 | **5.0** |
| | ✓ | ✗ | Temporal | Invert | MLP | 2.1907 | 6 | 0.9234 | 5 | 5.5 |
| ILI_36_M | ✓ | ✗ | Feature | Invert | Trans. | 2.1581 | 4 | 0.9414 | 9 | 6.5 |
| | ✓ | ✓ | Feature | Patch | Trans. | 2.2107 | 8 | 0.9257 | 6 | 7.0 |
| | ✓ | ✗ | Temporal | Freq | MLP | 2.2220 | 11 | 0.9190 | 4 | 7.5 |
| | ✓ | ✗ | Feature | Patch | Trans. | 2.2456 | 13 | 0.9117 | 3 | 8.0 |
| | ✓ | ✓ | Temporal | Token | MLP | 2.1651 | 5 | 0.9516 | 12 | 8.5 |
| | ✓ | ✓ | Temporal | Invert | MLP | 2.2197 | 10 | 0.9459 | 10 | 10.0 |
| | ✓ | ✗ | Feature | Patch | Trans. | **1.9918** | **2** | **0.8685** | **1** | **1.5** |
| | ✓ | ✗ | Temporal | Token | MLP | **1.8707** | **1** | 0.8860 | 3 | **2.0** |
| | ✓ | ✗ | Temporal | Freq | MLP | 2.0316 | 3 | **0.8844** | **2** | 2.5 |
| | ✓ | ✓ | Feature | Invert | Trans. | 2.0593 | 5 | 0.8933 | 4 | 4.5 |
| ILI_48_M | ✓ | ✓ | Temporal | Freq | MLP | 2.0393 | 4 | 0.9152 | 8 | 6.0 |
| | ✓ | ✗ | Feature | Invert | Trans. | 2.1405 | 7 | 0.9102 | 6 | 6.5 |
| | ✓ | ✓ | Feature | Patch | Trans. | 2.1567 | 8 | 0.9074 | 5 | 6.5 |
| | ✓ | ✗ | Temporal | Invert | MLP | 2.1383 | 6 | 0.9173 | 9 | 7.5 |
| | ✓ | ✗ | Feature | Invert | RNN | 2.1660 | 9 | 0.9137 | 7 | 8.0 |
| | ✓ | ✓ | Temporal | Token | MLP | 2.2079 | 12 | 0.9318 | 10 | 11.0 |
| | ✓ | ✗ | Feature | Patch | Trans. | **1.9860** | **1** | **0.8950** | **1** | **1.0** |
| | ✓ | ✓ | Feature | Invert | Trans. | **2.0064** | **2** | **0.8998** | **2** | **2.0** |
| | ✓ | ✗ | Temporal | Freq | MLP | 2.0119 | 3 | 0.9087 | 3 | 3.0 |
| | ✓ | ✗ | Feature | Invert | Trans. | 2.0342 | 4 | 0.9130 | 4 | 4.0 |
| ILI_60_M | ✓ | ✗ | Feature | Invert | RNN | 2.0992 | 6 | 0.9336 | 5 | 5.5 |
| | ✓ | ✓ | Temporal | Freq | MLP | 2.0898 | 5 | 0.9378 | 8 | 6.5 |
| | ✓ | ✗ | Temporal | Invert | MLP | 2.1317 | 8 | 0.9346 | 6 | 7.0 |
| | ✓ | ✓ | Feature | Patch | Trans. | 2.1360 | 9 | 0.9376 | 7 | 8.0 |
| | ✓ | ✓ | Feature | Invert | RNN | 2.1096 | 7 | 0.9380 | 9 | 8.0 |
| | ✓ | ✗ | Temporal | Patch | MLP | 2.1548 | 11 | 0.9435 | 10 | 10.5 |

Table 11: Top-10 configurations for the Illness (National Flu) dataset multivariate forecasting. IN: Instance Norm, SD: Series Decomposition. ✓ indicates module used, ✗ indicates not used. Red/blue highlights indicate best and second-best performances.

| Setup | IN | SD | Fusion | Embed | FF-Type | MSE | Rank | MAE | Rank | Total |
|---|---|---|---|---|---|---|---|---|---|---|
| ECL_96_M | ✓ | ✗ | Feature | Invert | Trans. | **0.1545** | **1** | **0.2455** | **1** | **1.0** |
|  | ✗ | ✗ | Feature | Invert | Trans. | **0.1554** | **2** | 0.2523 | 3 | **2.5** |
|  | ✓ | ✓ | Feature | Invert | Trans. | 0.1599 | 3 | **0.2508** | **2** | **2.5** |
|  | ✓ | ✓ | Feature | Invert | RNN | 0.1602 | 4 | 0.2550 | 4 | 4.0 |
|  | ✓ | ✗ | Feature | Invert | RNN | 0.1618 | 5 | 0.2562 | 6 | 5.5 |
|  | ✓ | ✓ | Feature | None | RNN | 0.1630 | 7 | 0.2562 | 7 | 7.0 |
|  | ✗ | ✓ | Feature | Invert | Trans. | 0.1624 | 6 | 0.2578 | 9 | 7.5 |
|  | ✓ | ✓ | Feature | None | Trans. | 0.1682 | 12 | 0.2560 | 5 | 8.5 |
|  | ✓ | ✗ | Feature | None | RNN | 0.1651 | 10 | 0.2581 | 10 | 10.0 |
|  | ✗ | ✗ | Feature | Invert | RNN | 0.1634 | 8 | 0.2628 | 16 | 12.0 |
| ECL_192_M | ✓ | ✗ | Feature | Invert | Trans. | **0.1670** | **1** | **0.2564** | **1** | **1.0** |
|  | ✗ | ✗ | Feature | Invert | Trans. | **0.1688** | **2** | 0.2640 | 3 | **2.5** |
|  | ✓ | ✓ | Feature | Invert | Trans. | 0.1707 | 3 | **0.2602** | **2** | **2.5** |
|  | ✗ | ✓ | Feature | Invert | Trans. | 0.1721 | 4 | 0.2663 | 5 | 4.5 |
|  | ✓ | ✓ | Feature | Invert | RNN | 0.1756 | 5 | 0.2682 | 10 | 7.5 |
|  | ✓ | ✗ | Feature | Patch | Trans. | 0.1802 | 12 | 0.2657 | 4 | 8.0 |
|  | ✗ | ✗ | Feature | Patch | Trans. | 0.1784 | 8 | 0.2676 | 9 | 8.5 |
|  | ✓ | ✓ | Feature | None | RNN | 0.1771 | 6 | 0.2683 | 11 | 8.5 |
|  | ✓ | ✓ | Feature | None | Trans. | 0.1797 | 11 | 0.2669 | 8 | 9.5 |
|  | ✓ | ✗ | Feature | Invert | RNN | 0.1772 | 7 | 0.2687 | 13 | 10.0 |
| ECL_336_M | ✓ | ✗ | Feature | Invert | Trans. | **0.1827** | **1** | **0.2732** | **1** | **1.0** |
|  | ✓ | ✓ | Feature | Invert | Trans. | 0.1864 | 3 | **0.2769** | **2** | **2.5** |
|  | ✗ | ✗ | Feature | Invert | Trans. | **0.1852** | **2** | 0.2830 | 6 | 4.0 |
|  | ✗ | ✗ | Feature | Patch | Trans. | 0.1924 | 4 | 0.2835 | 7 | 5.5 |
|  | ✓ | ✗ | Feature | Patch | Trans. | 0.1965 | 11 | 0.2823 | 3 | 7.0 |
|  | ✗ | ✗ | Feature | Patch | RNN | 0.1942 | 7 | 0.2850 | 8 | 7.5 |
|  | ✗ | ✓ | Feature | Patch | RNN | 0.1947 | 8 | 0.2850 | 9 | 8.5 |
|  | ✓ | ✓ | Feature | Invert | RNN | 0.1937 | 6 | 0.2856 | 11 | 8.5 |
|  | ✗ | ✓ | Feature | Invert | Trans. | 0.1928 | 5 | 0.2873 | 15 | 10.0 |
|  | ✓ | ✗ | Feature | Patch | RNN | 0.1978 | 15 | 0.2825 | 5 | 10.0 |
| ECL_720_M | ✓ | ✗ | Feature | Invert | Trans. | **0.2201** | **2** | **0.3061** | **1** | **1.5** |
|  | ✓ | ✓ | Feature | Invert | Trans. | 0.2221 | 3 | **0.3082** | **2** | **2.5** |
|  | ✗ | ✗ | Feature | Invert | Trans. | **0.2177** | **1** | 0.3153 | 5 | 3.0 |
|  | ✗ | ✗ | Feature | Patch | Trans. | 0.2275 | 5 | 0.3152 | 3 | 4.0 |
|  | ✗ | ✗ | Feature | Patch | RNN | 0.2299 | 6 | 0.3166 | 8 | 7.0 |
|  | ✗ | ✓ | Feature | Invert | Trans. | 0.2234 | 4 | 0.3176 | 11 | 7.5 |
|  | ✗ | ✓ | Feature | Patch | RNN | 0.2308 | 7 | 0.3167 | 9 | 8.0 |
|  | ✓ | ✗ | Feature | Patch | Trans. | 0.2364 | 14 | 0.3152 | 4 | 9.0 |
|  | ✓ | ✗ | Feature | Patch | RNN | 0.2385 | 17 | 0.3157 | 6 | 11.5 |
|  | ✓ | ✓ | Feature | None | Trans. | 0.2358 | 13 | 0.3168 | 10 | 11.5 |

Table 12: Top-10 configurations for the Electricity (ECL) dataset multivariate forecasting. IN: Instance Norm, SD: Series Decomposition. ✓ indicates module used, ✗ indicates not used. Red/blue highlights indicate best and second-best performances.

| Setup | IN | SD | Fusion | Embed | FF-Type | MSE | Rank | MAE | Rank | Total |
|---|---|---|---|---|---|---|---|---|---|---|
| | ✓ | ✓ | Feature | Token | RNN | **0.1588** | **1** | **0.2062** | **1** | **1.0** |
| | ✓ | ✓ | Temporal | Invert | RNN | **0.1589** | **2** | **0.2068** | **2** | **2.0** |
| | ✓ | ✗ | Feature | Token | RNN | 0.1614 | 3 | 0.2092 | 3 | 3.0 |
| | ✓ | ✗ | Temporal | Invert | RNN | 0.1617 | 4 | 0.2100 | 6 | 5.0 |
| Weather_96_M | ✓ | ✓ | Feature | Invert | RNN | 0.1634 | 7 | 0.2097 | 4 | 5.5 |
| | ✓ | ✓ | Feature | None | RNN | 0.1634 | 6 | 0.2097 | 5 | 5.5 |
| | ✓ | ✓ | Temporal | Token | MLP | 0.1624 | 5 | 0.2117 | 9 | 7.0 |
| | ✓ | ✗ | Feature | None | RNN | 0.1642 | 9 | 0.2104 | 7 | 8.0 |
| | ✓ | ✗ | Feature | Invert | RNN | 0.1649 | 13 | 0.2108 | 8 | 10.5 |
| | ✓ | ✗ | Temporal | Token | MLP | 0.1648 | 11 | 0.2131 | 11 | 11.0 |
| | ✓ | ✓ | Feature | Token | RNN | 0.2096 | 3 | **0.2528** | **1** | **2.0** |
| | ✓ | ✗ | Feature | Token | RNN | 0.2102 | 5 | **0.2531** | **2** | **3.5** |
| | ✓ | ✓ | Temporal | Invert | RNN | 0.2100 | 4 | 0.2542 | 6 | 5.0 |
| | ✓ | ✗ | Feature | None | RNN | 0.2119 | 10 | 0.2540 | 4 | 7.0 |
| Weather_192_M | ✓ | ✗ | Feature | Invert | RNN | 0.2126 | 13 | 0.2539 | 3 | 8.0 |
| | ✓ | ✓ | Feature | Invert | RNN | 0.2127 | 14 | 0.2541 | 5 | 9.5 |
| | ✓ | ✓ | Feature | None | RNN | 0.2125 | 12 | 0.2545 | 7 | 9.5 |
| | ✓ | ✗ | Temporal | Invert | RNN | 0.2127 | 15 | 0.2558 | 9 | 12.0 |
| | ✗ | ✗ | Feature | Patch | RNN | **0.2082** | **2** | 0.2656 | 38 | 20.0 |
| | ✗ | ✗ | Temporal | Freq | MLP | **0.2058** | **1** | 0.2660 | 41 | 21.0 |
| | ✓ | ✓ | Temporal | Invert | RNN | 0.2671 | 14 | **0.2954** | **1** | **7.5** |
| | ✓ | ✓ | Feature | Invert | RNN | 0.2700 | 18 | 0.2964 | 4 | **11.0** |
| | ✓ | ✗ | Feature | Invert | RNN | 0.2704 | 20 | 0.2960 | 3 | 11.5 |
| | ✓ | ✓ | Feature | None | RNN | 0.2705 | 22 | 0.2966 | 8 | 15.0 |
| Weather_336_M | ✗ | ✗ | Temporal | Freq | MLP | **0.2566** | **1** | 0.3054 | 34 | 17.5 |
| | ✓ | ✗ | Feature | None | RNN | 0.2706 | 23 | 0.2973 | 14 | 18.5 |
| | ✓ | ✓ | Feature | Token | RNN | 0.2719 | 26 | 0.2970 | 11 | 18.5 |
| | ✗ | ✓ | Temporal | None | MLP | 0.2628 | 3 | 0.3078 | 39 | 21.0 |
| | ✓ | ✗ | Feature | Patch | RNN | 0.2761 | 37 | 0.2965 | 7 | 22.0 |
| | ✗ | ✗ | Temporal | Patch | RNN | 0.2631 | 5 | 0.3079 | 40 | 22.5 |
| | ✓ | ✓ | Temporal | Invert | RNN | 0.3429 | 19 | **0.3465** | **1** | **10.0** |
| | ✓ | ✗ | Feature | Patch | RNN | 0.3536 | 38 | 0.3469 | 5 | **21.5** |
| | ✓ | ✗ | Temporal | Patch | MLP | 0.3534 | 37 | 0.3471 | 7 | 22.0 |
| | ✓ | ✗ | Temporal | Freq | MLP | 0.3548 | 41 | 0.3468 | 4 | 22.5 |
| Weather_720_M | ✓ | ✓ | Temporal | Token | MLP | 0.3481 | 24 | 0.3499 | 22 | 23.0 |
| | ✓ | ✓ | Temporal | None | MLP | 0.3552 | 44 | **0.3466** | **2** | 23.0 |
| | ✓ | ✓ | Feature | Invert | RNN | 0.3505 | 31 | 0.3485 | 15 | 23.0 |
| | ✓ | ✓ | Temporal | Freq | MLP | 0.3549 | 42 | 0.3470 | 6 | 24.0 |
| | ✗ | ✗ | Feature | Patch | MLP | 0.3355 | 10 | 0.3588 | 42 | 26.0 |
| | ✓ | ✗ | Feature | Invert | RNN | 0.3517 | 32 | 0.3492 | 20 | 26.0 |

Table 13: Top-10 configurations for the Weather dataset multivariate forecasting. IN: Instance Norm, SD: Series Decomposition. ✓ indicates module used, ✗ indicates not used. Red/blue highlights indicate best and second-best performances.

| Setup | IN | SD | Fusion | Embed | FF-Type | MSE | Rank | MAE | Rank | Total |
|---|---|---|---|---|---|---|---|---|---|---|
| | ✗ | ✓ | Feature | Invert | RNN | **0.0725** | **2** | **0.1777** | **1** | **1.5** |
| | ✓ | ✓ | Feature | Invert | Trans. | **0.0714** | **1** | 0.1781 | 3 | **2.0** |
| | ✗ | ✗ | Feature | Invert | RNN | 0.0726 | 3 | **0.1780** | **2** | 2.5 |
| | ✗ | ✗ | Feature | None | RNN | 0.0726 | 4 | 0.1782 | 4 | 4.0 |
| PEMS03_12_M | ✗ | ✓ | Feature | None | RNN | 0.0734 | 5 | 0.1793 | 5 | 5.0 |
| | ✓ | ✗ | Feature | Invert | Trans. | 0.0739 | 7 | 0.1812 | 6 | 6.5 |
| | ✓ | ✓ | Feature | None | Trans. | 0.0739 | 6 | 0.1820 | 8 | 7.0 |
| | ✗ | ✗ | Feature | Invert | Trans. | 0.0744 | 8 | 0.1814 | 7 | 7.5 |
| | ✗ | ✓ | Temporal | Freq | MLP | 0.0750 | 10 | 0.1830 | 10 | 10.0 |
| | ✓ | ✓ | Temporal | Invert | RNN | 0.0751 | 11 | 0.1823 | 9 | 10.0 |
| | ✗ | ✓ | Feature | Invert | RNN | **0.0985** | **1** | **0.2090** | **1** | **1.0** |
| | ✓ | ✓ | Temporal | Invert | RNN | **0.0990** | **2** | **0.2091** | **2** | **2.0** |
| | ✗ | ✗ | Feature | Invert | RNN | 0.1003 | 3 | 0.2108 | 3 | 3.0 |
| | ✗ | ✗ | Feature | None | RNN | 0.1006 | 4 | 0.2125 | 5 | 4.5 |
| PEMS03_24_M | ✗ | ✓ | Feature | None | RNN | 0.1009 | 5 | 0.2117 | 4 | 4.5 |
| | ✓ | ✓ | Feature | Token | RNN | 0.1023 | 6 | 0.2136 | 6 | 6.0 |
| | ✓ | ✗ | Temporal | Invert | RNN | 0.1045 | 8 | 0.2140 | 7 | 7.5 |
| | ✓ | ✓ | Feature | Invert | Trans. | 0.1030 | 7 | 0.2156 | 8 | 7.5 |
| | ✗ | ✓ | Feature | Invert | Trans. | 0.1051 | 10 | 0.2156 | 9 | 9.5 |
| | ✓ | ✓ | Temporal | Token | MLP | 0.1045 | 9 | 0.2176 | 11 | 10.0 |
| | ✗ | ✓ | Feature | Invert | RNN | **0.1220** | **1** | **0.2330** | **2** | **1.5** |
| | ✗ | ✓ | Temporal | Invert | RNN | 0.1251 | 3 | **0.2315** | **1** | **2.0** |
| | ✗ | ✗ | Feature | Invert | RNN | **0.1230** | **2** | 0.2348 | 4 | 3.0 |
| | ✗ | ✓ | Feature | Token | RNN | 0.1254 | 4 | 0.2341 | 3 | 3.5 |
| PEMS03_36_M | ✓ | ✓ | Temporal | Invert | RNN | 0.1259 | 5 | 0.2371 | 5 | 5.0 |
| | ✗ | ✗ | Feature | None | RNN | 0.1265 | 6 | 0.2386 | 6 | 6.0 |
| | ✗ | ✓ | Feature | None | RNN | 0.1280 | 7 | 0.2393 | 7 | 7.0 |
| | ✓ | ✓ | Feature | Token | RNN | 0.1294 | 8 | 0.2401 | 9 | 8.5 |
| | ✗ | ✗ | Feature | Token | RNN | 0.1314 | 10 | 0.2395 | 8 | 9.0 |
| | ✗ | ✓ | Feature | Token | Trans. | 0.1310 | 9 | 0.2410 | 11 | 10.0 |
| | ✗ | ✓ | Feature | Token | RNN | **0.1398** | **1** | **0.2489** | **1** | **1.0** |
| | ✗ | ✓ | Temporal | Invert | RNN | 0.1426 | 3 | 0.2499 | 3 | **3.0** |
| | ✗ | ✗ | Temporal | Token | Trans. | 0.1438 | 4 | 0.2530 | 5 | 4.5 |
| | ✗ | ✓ | Temporal | Token | RNN | **0.1412** | **2** | 0.2543 | 7 | 4.5 |
| PEMS03_48_M | ✗ | ✗ | Temporal | Invert | RNN | 0.1461 | 9 | **0.2498** | **2** | 5.5 |
| | ✗ | ✗ | Feature | Invert | RNN | 0.1445 | 5 | 0.2539 | 6 | 5.5 |
| | ✗ | ✗ | Feature | Token | RNN | 0.1455 | 8 | 0.2525 | 4 | 6.0 |
| | ✗ | ✓ | Feature | Invert | RNN | 0.1449 | 6 | 0.2552 | 8 | 7.0 |
| | ✗ | ✓ | Temporal | Token | Trans. | 0.1450 | 7 | 0.2580 | 9 | 8.0 |
| | ✗ | ✓ | Feature | Token | Trans. | 0.1499 | 11 | 0.2591 | 10 | 10.5 |

Table 14: Top-10 configurations for the PEMS03 dataset. IN: Instance Norm, SD: Series Decomposition. ✓ indicates module used, ✗ indicates not used. Red/blue highlights indicate best and second-best performances.

| Setup | IN | SD | Fusion | Embed | FF-Type | MSE | Rank | MAE | Rank | Total |
|---|---|---|---|---|---|---|---|---|---|---|
| | ✗ | ✓ | Feature | Invert | Trans. | **0.0814** | **1** | **0.1891** | **1** | **1.0** |
| | ✗ | ✓ | Feature | Invert | RNN | **0.0815** | **2** | **0.1894** | **2** | **2.0** |
| | ✗ | ✗ | Feature | Invert | RNN | 0.0816 | 3 | 0.1899 | 3 | 3.0 |
| | ✗ | ✗ | Feature | Invert | Trans. | 0.0825 | 4 | 0.1910 | 4 | 4.0 |
| | ✗ | ✗ | Feature | None | RNN | 0.0826 | 5 | 0.1911 | 5 | 5.0 |
| PEMS04_12_M | ✗ | ✓ | Feature | None | RNN | 0.0829 | 6 | 0.1913 | 7 | 6.5 |
| | ✓ | ✓ | Temporal | Invert | RNN | 0.0830 | 7 | 0.1912 | 6 | 6.5 |
| | ✓ | ✓ | Feature | Token | RNN | 0.0856 | 9 | 0.1953 | 8 | 8.5 |
| | ✓ | ✗ | Temporal | Invert | RNN | 0.0856 | 10 | 0.1955 | 9 | 9.5 |
| | ✓ | ✓ | Temporal | Token | MLP | 0.0848 | 8 | 0.1958 | 11 | 9.5 |
| | ✓ | ✓ | Temporal | Invert | RNN | **0.0992** | **1** | **0.2119** | **1** | **1.0** |
| | ✓ | ✗ | Temporal | Invert | RNN | **0.1013** | **2** | **0.2146** | **2** | **2.0** |
| | ✓ | ✓ | Temporal | Token | RNN | 0.1013 | 3 | 0.2146 | 3 | 3.0 |
| | ✓ | ✗ | Temporal | Token | Trans. | 0.1026 | 4 | 0.2151 | 6 | 5.0 |
| | ✗ | ✓ | Feature | Invert | RNN | 0.1029 | 5 | 0.2168 | 10 | 7.5 |
| PEMS04_24_M | ✓ | ✓ | Temporal | Token | Trans. | 0.1043 | 8 | 0.2163 | 8 | 8.0 |
| | ✗ | ✗ | Feature | Invert | RNN | 0.1032 | 6 | 0.2172 | 11 | 8.5 |
| | ✗ | ✓ | Temporal | Invert | RNN | 0.1057 | 12 | 0.2155 | 7 | 9.5 |
| | ✓ | ✓ | Feature | Token | RNN | 0.1034 | 7 | 0.2185 | 13 | 10.0 |
| | ✗ | ✓ | Feature | Invert | Trans. | 0.1048 | 10 | 0.2187 | 14 | 12.0 |
| | ✗ | ✓ | Temporal | Invert | RNN | **0.1108** | **1** | **0.2203** | **1** | **1.0** |
| | ✗ | ✓ | Temporal | Token | RNN | **0.1134** | **2** | **0.2208** | **2** | **2.0** |
| | ✗ | ✗ | Temporal | Token | Trans. | 0.1149 | 4 | 0.2228 | 3 | 3.5 |
| | ✗ | ✓ | Temporal | Token | Trans. | 0.1147 | 3 | 0.2230 | 4 | 3.5 |
| | ✗ | ✗ | Temporal | Invert | RNN | 0.1155 | 6 | 0.2240 | 5 | 5.5 |
| PEMS04_36_M | ✓ | ✓ | Temporal | Invert | RNN | 0.1151 | 5 | 0.2294 | 8 | 6.5 |
| | ✗ | ✓ | Feature | Token | RNN | 0.1157 | 7 | 0.2268 | 7 | 7.0 |
| | ✗ | ✗ | Temporal | Token | RNN | 0.1184 | 10 | 0.2268 | 6 | 8.0 |
| | ✓ | ✗ | Temporal | Token | Trans. | 0.1171 | 9 | 0.2318 | 12 | 10.5 |
| | ✓ | ✓ | Temporal | Token | RNN | 0.1165 | 8 | 0.2323 | 14 | 11.0 |
| | ✗ | ✓ | Temporal | Invert | RNN | **0.1149** | **1** | **0.2246** | **1** | **1.0** |
| | ✗ | ✗ | Temporal | Token | Trans. | 0.1168 | 3 | **0.2258** | **2** | **2.5** |
| | ✗ | ✓ | Temporal | Token | Trans. | **0.1164** | **2** | 0.2261 | 3 | **2.5** |
| | ✗ | ✗ | Temporal | Invert | RNN | 0.1174 | 4 | 0.2267 | 4 | 4.0 |
| | ✗ | ✗ | Temporal | Token | RNN | 0.1195 | 6 | 0.2284 | 5 | 5.5 |
| PEMS04_48_M | ✗ | ✓ | Temporal | Token | RNN | 0.1192 | 5 | 0.2286 | 6 | 5.5 |
| | ✗ | ✓ | Feature | Token | RNN | 0.1202 | 7 | 0.2312 | 7 | 7.0 |
| | ✗ | ✗ | Feature | Invert | MLP | 0.1213 | 8 | 0.2322 | 8 | 8.0 |
| | ✗ | ✓ | Temporal | None | RNN | 0.1236 | 9 | 0.2355 | 10 | 9.5 |
| | ✗ | ✗ | Temporal | None | RNN | 0.1242 | 11 | 0.2334 | 9 | 10.0 |

Table 15: Top-10 configurations for the PEMS04 dataset. IN: Instance Norm, SD: Series Decomposition. ✓ indicates module used, ✗ indicates not used. Red/blue highlights indicate best and second-best performances.

| Setup | IN | SD | Fusion | Embed | FF-Type | MSE | Rank | MAE | Rank | Total |
|---|---|---|---|---|---|---|---|---|---|---|
| | ✗ | ✓ | Feature | Invert | Trans. | **0.0663** | **1** | **0.1650** | **1** | **1.0** |
| | ✗ | ✓ | Feature | Invert | RNN | **0.0664** | **2** | 0.1682 | 3 | **2.5** |
| | ✓ | ✓ | Feature | Invert | Trans. | 0.0669 | 3 | **0.1676** | **2** | **2.5** |
| | ✗ | ✗ | Feature | Invert | RNN | 0.0673 | 4 | 0.1696 | 5 | 4.5 |
| PEMS07_12_M | ✗ | ✗ | Feature | Invert | Trans. | 0.0682 | 7 | 0.1691 | 4 | 5.5 |
| | ✗ | ✗ | Feature | None | RNN | 0.0676 | 5 | 0.1698 | 6 | 5.5 |
| | ✗ | ✓ | Feature | None | RNN | 0.0677 | 6 | 0.1699 | 7 | 6.5 |
| | ✓ | ✗ | Feature | Invert | Trans. | 0.0699 | 9 | 0.1712 | 8 | 8.5 |
| | ✓ | ✓ | Feature | None | Trans. | 0.0697 | 8 | 0.1717 | 9 | 8.5 |
| | ✓ | ✓ | Feature | Invert | RNN | 0.0703 | 10 | 0.1737 | 10 | 10.0 |
| | ✗ | ✓ | Feature | Invert | Trans. | **0.0901** | **1** | **0.1946** | **1** | **1.0** |
| | ✗ | ✗ | Feature | Invert | Trans. | **0.0912** | **2** | **0.1959** | **2** | **2.0** |
| | ✗ | ✓ | Feature | Invert | RNN | 0.0938 | 3 | 0.2009 | 3 | 3.0 |
| | ✗ | ✗ | Feature | Invert | RNN | 0.0954 | 5 | 0.2027 | 5 | 5.0 |
| PEMS07_24_M | ✗ | ✓ | Feature | None | RNN | 0.0953 | 4 | 0.2031 | 6 | 5.0 |
| | ✓ | ✓ | Temporal | Invert | RNN | 0.0999 | 8 | 0.2013 | 4 | 6.0 |
| | ✗ | ✗ | Feature | None | RNN | 0.0962 | 6 | 0.2043 | 8 | 7.0 |
| | ✓ | ✗ | Temporal | Invert | RNN | 0.1017 | 9 | 0.2037 | 7 | 8.0 |
| | ✓ | ✓ | Feature | Invert | Trans. | 0.0992 | 7 | 0.2053 | 9 | 8.0 |
| | ✓ | ✓ | Feature | Token | RNN | 0.1022 | 10 | 0.2057 | 10 | 10.0 |
| | ✗ | ✓ | Feature | Invert | Trans. | **0.1086** | **1** | **0.2159** | **1** | **1.0** |
| | ✗ | ✗ | Feature | Invert | Trans. | **0.1118** | **2** | 0.2197 | 3 | **2.5** |
| | ✓ | ✓ | Temporal | Invert | RNN | 0.1182 | 4 | **0.2195** | **2** | 3.0 |
| | ✓ | ✗ | Temporal | Invert | RNN | 0.1206 | 6 | 0.2218 | 4 | 5.0 |
| PEMS07_36_M | ✗ | ✓ | Feature | Invert | RNN | 0.1175 | 3 | 0.2258 | 8 | 5.5 |
| | ✓ | ✓ | Feature | Token | RNN | 0.1206 | 7 | 0.2234 | 5 | 6.0 |
| | ✓ | ✓ | Temporal | Token | RNN | 0.1208 | 8 | 0.2241 | 6 | 7.0 |
| | ✗ | ✗ | Feature | Invert | RNN | 0.1200 | 5 | 0.2282 | 12 | 8.5 |
| | ✓ | ✗ | Temporal | Token | RNN | 0.1218 | 11 | 0.2249 | 7 | 9.0 |
| | ✓ | ✓ | Feature | Invert | MLP | 0.1216 | 10 | 0.2259 | 9 | 9.5 |
| | ✗ | ✓ | Feature | Invert | Trans. | **0.1237** | **1** | **0.2319** | **1** | **1.0** |
| | ✗ | ✗ | Feature | Invert | Trans. | **0.1283** | **2** | **0.2367** | **2** | **2.0** |
| | ✓ | ✓ | Temporal | Invert | RNN | 0.1358 | 3 | 0.2378 | 3 | 3.0 |
| | ✓ | ✗ | Temporal | Token | RNN | 0.1368 | 4 | 0.2396 | 4 | 4.0 |
| PEMS07_48_M | ✓ | ✗ | Temporal | Invert | RNN | 0.1377 | 8 | 0.2397 | 5 | 6.5 |
| | ✓ | ✓ | Temporal | Token | RNN | 0.1373 | 6 | 0.2416 | 7 | 6.5 |
| | ✓ | ✗ | Feature | Token | RNN | 0.1377 | 7 | 0.2399 | 6 | 6.5 |
| | ✗ | ✓ | Feature | Invert | RNN | 0.1368 | 5 | 0.2457 | 10 | 7.5 |
| | ✗ | ✗ | Feature | Invert | RNN | 0.1393 | 9 | 0.2472 | 14 | 11.5 |
| | ✓ | ✓ | Feature | Invert | MLP | 0.1404 | 10 | 0.2472 | 13 | 11.5 |

Table 16: Top-10 configurations for the PEMS07 dataset. IN: Instance Norm, SD: Series Decomposition. ✓ indicates module used, ✗ indicates not used. Red/blue highlights indicate best and second-best performances.

| Setup | IN | SD | Fusion | Embed | FF-Type | MSE | Rank | MAE | Rank | Total |
|---|---|---|---|---|---|---|---|---|---|---|
| | ✓ | ✓ | Feature | Invert | Trans. | **0.0795** | **1** | **0.1819** | **1** | **1.0** |
| | ✓ | ✗ | Feature | Invert | Trans. | 0.0834 | 3 | **0.1868** | **2** | **2.5** |
| | ✓ | ✓ | Feature | None | Trans. | **0.0833** | **2** | 0.1877 | 3 | **2.5** |
| | ✓ | ✓ | Feature | Invert | RNN | 0.0844 | 4 | 0.1895 | 6 | 5.0 |
| PEMS08_12_M | ✓ | ✗ | Feature | Invert | RNN | 0.0856 | 5 | 0.1908 | 8 | 6.5 |
| | ✓ | ✗ | Feature | None | Trans. | 0.0866 | 7 | 0.1913 | 10 | 8.5 |
| | ✓ | ✓ | Feature | None | RNN | 0.0864 | 6 | 0.1918 | 11 | 8.5 |
| | ✓ | ✗ | Feature | None | RNN | 0.0867 | 8 | 0.1920 | 12 | 10.0 |
| | ✗ | ✓ | Temporal | Freq | MLP | 0.0870 | 9 | 0.1929 | 13 | 11.0 |
| | ✗ | ✗ | Temporal | Freq | MLP | 0.0891 | 10 | 0.1955 | 14 | 12.0 |
| | ✓ | ✓ | Feature | Invert | Trans. | **0.1162** | **1** | **0.2191** | **1** | **1.0** |
| | ✓ | ✗ | Feature | Invert | Trans. | **0.1226** | **2** | 0.2263 | 6 | **4.0** |
| | ✓ | ✓ | Feature | None | Trans. | 0.1267 | 3 | 0.2327 | 8 | 5.5 |
| | ✓ | ✓ | Temporal | Invert | RNN | 0.1318 | 6 | 0.2307 | 7 | 6.5 |
| PEMS08_24_M | ✓ | ✓ | Feature | Invert | RNN | 0.1296 | 4 | 0.2359 | 9 | 6.5 |
| | ✓ | ✗ | Feature | Invert | RNN | 0.1302 | 5 | 0.2361 | 10 | 7.5 |
| | ✗ | ✓ | Temporal | Freq | MLP | 0.1319 | 7 | 0.2382 | 12 | 9.5 |
| | ✓ | ✗ | Feature | None | RNN | 0.1325 | 9 | 0.2384 | 13 | 11.0 |
| | ✓ | ✓ | Feature | None | RNN | 0.1325 | 8 | 0.2386 | 14 | 11.0 |
| | ✓ | ✓ | Feature | Token | RNN | 0.1378 | 14 | 0.2376 | 11 | 12.5 |
| | ✓ | ✓ | Feature | Invert | Trans. | **0.1581** | **1** | 0.2555 | 5 | **3.0** |
| | ✓ | ✓ | Temporal | Invert | RNN | **0.1685** | **2** | **0.2638** | **7** | **4.5** |
| | ✓ | ✗ | Feature | Invert | Trans. | 0.1689 | 3 | 0.2679 | 9 | 6.0 |
| | ✓ | ✓ | Feature | Token | RNN | 0.1722 | 5 | 0.2681 | 10 | 7.5 |
| PEMS08_36_M | ✓ | ✓ | Feature | None | Trans. | 0.1711 | 4 | 0.2720 | 12 | 8.0 |
| | ✓ | ✗ | Temporal | Invert | RNN | 0.1736 | 6 | 0.2710 | 11 | 8.5 |
| | ✓ | ✗ | Feature | Token | RNN | 0.1745 | 7 | 0.2733 | 14 | 10.5 |
| | ✓ | ✗ | Temporal | Token | Trans. | 0.1801 | 12 | 0.2725 | 13 | 12.5 |
| | ✗ | ✓ | Temporal | Freq | MLP | 0.1768 | 8 | 0.2778 | 19 | 13.5 |
| | ✓ | ✗ | Feature | Token | Trans. | 0.1769 | 9 | 0.2774 | 18 | 13.5 |
| | ✗ | ✓ | Feature | Invert | RNN | 0.2135 | 5 | **0.2645** | **1** | **3.0** |
| | ✓ | ✓ | Feature | Invert | Trans. | **0.2013** | **1** | 0.2892 | 7 | **4.0** |
| | ✓ | ✓ | Temporal | Invert | RNN | **0.2052** | **2** | 0.2947 | 10 | 6.0 |
| | ✗ | ✗ | Feature | Invert | RNN | 0.2170 | 12 | **0.2673** | **2** | 7.0 |
| PEMS08_48_M | ✓ | ✗ | Temporal | Token | Trans. | 0.2112 | 4 | 0.2975 | 12 | 8.0 |
| | ✓ | ✓ | Feature | Token | RNN | 0.2101 | 3 | 0.2986 | 13 | 8.0 |
| | ✗ | ✓ | Feature | None | RNN | 0.2191 | 16 | 0.2714 | 3 | 9.5 |
| | ✗ | ✗ | Feature | None | RNN | 0.2267 | 22 | 0.2788 | 5 | 13.5 |
| | ✓ | ✗ | Feature | Token | RNN | 0.2161 | 9 | 0.3063 | 21 | 15.0 |
| | ✓ | ✗ | Feature | Invert | Trans. | 0.2170 | 11 | 0.3049 | 19 | 15.0 |

Table 17: Top-10 configurations for the PEMS08 dataset. IN: Instance Norm, SD: Series Decomposition. ✓ indicates module used, ✗ indicates not used. Red/blue highlights indicate best and second-best performances.

| Setup | IN | SD | Fusion | Embed | FF-Type | SMAPE | Rank | MASE | Rank | OWA | Rank | Total |
|---|---|---|---|---|---|---|---|---|---|---|---|---|
| M4_Hourly_S | ✗ | ✗ | Feature | Invert | Trans. | 18.0080 | 7 | **2.6195** | **3** | **1.0367** | **2** | **4.00** |
| | ✗ | ✓ | Temporal | Invert | MLP | 17.8405 | 3 | 2.9283 | 5 | 1.0968 | 4 | **4.00** |
| | ✗ | ✗ | Feature | Patch | Trans. | 18.2368 | 12 | **2.4683** | **1** | **1.0115** | **1** | 4.67 |
| | ✗ | ✗ | Temporal | Patch | MLP | **17.6707** | **1** | 3.0920 | 8 | 1.1260 | 7 | 5.33 |
| | ✗ | ✗ | Temporal | Invert | MLP | 17.9402 | 4 | 3.0478 | 7 | 1.1240 | 6 | 5.67 |
| | ✗ | ✓ | Temporal | Patch | MLP | **17.8320** | **2** | 3.1867 | 9 | 1.1500 | 8 | 6.33 |
| | ✗ | ✗ | Temporal | Patch | Trans. | 18.7640 | 20 | 2.5265 | 2 | 1.0377 | 3 | 8.33 |
| | ✗ | ✓ | Feature | Patch | RNN | 18.0900 | 8 | 3.2920 | 12 | 1.1793 | 10 | 10.00 |
| | ✗ | ✓ | Feature | Invert | Trans. | 19.0040 | 22 | 2.8540 | 4 | 1.1128 | 5 | 10.33 |
| | ✗ | ✓ | Temporal | Patch | RNN | 17.9512 | 5 | 3.3385 | 15 | 1.1850 | 12 | 10.67 |
| M4_Daily_S | ✓ | ✓ | Temporal | None | MLP | **2.9520** | **1** | **3.1000** | **1** | **0.9575** | **1** | **1.00** |
| | ✓ | ✓ | Temporal | Freq | MLP | **2.9578** | **2** | **3.1227** | **2** | **0.9620** | **2** | **2.00** |
| | ✗ | ✓ | Feature | Patch | RNN | 3.0127 | 4 | 3.1990 | 3 | 0.9825 | 3 | 3.33 |
| | ✗ | ✗ | Feature | Patch | RNN | 3.0122 | 3 | 3.2135 | 4 | 0.9848 | 4 | 3.67 |
| | ✗ | ✓ | Feature | Token | RNN | 3.0137 | 5 | 3.2145 | 5 | 0.9850 | 5 | 5.00 |
| | ✗ | ✓ | Temporal | None | MLP | 3.0150 | 6 | 3.2190 | 7 | 0.9858 | 6 | 6.33 |
| | ✗ | ✓ | Temporal | Invert | MLP | 3.0265 | 11 | 3.2172 | 6 | 0.9875 | 7 | 8.00 |
| | ✗ | ✗ | Temporal | Patch | MLP | 3.0213 | 7 | 3.2287 | 10 | 0.9885 | 8 | 8.33 |
| | ✗ | ✗ | Temporal | Freq | MLP | 3.0215 | 8 | 3.2322 | 12 | 0.9890 | 9 | 9.67 |
| | ✗ | ✓ | Feature | Invert | RNN | 3.0270 | 12 | 3.2283 | 9 | 0.9892 | 11 | 10.67 |
| M4_Weekly_S | ✗ | ✓ | Feature | Invert | Trans. | **9.3830** | **1** | **2.7800** | **1** | **1.0125** | **1** | **1.00** |
| | ✗ | ✓ | Feature | Patch | Trans. | **9.4917** | **2** | **2.8413** | **2** | **1.0295** | **2** | **2.00** |
| | ✗ | ✗ | Feature | Patch | Trans. | 9.5275 | 3 | 2.8433 | 3 | 1.0317 | 3 | 3.00 |
| | ✗ | ✗ | Feature | Patch | MLP | 9.5322 | 4 | 2.9158 | 5 | 1.0453 | 4 | 4.33 |
| | ✗ | ✓ | Temporal | Patch | MLP | 9.6020 | 6 | 2.9263 | 6 | 1.0507 | 5 | 5.67 |
| | ✗ | ✗ | Feature | Invert | Trans. | 9.5980 | 5 | 2.9268 | 7 | 1.0508 | 6 | 6.00 |
| | ✗ | ✓ | Temporal | Invert | MLP | 9.9285 | 12 | 2.8515 | 4 | 1.0550 | 7 | 7.67 |
| | ✗ | ✓ | Feature | Patch | MLP | 9.6543 | 8 | 2.9610 | 10 | 1.0600 | 8 | 8.67 |
| | ✗ | ✗ | Temporal | Patch | MLP | 9.6170 | 7 | 2.9830 | 11 | 1.0617 | 9 | 9.00 |
| | ✗ | ✗ | Feature | Patch | RNN | 9.7745 | 9 | 2.9348 | 9 | 1.0620 | 10 | 9.33 |
| M4_Monthly_S | ✓ | ✓ | Feature | Invert | Trans. | 12.5925 | 4 | **0.9193** | **1** | **0.8682** | **1** | **2.00** |
| | ✗ | ✓ | Temporal | Freq | MLP | **12.5025** | **1** | 0.9273 | 6 | **0.8693** | **2** | **3.00** |
| | ✓ | ✓ | Temporal | Freq | MLP | 12.5617 | 3 | 0.9253 | 4 | 0.8705 | 3 | 3.33 |
| | ✓ | ✗ | Feature | Invert | Trans. | 12.6427 | 6 | **0.9195** | **2** | 0.8708 | 5 | 4.33 |
| | ✗ | ✓ | Temporal | None | MLP | **12.5102** | **2** | 0.9290 | 10 | 0.8708 | 4 | 5.33 |
| | ✓ | ✓ | Feature | Patch | RNN | 12.6077 | 5 | 0.9258 | 5 | 0.8725 | 7 | 5.67 |
| | ✓ | ✓ | Feature | Token | RNN | 12.6585 | 8 | 0.9223 | 3 | 0.8723 | 6 | 5.67 |
| | ✓ | ✗ | Feature | Token | RNN | 12.6908 | 15 | 0.9277 | 7 | 0.8760 | 8 | 10.00 |
| | ✓ | ✓ | Temporal | Token | MLP | 12.6453 | 7 | 0.9335 | 13 | 0.8770 | 11 | 10.33 |
| | ✓ | ✓ | Feature | Patch | Trans. | 12.6900 | 14 | 0.9280 | 8 | 0.8762 | 9 | 10.33 |
| M4_Quarterly_S | ✗ | ✓ | Temporal | None | MLP | **9.9145** | **1** | **1.1500** | **1** | **0.8693** | **1** | **1.00** |
| | ✗ | ✓ | Temporal | Freq | MLP | **9.9375** | **2** | **1.1542** | **2** | **0.8722** | **2** | **2.00** |
| | ✗ | ✗ | Temporal | Freq | MLP | 10.0032 | 3 | 1.1687 | 5 | 0.8802 | 3 | 3.67 |
| | ✗ | ✓ | Feature | Invert | Trans. | 10.0330 | 6 | 1.1667 | 4 | 0.8810 | 4 | 4.67 |
| | ✗ | ✓ | Temporal | Freq | MLP | 10.0080 | 4 | 1.1700 | 7 | 0.8810 | 5 | 5.33 |
| | ✗ | ✗ | Feature | Invert | Trans. | 10.0510 | 9 | 1.1667 | 3 | 0.8820 | 6 | 6.00 |
| | ✗ | ✓ | Temporal | Token | MLP | 10.0237 | 5 | 1.1720 | 9 | 0.8825 | 8 | 7.33 |
| | ✗ | ✓ | Temporal | Invert | MLP | 10.0347 | 7 | 1.1712 | 8 | 0.8825 | 7 | 7.33 |
| | ✗ | ✗ | Feature | Patch | Trans. | 10.0590 | 12 | 1.1692 | 6 | 0.8830 | 9 | 9.00 |
| | ✗ | ✗ | Temporal | Invert | MLP | 10.0503 | 8 | 1.1727 | 12 | 0.8840 | 11 | 10.33 |
| M4_Yearly_S | ✗ | ✓ | Temporal | Freq | MLP | **13.2940** | **1** | 2.9985 | 2 | **0.7840** | **1** | **1.33** |
| | ✗ | ✗ | Temporal | Freq | MLP | **13.3235** | **2** | 2.9930 | 1 | **0.7843** | **2** | **1.67** |
| | ✗ | ✓ | Temporal | Invert | MLP | 13.3570 | 3 | 3.0072 | 3 | 0.7870 | 3 | 3.00 |
| | ✗ | ✓ | Temporal | Token | MLP | 13.3673 | 5 | 3.0122 | 5 | 0.7880 | 5 | 5.00 |
| | ✗ | ✓ | Feature | Patch | MLP | 13.3887 | 8 | 3.0077 | 4 | 0.7880 | 4 | 5.33 |
| | ✗ | ✗ | Temporal | Freq | MLP | 13.3937 | 9 | 3.0202 | 11 | 0.7898 | 6 | 8.67 |
| | ✗ | ✓ | Temporal | Invert | RNN | 13.4353 | 14 | 3.0130 | 6 | 0.7900 | 7 | 9.00 |
| | ✗ | ✗ | Temporal | None | MLP | 13.3950 | 10 | 3.0230 | 13 | 0.7903 | 8 | 10.33 |
| | ✓ | ✗ | Temporal | Freq | MLP | 13.3723 | 6 | 3.0380 | 21 | 0.7913 | 9 | 12.00 |
| | ✓ | ✗ | Temporal | None | MLP | 13.3663 | 4 | 3.0393 | 23 | 0.7913 | 10 | 12.33 |

Table 18: Top-10 configurations for the M4 dataset. IN: Instance Norm, SD: Series Decomposition. ✓ indicates module used, ✗ indicates not used. Red/blue highlights indicate best and second-best performances.

| Setup | IN | SD | Fusion | Embed | FF-Type | MSE | Rank | MAE | Rank | Total |
|---|---|---|---|---|---|---|---|---|---|---|
| | ✓ | ✗ | Temporal | Patch | RNN | **0.0550** | **1** | **0.1781** | **1** | **1.0** |
| | ✓ | ✗ | Temporal | None | MLP | **0.0550** | **2** | 0.1784 | 5 | **3.5** |
| | ✓ | ✗ | Feature | Patch | MLP | 0.0552 | 5 | **0.1782** | **2** | **3.5** |
| | ✓ | ✗ | Feature | None | RNN | 0.0552 | 4 | 0.1783 | 3 | **3.5** |
| ETTh1_96_S | ✓ | ✓ | Temporal | Patch | RNN | 0.0552 | 3 | 0.1785 | 6 | 4.5 |
| | ✓ | ✓ | Feature | Invert | MLP | 0.0553 | 6 | 0.1785 | 7 | 6.5 |
| | ✓ | ✓ | Temporal | Invert | RNN | 0.0554 | 10 | 0.1783 | 4 | 7.0 |
| | ✓ | ✓ | Temporal | None | MLP | 0.0554 | 9 | 0.1785 | 8 | 8.5 |
| | ✓ | ✓ | Feature | Invert | RNN | 0.0556 | 14 | 0.1787 | 9 | 11.5 |
| | ✓ | ✗ | Temporal | Invert | MLP | 0.0555 | 11 | 0.1792 | 14 | 12.5 |
| | ✓ | ✗ | Temporal | Freq | MLP | **0.0716** | **1** | 0.2028 | 3 | **2.0** |
| | ✓ | ✗ | Temporal | None | MLP | **0.0717** | **2** | **0.2027** | **2** | **2.0** |
| | ✓ | ✗ | Feature | None | RNN | 0.0718 | 4 | **0.2026** | **1** | **2.5** |
| | ✓ | ✗ | Feature | Invert | RNN | 0.0718 | 3 | 0.2030 | 4 | 3.5 |
| ETTh1_192_S | ✓ | ✗ | Temporal | Token | MLP | 0.0722 | 8 | 0.2035 | 7 | 7.5 |
| | ✓ | ✗ | Temporal | Invert | MLP | 0.0720 | 6 | 0.2036 | 9 | 7.5 |
| | ✓ | ✗ | Feature | Token | RNN | 0.0719 | 5 | 0.2038 | 12 | 8.5 |
| | ✓ | ✓ | Feature | Token | RNN | 0.0722 | 9 | 0.2035 | 8 | 8.5 |
| | ✓ | ✓ | Feature | Patch | MLP | 0.0725 | 14 | 0.2033 | 5 | 9.5 |
| | ✓ | ✗ | Temporal | Patch | RNN | 0.0721 | 7 | 0.2039 | 13 | 10.0 |
| | ✓ | ✓ | Temporal | Freq | MLP | **0.0830** | **1** | **0.2242** | **1** | **1.0** |
| | ✓ | ✓ | Temporal | None | RNN | 0.0833 | 3 | 0.2261 | 4 | **3.5** |
| | ✓ | ✓ | Temporal | Invert | RNN | 0.0840 | 4 | 0.2263 | 6 | 5.0 |
| | ✓ | ✓ | Feature | None | RNN | 0.0844 | 6 | 0.2261 | 5 | 5.5 |
| ETTh1_336_S | ✓ | ✗ | Temporal | Patch | RNN | 0.0842 | 5 | 0.2265 | 9 | 7.0 |
| | ✓ | ✓ | Feature | Token | RNN | 0.0845 | 7 | 0.2264 | 7 | 7.0 |
| | ✓ | ✗ | Temporal | None | MLP | 0.0850 | 13 | **0.2256** | **2** | 7.5 |
| | ✓ | ✓ | Temporal | Token | MLP | 0.0849 | 12 | 0.2258 | 3 | 7.5 |
| | ✓ | ✓ | Temporal | Patch | MLP | **0.0830** | **2** | 0.2270 | 13 | 7.5 |
| | ✓ | ✗ | Temporal | Invert | RNN | 0.0845 | 8 | 0.2266 | 10 | 9.0 |
| | ✓ | ✓ | Temporal | Token | RNN | **0.0834** | **1** | **0.2284** | **2** | **1.5** |
| | ✓ | ✓ | Temporal | Patch | RNN | **0.0841** | **2** | **0.2280** | **1** | **1.5** |
| | ✓ | ✓ | Feature | Token | MLP | 0.0849 | 3 | 0.2302 | 3 | **3.0** |
| | ✓ | ✓ | Temporal | Token | Trans. | 0.0851 | 4 | 0.2306 | 4 | 4.0 |
| ETTh1_720_S | ✓ | ✗ | Temporal | Patch | Trans. | 0.0862 | 5 | 0.2314 | 5 | 5.0 |
| | ✓ | ✗ | Temporal | Patch | RNN | 0.0871 | 8 | 0.2317 | 7 | 7.5 |
| | ✓ | ✗ | Feature | Token | MLP | 0.0867 | 6 | 0.2329 | 9 | 7.5 |
| | ✓ | ✓ | Feature | Patch | MLP | 0.0871 | 9 | 0.2315 | 6 | 7.5 |
| | ✓ | ✓ | Feature | None | MLP | 0.0871 | 7 | 0.2331 | 10 | 8.5 |
| | ✓ | ✗ | Temporal | Invert | RNN | 0.0873 | 10 | 0.2321 | 8 | 9.0 |

Table 19: Top-10 configurations for the ETTh1 dataset univariate forecasting. IN: Instance Norm, SD: Series Decomposition. ✓ indicates module used, ✗ indicates not used. Red/blue highlights indicate best and second-best performances.

| Setup | IN | SD | Fusion | Embed | FF-Type | MSE | Rank | MAE | Rank | Total |
|---|---|---|---|---|---|---|---|---|---|---|
| | ✓ | ✓ | Temporal | Patch | RNN | **0.1275** | **1** | **0.2732** | **1** | **1.0** |
| | ✓ | ✗ | Temporal | Patch | RNN | **0.1275** | **2** | **0.2734** | **2** | **2.0** |
| | ✓ | ✗ | Feature | Patch | MLP | 0.1284 | 3 | 0.2747 | 3 | 3.0 |
| | ✓ | ✗ | Feature | Invert | MLP | 0.1285 | 4 | 0.2747 | 4 | 4.0 |
| ETTh2_96_S | ✓ | ✓ | Temporal | Invert | RNN | 0.1287 | 5 | 0.2750 | 5 | 5.0 |
| | ✓ | ✗ | Feature | None | MLP | 0.1290 | 6 | 0.2759 | 8 | 7.0 |
| | ✓ | ✓ | Temporal | None | MLP | 0.1292 | 8 | 0.2756 | 6 | 7.0 |
| | ✓ | ✗ | Temporal | Invert | RNN | 0.1293 | 9 | 0.2758 | 7 | 8.0 |
| | ✓ | ✓ | Feature | Invert | MLP | 0.1292 | 7 | 0.2760 | 9 | 8.0 |
| | ✓ | ✓ | Feature | Patch | MLP | 0.1296 | 10 | 0.2762 | 10 | 10.0 |
| | ✗ | ✓ | Temporal | Freq | MLP | **0.1680** | **1** | **0.3195** | **1** | **1.0** |
| | ✓ | ✓ | Feature | Patch | MLP | 0.1780 | 4 | **0.3293** | **2** | **3.0** |
| | ✓ | ✓ | Temporal | Patch | MLP | 0.1785 | 5 | 0.3315 | 6 | 5.5 |
| | ✓ | ✗ | Temporal | Invert | RNN | 0.1797 | 11 | 0.3297 | 3 | 7.0 |
| ETTh2_192_S | ✓ | ✓ | Temporal | None | MLP | 0.1794 | 9 | 0.3315 | 7 | 8.0 |
| | ✓ | ✗ | Feature | None | MLP | 0.1804 | 15 | 0.3309 | 4 | 9.5 |
| | ✗ | ✓ | Feature | Invert | MLP | 0.1796 | 10 | 0.3325 | 11 | 10.5 |
| | ✗ | ✓ | Temporal | None | MLP | 0.1787 | 7 | 0.3329 | 15 | 11.0 |
| | ✓ | ✓ | Temporal | Patch | RNN | 0.1802 | 14 | 0.3316 | 8 | 11.0 |
| | ✓ | ✓ | Feature | Invert | MLP | 0.1800 | 13 | 0.3319 | 9 | 11.0 |
| | ✗ | ✗ | Feature | Patch | MLP | **0.1933** | **2** | **0.3464** | **1** | **1.5** |
| | ✗ | ✓ | Feature | Patch | MLP | **0.1908** | **1** | **0.3472** | **2** | **1.5** |
| | ✗ | ✗ | Temporal | Freq | MLP | 0.1947 | 3 | 0.3507 | 3 | **3.0** |
| | ✗ | ✓ | Temporal | Freq | MLP | 0.1969 | 5 | 0.3514 | 4 | 4.5 |
| ETTh2_336_S | ✗ | ✗ | Temporal | Patch | Trans. | 0.2005 | 8 | 0.3531 | 5 | 6.5 |
| | ✗ | ✓ | Temporal | Token | MLP | 0.1969 | 4 | 0.3575 | 9 | 6.5 |
| | ✗ | ✓ | Feature | Invert | MLP | 0.1984 | 6 | 0.3534 | 7 | 6.5 |
| | ✗ | ✓ | Feature | Invert | RNN | 0.1988 | 7 | 0.3532 | 6 | 6.5 |
| | ✗ | ✓ | Feature | Invert | Trans. | 0.2025 | 9 | 0.3564 | 8 | 8.5 |
| | ✗ | ✗ | Feature | Invert | Trans. | 0.2051 | 11 | 0.3586 | 10 | 10.5 |
| | ✓ | ✗ | Temporal | Patch | Trans. | **0.2214** | **1** | **0.3782** | **1** | **1.0** |
| | ✓ | ✗ | Feature | Patch | MLP | **0.2217** | **2** | **0.3790** | **2** | **2.0** |
| | ✓ | ✓ | Temporal | Invert | MLP | 0.2238 | 3 | 0.3806 | 3 | 3.0 |
| | ✓ | ✗ | Feature | Invert | Trans. | 0.2257 | 5 | 0.3814 | 4 | 4.5 |
| ETTh2_720_S | ✓ | ✓ | Feature | Invert | RNN | 0.2256 | 4 | 0.3823 | 5 | 4.5 |
| | ✓ | ✓ | Feature | Patch | MLP | 0.2260 | 6 | 0.3828 | 6 | 6.0 |
| | ✓ | ✗ | Temporal | Token | MLP | 0.2267 | 7 | 0.3834 | 8 | 7.5 |
| | ✓ | ✓ | Temporal | None | MLP | 0.2269 | 8 | 0.3833 | 7 | 7.5 |
| | ✓ | ✗ | Feature | Invert | RNN | 0.2275 | 9 | 0.3840 | 9 | 9.0 |
| | ✓ | ✗ | Temporal | Patch | MLP | 0.2297 | 11 | 0.3847 | 10 | 10.5 |

Table 20: Top-10 configurations for the ETTh2 dataset univariate forecasting. IN: Instance Norm, SD: Series Decomposition. ✓ indicates module used, ✗ indicates not used. Red/blue highlights indicate best and second-best performances.

| Setup | IN | SD | Fusion | Embed | FF-Type | MSE | Rank | MAE | Rank | Total |
|---|---|---|---|---|---|---|---|---|---|---|
| | ✓ | ✓ | Temporal | Patch | RNN | **0.0286** | **1** | **0.1255** | **1** | **1.0** |
| | ✓ | ✓ | Feature | Patch | MLP | **0.0287** | **2** | 0.1258 | 3 | **2.5** |
| | ✓ | ✓ | Feature | Invert | MLP | 0.0287 | 3 | **0.1256** | **2** | **2.5** |
| | ✓ | ✓ | Feature | Patch | RNN | 0.0287 | 4 | 0.1260 | 4 | 4.0 |
| ETTm1_96_S | ✓ | ✓ | Temporal | Invert | RNN | 0.0288 | 6 | 0.1260 | 5 | 5.5 |
| | ✓ | ✓ | Feature | None | RNN | 0.0288 | 5 | 0.1260 | 6 | 5.5 |
| | ✓ | ✓ | Feature | Invert | RNN | 0.0288 | 8 | 0.1261 | 8 | 8.0 |
| | ✓ | ✗ | Temporal | Invert | RNN | 0.0288 | 7 | 0.1263 | 11 | 9.0 |
| | ✓ | ✓ | Feature | None | MLP | 0.0289 | 10 | 0.1263 | 12 | 11.0 |
| | ✓ | ✓ | Temporal | Invert | MLP | 0.0289 | 13 | 0.1262 | 10 | 11.5 |
| | ✓ | ✓ | Temporal | None | RNN | 0.0433 | **1** | 0.1583 | 3 | **2.0** |
| | ✓ | ✓ | Feature | None | MLP | 0.0436 | 3 | **0.1582** | **1** | **2.0** |
| | ✓ | ✗ | Feature | None | MLP | **0.0435** | **2** | 0.1585 | 4 | **3.0** |
| | ✓ | ✓ | Feature | Invert | MLP | 0.0437 | 5 | **0.1582** | **2** | 3.5 |
| ETTm1_192_S | ✓ | ✓ | Feature | Patch | RNN | 0.0436 | 4 | 0.1586 | 6 | 5.0 |
| | ✓ | ✓ | Temporal | Patch | MLP | 0.0438 | 7 | 0.1586 | 5 | 6.0 |
| | ✓ | ✓ | Feature | Token | MLP | 0.0437 | 6 | 0.1587 | 7 | 6.5 |
| | ✓ | ✓ | Temporal | Invert | RNN | 0.0439 | 9 | 0.1588 | 8 | 8.5 |
| | ✓ | ✗ | Temporal | Patch | MLP | 0.0439 | 10 | 0.1588 | 9 | 9.5 |
| | ✓ | ✓ | Temporal | Patch | RNN | 0.0440 | 11 | 0.1588 | 10 | 10.5 |
| | ✓ | ✗ | Feature | None | MLP | **0.0568** | **1** | 0.1837 | **2** | **1.5** |
| | ✓ | ✓ | Feature | None | MLP | **0.0572** | **2** | **0.1831** | **1** | **1.5** |
| | ✓ | ✓ | Temporal | Patch | MLP | 0.0574 | 3 | 0.1840 | 4 | **3.5** |
| | ✓ | ✓ | Feature | Patch | RNN | 0.0576 | 5 | 0.1839 | 3 | 4.0 |
| ETTm1_336_S | ✓ | ✓ | Feature | Token | MLP | 0.0575 | 4 | 0.1843 | 7 | 5.5 |
| | ✓ | ✓ | Temporal | None | MLP | 0.0577 | 7 | 0.1840 | 5 | 6.0 |
| | ✓ | ✓ | Feature | Invert | MLP | 0.0578 | 8 | 0.1841 | 6 | 7.0 |
| | ✓ | ✓ | Temporal | Invert | RNN | 0.0579 | 9 | 0.1845 | 9 | 9.0 |
| | ✓ | ✓ | Temporal | None | RNN | 0.0576 | 6 | 0.1848 | 12 | 9.0 |
| | ✓ | ✓ | Temporal | Freq | MLP | 0.0581 | 11 | 0.1845 | 8 | 9.5 |
| | ✓ | ✓ | Temporal | Invert | RNN | **0.0809** | **2** | **0.2172** | **1** | **1.5** |
| | ✓ | ✓ | Feature | None | MLP | **0.0807** | **1** | **0.2175** | **2** | **1.5** |
| | ✓ | ✗ | Feature | Freq | MLP | 0.0810 | 3 | 0.2179 | 5 | **4.0** |
| | ✓ | ✓ | Temporal | None | MLP | 0.0811 | 4 | 0.2179 | 6 | 5.0 |
| ETTm1_720_S | ✓ | ✓ | Feature | Token | MLP | 0.0811 | 5 | 0.2180 | 9 | 7.0 |
| | ✓ | ✓ | Feature | Invert | RNN | 0.0815 | 11 | 0.2177 | 3 | 7.0 |
| | ✓ | ✓ | Temporal | Freq | MLP | 0.0815 | 12 | 0.2177 | 4 | 8.0 |
| | ✓ | ✓ | Temporal | Token | MLP | 0.0815 | 9 | 0.2182 | 10 | 9.5 |
| | ✓ | ✓ | Temporal | Patch | MLP | 0.0816 | 13 | 0.2179 | 7 | 10.0 |
| | ✓ | ✗ | Temporal | Patch | MLP | 0.0812 | 6 | 0.2185 | 17 | 11.5 |

Table 21: Top-10 configurations for the ETTm1 dataset univariate forecasting. IN: Instance Norm, SD: Series Decomposition. ✓ indicates module used, ✗ indicates not used. Red/blue highlights indicate best and second-best performances.

| Setup | IN | SD | Fusion | Embed | FF-Type | MSE | Rank | MAE | Rank | Total |
|---|---|---|---|---|---|---|---|---|---|---|
| | ✓ | ✓ | Feature | None | RNN | **0.0655** | **1** | **0.1823** | **1** | **1.0** |
| | ✓ | ✗ | Feature | Patch | MLP | 0.0658 | 5 | **0.1828** | **2** | **3.5** |
| | ✓ | ✓ | Feature | Invert | RNN | 0.0657 | 3 | 0.1834 | 4 | **3.5** |
| | ✓ | ✓ | Temporal | Invert | RNN | 0.0658 | 6 | 0.1831 | 3 | 4.5 |
| ETTm2_96_S | ✓ | ✗ | Feature | Invert | Trans | 0.0657 | 4 | 0.1838 | 8 | 6.0 |
| | ✓ | ✗ | Feature | None | RNN | 0.0660 | 7 | 0.1836 | 5 | 6.0 |
| | ✓ | ✗ | Temporal | Patch | MLP | 0.0660 | 8 | 0.1836 | 7 | 7.5 |
| | ✓ | ✓ | Feature | Patch | MLP | 0.0661 | 9 | 0.1836 | 6 | 7.5 |
| | ✓ | ✗ | Temporal | Freq | MLP | **0.0657** | **2** | 0.1848 | 14 | 8.0 |
| | ✓ | ✗ | Temporal | Invert | RNN | 0.0665 | 13 | 0.1844 | 10 | 11.5 |
| | ✓ | ✗ | Feature | Patch | RNN | **0.0989** | **1** | **0.2325** | **1** | **1.0** |
| | ✓ | ✓ | Temporal | Invert | RNN | 0.0996 | 3 | **0.2332** | **2** | **2.5** |
| | ✓ | ✓ | Feature | Patch | MLP | 0.0996 | 5 | 0.2335 | 3 | 4.0 |
| | ✓ | ✓ | Feature | Invert | MLP | 0.0996 | 4 | 0.2338 | 5 | 4.5 |
| ETTm2_192_S | ✓ | ✓ | Feature | None | RNN | 0.0998 | 7 | 0.2337 | 4 | 5.5 |
| | ✓ | ✗ | Feature | None | RNN | 0.0997 | 6 | 0.2341 | 7 | 6.5 |
| | ✓ | ✓ | Temporal | None | MLP | **0.0995** | **2** | 0.2347 | 12 | 7.0 |
| | ✓ | ✓ | Temporal | Patch | MLP | 0.1005 | 13 | 0.2340 | 6 | 9.5 |
| | ✓ | ✓ | Feature | Invert | RNN | 0.1000 | 9 | 0.2345 | 10 | 9.5 |
| | ✓ | ✗ | Temporal | Patch | MLP | 0.1005 | 14 | 0.2344 | 8 | 11.0 |
| | ✓ | ✓ | Feature | None | MLP | **0.1289** | **1** | **0.2723** | **1** | **1.0** |
| | ✓ | ✗ | Feature | None | MLP | **0.1290** | **2** | **0.2728** | **2** | **2.0** |
| | ✓ | ✓ | Feature | Invert | MLP | 0.1297 | 3 | 0.2732 | 3 | 3.0 |
| | ✓ | ✓ | Feature | Patch | MLP | 0.1304 | 6 | 0.2736 | 4 | 5.0 |
| ETTm2_336_S | ✓ | ✓ | Temporal | Token | MLP | 0.1303 | 5 | 0.2744 | 7 | 6.0 |
| | ✓ | ✓ | Temporal | Patch | RNN | 0.1308 | 8 | 0.2743 | 6 | 7.0 |
| | ✓ | ✗ | Temporal | Token | MLP | 0.1302 | 4 | 0.2746 | 12 | 8.0 |
| | ✓ | ✗ | Temporal | Patch | RNN | 0.1311 | 13 | 0.2740 | 5 | 9.0 |
| | ✓ | ✗ | Temporal | Invert | RNN | 0.1308 | 7 | 0.2745 | 11 | 9.0 |
| | ✓ | ✓ | Temporal | Invert | RNN | 0.1309 | 9 | 0.2744 | 9 | 9.0 |
| | ✓ | ✗ | Feature | None | MLP | **0.1795** | **1** | **0.3290** | **2** | **1.5** |
| | ✓ | ✓ | Feature | None | MLP | **0.1802** | **2** | **0.3287** | **1** | **1.5** |
| | ✓ | ✓ | Temporal | Patch | MLP | 0.1824 | 5 | 0.3306 | 3 | **4.0** |
| | ✓ | ✓ | Temporal | None | RNN | 0.1822 | 3 | 0.3315 | 5 | **4.0** |
| ETTm2_720_S | ✓ | ✓ | Feature | Invert | MLP | 0.1823 | 4 | 0.3312 | 4 | 4.0 |
| | ✓ | ✓ | Temporal | Token | MLP | 0.1827 | 6 | 0.3318 | 7 | 6.5 |
| | ✓ | ✓ | Temporal | Invert | RNN | 0.1829 | 8 | 0.3318 | 6 | 7.0 |
| | ✓ | ✓ | Temporal | Freq | MLP | 0.1830 | 9 | 0.3320 | 9 | 9.0 |
| | ✓ | ✗ | Temporal | Freq | MLP | 0.1828 | 7 | 0.3322 | 13 | 10.0 |
| | ✓ | ✗ | Temporal | Invert | RNN | 0.1833 | 11 | 0.3322 | 11 | 11.0 |

Table 22: Top-10 configurations for the ETTm2 dataset univariate forecasting. IN: Instance Norm, SD: Series Decomposition. ✓ indicates module used, ✗ indicates not used. Red/blue highlights indicate best and second-best performances.

| Setup | IN | SD | Fusion | Embed | FF-Type | MSE | Rank | MAE | Rank | Total |
|-------|----|----|--------|-------|---------|-----|------|-----|------|-------|
| | ✓ | ✓ | Temporal | Patch | RNN | **0.0887** | **1** | **0.2202** | **1** | **1.0** |
| | ✓ | ✓ | Temporal | Patch | MLP | **0.0923** | **2** | **0.2230** | **2** | **2.0** |
| | ✓ | ✗ | Temporal | Token | MLP | 0.0925 | 3 | 0.2246 | 4 | 3.5 |
| | ✓ | ✗ | Feature | Patch | MLP | 0.0942 | 4 | 0.2237 | 3 | 3.5 |
| Exchange_96_S | ✓ | ✓ | Feature | Invert | MLP | 0.0974 | 7 | 0.2283 | 6 | 6.5 |
| | ✓ | ✓ | Temporal | Freq | MLP | 0.0971 | 6 | 0.2285 | 8 | 7.0 |
| | ✓ | ✗ | Temporal | Invert | RNN | 0.0969 | 5 | 0.2298 | 10 | 7.5 |
| | ✓ | ✓ | Feature | Patch | MLP | 0.0995 | 11 | 0.2281 | 5 | 8.0 |
| | ✓ | ✓ | Temporal | None | MLP | 0.0978 | 8 | 0.2293 | 9 | 8.5 |
| | ✓ | ✓ | Feature | None | RNN | 0.0986 | 10 | 0.2299 | 11 | 10.5 |
| | ✓ | ✗ | Temporal | Patch | MLP | **0.1998** | **1** | **0.3336** | **1** | **1.0** |
| | ✓ | ✓ | Temporal | Patch | RNN | 0.2047 | 4 | **0.3353** | **2** | **3.0** |
| | ✓ | ✓ | Temporal | Invert | RNN | 0.2046 | 3 | 0.3356 | 5 | 4.0 |
| | ✓ | ✗ | Temporal | Invert | RNN | 0.2056 | 7 | 0.3355 | 4 | 5.5 |
| Exchange_192_S | ✓ | ✗ | Temporal | Token | MLP | **0.2038** | **2** | 0.3390 | 11 | 6.5 |
| | ✓ | ✓ | Feature | None | RNN | 0.2068 | 9 | 0.3369 | 6 | 7.5 |
| | ✓ | ✗ | Feature | Patch | MLP | 0.2082 | 15 | 0.3354 | 3 | 9.0 |
| | ✓ | ✗ | Feature | Invert | MLP | 0.2053 | 5 | 0.3393 | 13 | 9.0 |
| | ✓ | ✓ | Feature | Patch | MLP | 0.2072 | 12 | 0.3370 | 7 | 9.5 |
| | ✓ | ✓ | Temporal | Invert | MLP | 0.2055 | 6 | 0.3404 | 14 | 10.0 |
| | ✓ | ✗ | Feature | None | Trans | **0.4017** | **1** | **0.4832** | **1** | **1.0** |
| | ✓ | ✗ | Feature | None | RNN | 0.4168 | 4 | 0.4854 | 5 | **4.5** |
| | ✓ | ✗ | Feature | Patch | RNN | 0.4204 | 8 | 0.4852 | 3 | 5.5 |
| | ✓ | ✓ | Feature | Patch | RNN | 0.4189 | 6 | 0.4865 | 6 | 6.0 |
| Exchange_336_S | ✓ | ✗ | Temporal | Patch | RNN | 0.4217 | 11 | **0.4841** | **2** | 6.5 |
| | ✓ | ✓ | Feature | None | RNN | 0.4205 | 9 | 0.4853 | 4 | 6.5 |
| | ✓ | ✗ | Temporal | None | MLP | **0.4166** | **2** | 0.4898 | 13 | 7.5 |
| | ✓ | ✗ | Temporal | Token | MLP | 0.4182 | 5 | 0.4901 | 17 | 11.0 |
| | ✓ | ✓ | Temporal | Patch | MLP | 0.4247 | 15 | 0.4872 | 8 | 11.5 |
| | ✓ | ✓ | Feature | Invert | RNN | 0.4247 | 16 | 0.4880 | 10 | 13.0 |
| | ✗ | ✗ | Temporal | Token | RNN | **0.7083** | **1** | **0.7244** | **1** | **1.0** |
| | ✗ | ✗ | Temporal | Invert | MLP | 0.8477 | 3 | **0.7296** | **2** | 2.5 |
| | ✗ | ✓ | Feature | Token | MLP | **0.8456** | **2** | 0.7729 | 5 | 3.5 |
| | ✗ | ✗ | Temporal | Invert | RNN | 0.8950 | 5 | 0.7391 | 3 | 4.0 |
| Exchange_720_S | ✗ | ✗ | Temporal | Freq | MLP | 0.9377 | 6 | 0.7404 | 4 | 5.0 |
| | ✗ | ✗ | Feature | Token | MLP | 0.8575 | 4 | 0.7779 | 6 | 5.0 |
| | ✓ | ✗ | Feature | None | Trans | 1.0907 | 11 | 0.7944 | 8 | 9.5 |
| | ✓ | ✗ | Temporal | None | MLP | 1.0925 | 14 | 0.7931 | 7 | 10.5 |
| | ✓ | ✓ | Feature | None | MLP | 1.0923 | 13 | 0.7972 | 9 | 11.0 |
| | ✓ | ✗ | Feature | Token | Trans | 1.0908 | 12 | 0.8024 | 11 | 11.5 |

Table 23: Top-10 configurations for the Exchange Rate dataset univariate forecasting. IN: Instance Norm, SD: Series Decomposition. ✓ indicates module used, ✗ indicates not used. Red/blue highlights indicate best and second-best performances.

| Setup | IN | SD | Fusion | Embed | FF-Type | MSE | Rank | MAE | Rank | Total |
|---|---|---|---|---|---|---|---|---|---|---|
| | ✓ | ✓ | Temporal | Patch | RNN | **0.6131** | **1** | 0.5881 | 3 | **2.0** |
| | ✓ | ✓ | Feature | Patch | Trans. | **0.6292** | **2** | **0.5877** | **2** | **2.0** |
| | ✓ | ✓ | Temporal | Patch | MLP | 0.6405 | 4 | **0.5831** | **1** | **2.5** |
| | ✓ | ✓ | Temporal | Invert | RNN | 0.6338 | 3 | 0.5981 | 6 | 4.5 |
| ILI_24_S | ✓ | ✓ | Feature | Patch | RNN | 0.6613 | 7 | 0.5920 | 4 | 5.5 |
| | ✓ | ✓ | Feature | None | Trans. | 0.6568 | 6 | 0.5949 | 5 | 5.5 |
| | ✓ | ✗ | Feature | None | Trans. | 0.6564 | 5 | 0.6093 | 12 | 8.5 |
| | ✓ | ✗ | Temporal | Patch | RNN | 0.6904 | 10 | 0.6036 | 8 | 9.0 |
| | ✓ | ✓ | Temporal | Token | MLP | 0.6968 | 12 | 0.6036 | 7 | 9.5 |
| | ✓ | ✓ | Feature | Invert | MLP | 0.6791 | 9 | 0.6081 | 10 | 9.5 |
| | ✓ | ✓ | Temporal | Patch | MLP | **0.6331** | **1** | **0.6096** | **1** | **1.0** |
| | ✓ | ✓ | Feature | Patch | Trans | **0.6416** | **2** | **0.6118** | **2** | **2.0** |
| | ✓ | ✓ | Temporal | Token | RNN | 0.6491 | 3 | 0.6270 | 3 | 3.0 |
| | ✓ | ✓ | Feature | Invert | MLP | 0.6713 | 6 | 0.6312 | 5 | 5.5 |
| ILI_36_S | ✓ | ✓ | Temporal | Patch | RNN | 0.6636 | 4 | 0.6345 | 8 | 6.0 |
| | ✓ | ✓ | Feature | Patch | RNN | 0.6790 | 8 | 0.6302 | 4 | 6.0 |
| | ✓ | ✗ | Temporal | Patch | RNN | 0.6649 | 5 | 0.6381 | 10 | 7.5 |
| | ✓ | ✓ | Feature | Patch | MLP | 0.7038 | 12 | 0.6321 | 6 | 9.0 |
| | ✓ | ✓ | Temporal | Invert | RNN | 0.6750 | 7 | 0.6462 | 12 | 9.5 |
| | ✓ | ✗ | Feature | Patch | Trans. | 0.7069 | 14 | 0.6327 | 7 | 10.5 |
| | ✓ | ✓ | Feature | Patch | Trans. | **0.6473** | **1** | **0.6268** | **1** | **1.0** |
| | ✓ | ✓ | Temporal | Patch | MLP | **0.6568** | **2** | **0.6412** | **2** | **2.0** |
| | ✓ | ✗ | Feature | Invert | Trans. | 0.6850 | 4 | 0.6439 | 3 | 3.5 |
| | ✓ | ✓ | Feature | Patch | RNN | 0.6836 | 3 | 0.6539 | 5 | 4.0 |
| ILI_48_S | ✓ | ✗ | Feature | Patch | RNN | 0.7004 | 6 | 0.6603 | 8 | 7.0 |
| | ✓ | ✗ | Feature | Patch | Trans. | 0.7013 | 8 | 0.6584 | 7 | 7.5 |
| | ✓ | ✓ | Temporal | Token | RNN | 0.6879 | 5 | 0.6635 | 10 | 7.5 |
| | ✓ | ✓ | Temporal | Invert | MLP | 0.7047 | 12 | 0.6622 | 9 | 10.5 |
| | ✓ | ✗ | Temporal | Invert | MLP | 0.7113 | 18 | 0.6645 | 11 | 14.5 |
| | ✓ | ✗ | Feature | Token | RNN | 0.7095 | 15 | 0.6671 | 14 | 14.5 |
| | ✓ | ✗ | Feature | Patch | Trans. | **0.6730** | **1** | **0.6682** | **1** | **1.0** |
| | ✓ | ✗ | Feature | Invert | Trans. | **0.6822** | **2** | 0.6739 | 3 | **2.5** |
| | ✓ | ✓ | Temporal | Patch | MLP | 0.6992 | 4 | 0.6830 | 7 | 5.5 |
| | ✓ | ✓ | Feature | Patch | Trans. | 0.7022 | 6 | 0.6784 | 5 | 5.5 |
| ILI_60_S | ✓ | ✗ | Temporal | Patch | MLP | 0.6969 | 3 | 0.6849 | 9 | 6.0 |
| | ✓ | ✗ | Temporal | Patch | Trans. | 0.7154 | 13 | 0.6744 | 4 | 8.5 |
| | ✓ | ✗ | Temporal | Invert | MLP | 0.7008 | 5 | 0.6883 | 13 | 9.0 |
| | ✓ | ✓ | Temporal | Patch | Trans. | 0.7215 | 16 | **0.6715** | **2** | 9.0 |
| | ✓ | ✓ | Feature | Invert | Trans. | 0.7085 | 8 | 0.6872 | 11 | 9.5 |
| | ✓ | ✓ | Temporal | Invert | MLP | 0.7113 | 10 | 0.6866 | 10 | 10.0 |

Table 24: Top-10 configurations for the Illness (National Flu) dataset univariate forecasting. IN: Instance Norm, SD: Series Decomposition. ✓ indicates module used, ✗ indicates not used. Red/blue highlights indicate best and second-best performances.

| Setup | IN | SD | Fusion | Embed | FF-Type | MSE | Rank | MAE | Rank | Total |
|---|---|---|---|---|---|---|---|---|---|---|
| | ✗ | ✓ | Feature | Invert | Trans. | **0.2829** | **2** | **0.3794** | **2** | **2.0** |
| | ✗ | ✓ | Feature | None | Trans. | **0.2825** | **1** | 0.3798 | 3 | **2.0** |
| | ✗ | ✗ | Feature | Invert | Trans. | 0.2899 | 5 | **0.3768** | **1** | **3.0** |
| | ✗ | ✗ | Feature | Invert | MLP | 0.2879 | 4 | 0.3829 | 4 | 4.0 |
| ECL_96_S | ✗ | ✓ | Feature | Invert | MLP | 0.2868 | 3 | 0.3833 | 5 | 4.0 |
| | ✓ | ✗ | Feature | Invert | Trans. | 0.2946 | 7 | 0.3848 | 7 | 7.0 |
| | ✗ | ✗ | Feature | None | Trans. | 0.2930 | 6 | 0.3852 | 9 | 7.5 |
| | ✓ | ✗ | Feature | None | RNN | 0.2973 | 11 | 0.3838 | 6 | 8.5 |
| | ✗ | ✗ | Feature | Patch | RNN | 0.2962 | 10 | 0.3852 | 8 | 9.0 |
| | ✓ | ✓ | Feature | Invert | MLP | 0.2947 | 8 | 0.3899 | 14 | 11.0 |
| | ✗ | ✓ | Feature | Invert | MLP | **0.3013** | **1** | **0.3917** | **2** | **1.5** |
| | ✗ | ✓ | Feature | None | Trans. | **0.3019** | **2** | **0.3912** | **1** | **1.5** |
| | ✗ | ✗ | Feature | Invert | MLP | 0.3107 | 3 | 0.3961 | 5 | **4.0** |
| | ✗ | ✗ | Feature | Invert | Trans. | 0.3128 | 5 | 0.3931 | 3 | **4.0** |
| ECL_192_S | ✗ | ✗ | Feature | None | Trans. | 0.3112 | 4 | 0.3957 | 4 | **4.0** |
| | ✗ | ✓ | Feature | Invert | Trans. | 0.3145 | 6 | 0.4006 | 9 | 7.5 |
| | ✓ | ✗ | Feature | Invert | Trans. | 0.3199 | 9 | 0.3962 | 6 | 7.5 |
| | ✓ | ✓ | Feature | Invert | MLP | 0.3193 | 8 | 0.4007 | 10 | 9.0 |
| | ✗ | ✓ | Temporal | Patch | RNN | 0.3223 | 11 | 0.4009 | 11 | 11.0 |
| | ✗ | ✓ | Temporal | Invert | RNN | 0.3183 | 7 | 0.4043 | 20 | 13.5 |
| | ✗ | ✓ | Feature | Invert | MLP | **0.3437** | **1** | 0.4245 | 3 | **2.0** |
| | ✗ | ✓ | Feature | Invert | Trans. | **0.3452** | **2** | **0.4243** | **2** | **2.0** |
| | ✗ | ✓ | Feature | None | Trans. | 0.3462 | 3 | 0.4250 | 4 | **3.5** |
| | ✗ | ✗ | Feature | Invert | Trans. | 0.3575 | 7 | **0.4226** | **1** | 4.0 |
| ECL_336_S | ✗ | ✓ | Temporal | Invert | RNN | 0.3489 | 4 | 0.4277 | 6 | 5.0 |
| | ✗ | ✗ | Feature | None | Trans. | 0.3554 | 6 | 0.4270 | 5 | 5.5 |
| | ✗ | ✗ | Feature | Invert | MLP | 0.3546 | 5 | 0.4287 | 8 | 6.5 |
| | ✓ | ✓ | Feature | Invert | MLP | 0.3665 | 9 | 0.4286 | 7 | 8.0 |
| | ✗ | ✓ | Feature | Patch | RNN | 0.3578 | 8 | 0.4295 | 9 | 8.5 |
| | ✗ | ✗ | Feature | Patch | Trans. | 0.3665 | 10 | 0.4327 | 13 | 11.5 |
| | ✗ | ✓ | Feature | Invert | MLP | **0.3867** | **1** | **0.4638** | **1** | **1.0** |
| | ✗ | ✗ | Feature | Invert | MLP | 0.3997 | 3 | 0.4682 | 3 | **3.0** |
| | ✗ | ✓ | Feature | None | Trans. | **0.3974** | **2** | 0.4691 | 4 | **3.0** |
| | ✗ | ✗ | Feature | Invert | Trans. | 0.4095 | 6 | **0.4646** | **2** | 4.0 |
| ECL_720_S | ✗ | ✓ | Feature | Invert | Trans. | 0.4067 | 4 | 0.4745 | 7 | 5.5 |
| | ✗ | ✗ | Feature | None | Trans. | 0.4167 | 10 | 0.4706 | 5 | 7.5 |
| | ✗ | ✓ | Temporal | Invert | RNN | 0.4112 | 7 | 0.4756 | 9 | 8.0 |
| | ✗ | ✓ | Feature | Freq | MLP | 0.4076 | 5 | 0.4764 | 11 | 8.0 |
| | ✗ | ✗ | Feature | Patch | MLP | 0.4165 | 9 | 0.4756 | 8 | 8.5 |
| | ✗ | ✗ | Temporal | Patch | RNN | 0.4237 | 12 | 0.4724 | 6 | 9.0 |

Table 25: Top-10 configurations for the Electricity (ECL) dataset univariate forecasting. IN: Instance Norm, SD: Series Decomposition. ✓ indicates module used, ✗ indicates not used. Red/blue highlights indicate best and second-best performances.

| Setup | IN | SD | Fusion | Embed | FF-Type | MSE | Rank | MAE | Rank | Total |
|---|---|---|---|---|---|---|---|---|---|---|
| | ✓ | ✓ | Temporal | Patch | RNN | **0.0012** | **1** | **0.0254** | **1** | **1.0** |
| | ✓ | ✗ | Temporal | Patch | RNN | **0.0012** | **2** | **0.0255** | **2** | **2.0** |
| | ✓ | ✓ | Temporal | Invert | RNN | 0.0013 | 3 | 0.0258 | 4 | 3.5 |
| | ✓ | ✗ | Temporal | Invert | RNN | 0.0013 | 4 | 0.0260 | 5 | 4.5 |
| Weather_96_S | ✓ | ✗ | Feature | Patch | RNN | 0.0013 | 6 | 0.0256 | 3 | 4.5 |
| | ✓ | ✓ | Feature | None | MLP | 0.0013 | 5 | 0.0261 | 7 | 6.0 |
| | ✓ | ✓ | Feature | Patch | MLP | 0.0013 | 8 | 0.0261 | 6 | 7.0 |
| | ✓ | ✗ | Feature | None | MLP | 0.0013 | 7 | 0.0263 | 8 | 7.5 |
| | ✓ | ✓ | Feature | Patch | RNN | 0.0013 | 9 | 0.0264 | 10 | 9.5 |
| | ✓ | ✗ | Feature | Patch | MLP | 0.0013 | 11 | 0.0264 | 9 | 10.0 |
| | ✓ | ✗ | Temporal | Token | RNN | **0.0015** | **1** | **0.0280** | **1** | **1.0** |
| | ✓ | ✗ | Temporal | Token | Trans. | **0.0015** | **2** | 0.0286 | 3 | **2.5** |
| | ✓ | ✓ | Temporal | Token | RNN | 0.0015 | 3 | 0.0287 | 6 | 4.5 |
| | ✓ | ✓ | Temporal | Freq | MLP | 0.0015 | 7 | **0.0286** | **2** | 4.5 |
| Weather_192_S | ✓ | ✓ | Temporal | Patch | RNN | 0.0015 | 5 | 0.0287 | 5 | 5.0 |
| | ✓ | ✓ | Temporal | Invert | RNN | 0.0015 | 4 | 0.0288 | 7 | 5.5 |
| | ✓ | ✗ | Feature | Patch | RNN | 0.0015 | 9 | 0.0286 | 4 | 6.5 |
| | ✓ | ✗ | Feature | None | MLP | 0.0015 | 6 | 0.0288 | 8 | 7.0 |
| | ✓ | ✗ | Feature | Token | MLP | 0.0015 | 8 | 0.0291 | 11 | 9.5 |
| | ✓ | ✓ | Feature | Patch | RNN | 0.0015 | 10 | 0.0291 | 13 | 11.5 |
| | ✓ | ✗ | Feature | Patch | RNN | 0.0017 | 4 | **0.0300** | **1** | **2.5** |
| | ✓ | ✓ | Temporal | Token | RNN | **0.0016** | **1** | 0.0302 | 6 | **3.5** |
| | ✓ | ✗ | Temporal | Token | Trans. | 0.0017 | 6 | **0.0301** | **2** | 4.0 |
| | ✓ | ✓ | Temporal | Patch | RNN | **0.0016** | **2** | 0.0303 | 9 | 5.5 |
| Weather_336_S | ✓ | ✓ | Feature | None | MLP | 0.0017 | 7 | 0.0301 | 4 | 5.5 |
| | ✓ | ✗ | Temporal | None | MLP | 0.0017 | 3 | 0.0304 | 11 | 7.0 |
| | ✓ | ✓ | Temporal | Freq | MLP | 0.0017 | 13 | 0.0301 | 3 | 8.0 |
| | ✓ | ✗ | Feature | Token | MLP | 0.0017 | 11 | 0.0303 | 7 | 9.0 |
| | ✓ | ✗ | Feature | Invert | MLP | 0.0017 | 5 | 0.0307 | 13 | 9.0 |
| | ✓ | ✗ | Temporal | Freq | MLP | 0.0017 | 15 | 0.0302 | 5 | 10.0 |
| | ✓ | ✗ | Temporal | Token | Trans. | **0.0021** | **2** | **0.0335** | **1** | **1.5** |
| | ✓ | ✓ | Temporal | Freq | MLP | **0.0021** | **1** | **0.0337** | **2** | **1.5** |
| | ✓ | ✗ | Temporal | Freq | MLP | 0.0021 | 3 | 0.0337 | 3 | **3.0** |
| | ✓ | ✗ | Feature | Patch | RNN | 0.0021 | 4 | 0.0341 | 5 | 4.5 |
| Weather_720_S | ✓ | ✗ | Feature | Token | MLP | 0.0021 | 6 | 0.0340 | 4 | 5.0 |
| | ✓ | ✗ | Feature | None | MLP | 0.0021 | 10 | 0.0341 | 6 | 8.0 |
| | ✓ | ✓ | Temporal | None | MLP | 0.0021 | 5 | 0.0344 | 11 | 8.0 |
| | ✓ | ✓ | Temporal | Token | RNN | 0.0021 | 7 | 0.0346 | 13 | 10.0 |
| | ✓ | ✓ | Feature | Token | MLP | 0.0022 | 14 | 0.0341 | 7 | 10.5 |
| | ✓ | ✗ | Temporal | None | MLP | 0.0021 | 9 | 0.0347 | 14 | 11.5 |

Table 26: Top-10 configurations for the Weather dataset univariate forecasting. IN: Instance Norm, SD: Series Decomposition. ✓ indicates module used, ✗ indicates not used. Red/blue highlights indicate best and second-best performances.

| Setup | IN | SD | Fusion | Embed | FF-Type | MSE | Rank | MAE | Rank | Total |
|---|---|---|---|---|---|---|---|---|---|---|
| | ✓ | ✓ | Temporal | Freq | MLP | **0.0325** | **1** | **0.1331** | **1** | **1.0** |
| | ✗ | ✓ | Temporal | Freq | MLP | **0.0333** | **2** | **0.1340** | **2** | 2.0 |
| | ✗ | ✗ | Temporal | Token | MLP | 0.0346 | 3 | 0.1357 | 4 | 3.5 |
| | ✓ | ✓ | Temporal | Token | MLP | 0.0346 | 4 | 0.1353 | 3 | 3.5 |
| PEMS03_12_S | ✗ | ✗ | Feature | Token | RNN | 0.0350 | 5 | 0.1368 | 5 | 5.0 |
| | ✗ | ✗ | Temporal | Token | RNN | 0.0350 | 6 | 0.1371 | 6 | 6.0 |
| | ✓ | ✗ | Feature | Patch | RNN | 0.0354 | 7 | 0.1381 | 8 | 7.5 |
| | ✗ | ✗ | Temporal | None | MLP | 0.0359 | 11 | 0.1378 | 7 | 9.0 |
| | ✓ | ✗ | Feature | Token | RNN | 0.0358 | 9 | 0.1383 | 9 | 9.0 |
| | ✗ | ✗ | Feature | Patch | RNN | 0.0358 | 10 | 0.1384 | 10 | 10.0 |
| | ✓ | ✓ | Temporal | Freq | MLP | **0.0467** | **1** | **0.1548** | **1** | **1.0** |
| | ✗ | ✗ | Feature | Patch | RNN | 0.0473 | 3 | **0.1564** | **2** | 2.5 |
| | ✓ | ✗ | Feature | Token | RNN | **0.0471** | **2** | 0.1565 | 3 | 2.5 |
| | ✗ | ✗ | Temporal | Token | RNN | 0.0480 | 4 | 0.1582 | 7 | 5.5 |
| PEMS03_24_S | ✓ | ✓ | Temporal | None | MLP | 0.0488 | 6 | 0.1580 | 6 | 6.0 |
| | ✓ | ✗ | Feature | Patch | RNN | 0.0490 | 7 | 0.1577 | 5 | 6.0 |
| | ✗ | ✗ | Temporal | Token | MLP | 0.0499 | 10 | **0.1575** | **4** | 7.0 |
| | ✗ | ✗ | Feature | Token | RNN | 0.0483 | 5 | 0.1595 | 10 | 7.5 |
| | ✗ | ✗ | Temporal | Freq | MLP | 0.0499 | 9 | 0.1582 | 9 | 9.0 |
| | ✗ | ✗ | Feature | Patch | Trans. | 0.0493 | 8 | 0.1598 | 11 | 9.5 |
| | ✗ | ✗ | Feature | Patch | RNN | **0.0575** | **1** | **0.1678** | **1** | **1.0** |
| | ✗ | ✗ | Feature | Patch | Trans. | **0.0586** | **2** | 0.1728 | 5 | 3.5 |
| | ✗ | ✗ | Temporal | None | MLP | 0.0593 | 4 | 0.1715 | 4 | 4.0 |
| | ✓ | ✗ | Feature | Patch | RNN | 0.0607 | 6 | **0.1690** | **2** | 4.0 |
| PEMS03_36_S | ✓ | ✗ | Feature | Token | RNN | 0.0598 | 5 | 0.1713 | 3 | 4.0 |
| | ✗ | ✗ | Temporal | Token | RNN | 0.0587 | 3 | 0.1736 | 6 | 4.5 |
| | ✓ | ✓ | Temporal | None | MLP | 0.0614 | 7 | 0.1741 | 7 | 7.0 |
| | ✗ | ✗ | Temporal | Freq | MLP | 0.0634 | 9 | 0.1760 | 9 | 9.0 |
| | ✗ | ✗ | Feature | Token | RNN | 0.0643 | 10 | 0.1758 | 8 | 9.0 |
| | ✓ | ✗ | Feature | Patch | Trans. | 0.0619 | 8 | 0.1797 | 11 | 9.5 |
| | ✗ | ✗ | Feature | Patch | RNN | 0.0675 | 2 | **0.1771** | **1** | **1.5** |
| | ✗ | ✗ | Temporal | None | MLP | 0.0702 | 4 | **0.1807** | **3** | 3.5 |
| | ✗ | ✗ | Feature | Patch | Trans. | **0.0672** | **1** | 0.1858 | 7 | 4.0 |
| | ✓ | ✗ | Feature | Patch | RNN | 0.0720 | 7 | **0.1778** | **2** | 4.5 |
| PEMS03_48_S | ✗ | ✗ | Feature | Token | RNN | 0.0714 | 5 | 0.1843 | 6 | 5.5 |
| | ✓ | ✓ | Temporal | None | MLP | 0.0741 | 9 | 0.1827 | 4 | 6.5 |
| | ✓ | ✗ | Feature | Patch | Trans. | **0.0701** | **3** | 0.1896 | 10 | 6.5 |
| | ✓ | ✗ | Feature | Token | RNN | 0.0739 | 8 | 0.1840 | 5 | 6.5 |
| | ✗ | ✗ | Temporal | Token | RNN | 0.0718 | 6 | 0.1875 | 8 | 7.0 |
| | ✓ | ✓ | Temporal | Token | RNN | 0.0778 | 11 | 0.1894 | 9 | 10.0 |

Table 27: Top-10 configurations for the PEMS03 dataset univariate forecasting. IN: Instance Norm, SD: Series Decomposition. ✓ indicates module used, ✗ indicates not used. Red/blue highlights indicate best and second-best performances.

| Setup | IN | SD | Fusion | Embed | FF-Type | MSE | Rank | MAE | Rank | Total |
|---|---|---|---|---|---|---|---|---|---|---|
| | ✓ | ✓ | Temporal | Freq | MLP | **0.0483** | **1** | **0.1638** | **2** | **1.5** |
| | ✗ | ✗ | Temporal | Freq | MLP | **0.0488** | **3** | **0.1637** | **1** | 2.0 |
| | ✗ | ✗ | Feature | Invert | Trans. | 0.0485 | 2 | 0.1646 | 3 | 2.5 |
| | ✗ | ✗ | Feature | Token | RNN | 0.0495 | 4 | 0.1648 | 4 | 4.0 |
| PEMS04_12_S | ✓ | ✓ | Temporal | Invert | MLP | 0.0496 | 5 | 0.1656 | 5 | 5.0 |
| | ✗ | ✗ | Feature | Patch | Trans. | 0.0496 | 6 | 0.1658 | 6 | 6.0 |
| | ✗ | ✗ | Feature | Patch | RNN | 0.0503 | 7 | 0.1660 | 7 | 7.0 |
| | ✗ | ✗ | Temporal | Invert | MLP | 0.0508 | 11 | 0.1670 | 8 | 9.5 |
| | ✗ | ✗ | Feature | Invert | RNN | 0.0507 | 10 | 0.1675 | 9 | 9.5 |
| | ✓ | ✓ | Temporal | None | MLP | 0.0506 | 9 | 0.1678 | 11 | 10.0 |
| | ✓ | ✗ | Feature | Patch | Trans. | **0.0607** | **1** | **0.1838** | **1** | **1.0** |
| | ✗ | ✗ | Feature | Invert | Trans. | **0.0655** | **2** | **0.1838** | **2** | 2.0 |
| | ✗ | ✗ | Feature | Token | RNN | 0.0687 | 4 | 0.1866 | 3 | 3.5 |
| | ✓ | ✗ | Feature | Invert | Trans. | 0.0700 | 6 | 0.1916 | 7 | 6.5 |
| PEMS04_24_S | ✓ | ✗ | Feature | None | Trans. | 0.0703 | 9 | 0.1918 | 8 | 8.5 |
| | ✓ | ✗ | Feature | Token | RNN | 0.0713 | 12 | 0.1914 | 6 | 9.0 |
| | ✗ | ✗ | Feature | Invert | RNN | 0.0711 | 10 | 0.1926 | 10 | 10.0 |
| | ✗ | ✗ | Temporal | Patch | Trans. | 0.0669 | 3 | 0.1952 | 18 | 10.5 |
| | ✗ | ✗ | Feature | Patch | RNN | 0.0720 | 17 | 0.1898 | 4 | 10.5 |
| | ✗ | ✗ | Feature | Patch | Trans. | 0.0719 | 16 | 0.1906 | 5 | 10.5 |
| | ✗ | ✗ | Feature | Patch | RNN | **0.0652** | **2** | **0.1867** | **1** | **1.5** |
| | ✗ | ✗ | Feature | Invert | Trans. | **0.0645** | **1** | 0.1877 | 3 | 2.0 |
| | ✗ | ✗ | Feature | Patch | Trans. | 0.0652 | 3 | **0.1874** | **2** | 2.5 |
| | ✓ | ✗ | Feature | Patch | RNN | 0.0659 | 4 | 0.1885 | 4 | 4.0 |
| PEMS04_36_S | ✗ | ✗ | Temporal | Freq | MLP | 0.0667 | 5 | 0.1909 | 5 | 5.0 |
| | ✓ | ✗ | Feature | Invert | Trans. | 0.0686 | 6 | 0.1936 | 6 | 6.0 |
| | ✓ | ✗ | Feature | Patch | Trans. | 0.0687 | 7 | 0.1948 | 7 | 7.0 |
| | ✗ | ✗ | Temporal | Invert | MLP | 0.0702 | 8 | 0.1957 | 9 | 8.5 |
| | ✗ | ✗ | Feature | Token | RNN | 0.0711 | 10 | 0.1954 | 8 | 9.0 |
| | ✗ | ✗ | Temporal | None | MLP | 0.0708 | 9 | 0.1958 | 10 | 9.5 |
| | ✗ | ✗ | Feature | Patch | Trans. | **0.0670** | **1** | **0.1900** | **1** | **1.0** |
| | ✗ | ✗ | Feature | Patch | RNN | 0.0709 | 3 | **0.1939** | **2** | 2.5 |
| | ✓ | ✗ | Feature | Patch | RNN | **0.0706** | **2** | 0.1949 | 3 | 2.5 |
| | ✓ | ✗ | Feature | Patch | Trans. | 0.0721 | 4 | 0.1970 | 4 | 4.0 |
| PEMS04_48_S | ✗ | ✗ | Feature | Invert | Trans. | 0.0729 | 5 | 0.1978 | 5 | 5.0 |
| | ✓ | ✗ | Feature | Token | RNN | 0.0750 | 6 | 0.2004 | 6 | 6.0 |
| | ✗ | ✗ | Feature | Token | RNN | 0.0758 | 7 | 0.2007 | 7 | 7.0 |
| | ✓ | ✗ | Feature | Invert | Trans. | 0.0763 | 8 | 0.2043 | 9 | 8.5 |
| | ✗ | ✗ | Temporal | None | MLP | 0.0771 | 10 | 0.2027 | 8 | 9.0 |
| | ✗ | ✗ | Temporal | Freq | MLP | 0.0766 | 9 | 0.2043 | 10 | 9.5 |

Table 28: Top-10 configurations for the PEMS04 dataset univariate forecasting. IN: Instance Norm, SD: Series Decomposition. ✓ indicates module used, ✗ indicates not used. Red/blue highlights indicate best and second-best performances.

| Setup | IN | SD | Fusion | Embed | FF-Type | MSE | Rank | MAE | Rank | Total |
|---|---|---|---|---|---|---|---|---|---|---|
| | ✗ | ✗ | Temporal | Token | RNN | **0.0795** | **1** | **0.2047** | **1** | **1.0** |
| | ✗ | ✗ | Feature | Patch | Trans. | **0.0823** | **2** | 0.2080 | 3 | 2.5 |
| | ✗ | ✗ | Feature | Token | RNN | 0.0830 | 3 | **0.2079** | **2** | 2.5 |
| | ✗ | ✗ | Temporal | Freq | MLP | 0.0834 | 4 | 0.2095 | 5 | 4.5 |
| PEMS07_12_S | ✗ | ✗ | Temporal | None | MLP | 0.0834 | 5 | 0.2096 | 6 | 5.5 |
| | ✗ | ✗ | Feature | Invert | RNN | 0.0839 | 7 | 0.2093 | 4 | 5.5 |
| | ✓ | ✗ | Feature | Patch | Trans. | 0.0835 | 6 | 0.2111 | 7 | 6.5 |
| | ✓ | ✗ | Feature | None | Trans. | 0.0848 | 8 | 0.2114 | 9 | 8.5 |
| | ✗ | ✗ | Feature | Invert | Trans. | 0.0857 | 10 | 0.2112 | 8 | 9.0 |
| | ✗ | ✗ | Feature | None | RNN | 0.0856 | 9 | 0.2122 | 10 | 9.5 |
| | ✗ | ✗ | Feature | Patch | RNN | **0.1015** | **1** | **0.2292** | **1** | **1.0** |
| | ✓ | ✗ | Feature | Patch | RNN | **0.1059** | **2** | **0.2331** | **2** | 2.0 |
| | ✗ | ✗ | Temporal | Invert | MLP | 0.1085 | 4 | 0.2333 | 3 | 3.5 |
| | ✗ | ✗ | Feature | Patch | Trans. | 0.1065 | 3 | 0.2347 | 4 | 3.5 |
| PEMS07_24_S | ✓ | ✓ | Temporal | Invert | MLP | 0.1112 | 6 | 0.2368 | 5 | 5.5 |
| | ✗ | ✗ | Temporal | Token | RNN | 0.1087 | 5 | 0.2410 | 8 | 6.5 |
| | ✗ | ✗ | Temporal | None | MLP | 0.1121 | 9 | 0.2395 | 6 | 7.5 |
| | ✗ | ✗ | Feature | Invert | RNN | 0.1113 | 7 | 0.2418 | 9 | 8.0 |
| | ✗ | ✗ | Temporal | Freq | MLP | 0.1136 | 10 | 0.2396 | 7 | 8.5 |
| | ✗ | ✗ | Temporal | Token | Trans. | 0.1117 | 8 | 0.2420 | 10 | 9.0 |
| | ✗ | ✗ | Feature | Patch | RNN | **0.1241** | **2** | **0.2491** | **1** | **1.5** |
| | ✗ | ✗ | Temporal | Token | RNN | **0.1204** | **1** | 0.2560 | 4 | 2.5 |
| | ✗ | ✗ | Feature | Patch | Trans. | 0.1284 | 3 | **0.2554** | **3** | 3.0 |
| | ✗ | ✗ | Temporal | Invert | MLP | 0.1321 | 6 | 0.2530 | 2 | 4.0 |
| PEMS07_36_S | ✗ | ✗ | Temporal | None | MLP | 0.1310 | 4 | 0.2575 | 6 | 5.0 |
| | ✓ | ✗ | Feature | Patch | RNN | 0.1319 | 5 | 0.2570 | 5 | 5.0 |
| | ✓ | ✓ | Temporal | Invert | MLP | 0.1346 | 7 | 0.2587 | 8 | 7.5 |
| | ✗ | ✗ | Feature | Token | RNN | 0.1362 | 9 | 0.2583 | 7 | 8.0 |
| | ✓ | ✗ | Feature | Token | Trans. | 0.1358 | 8 | 0.2625 | 11 | 9.5 |
| | ✗ | ✗ | Feature | None | Trans. | 0.1366 | 11 | 0.2619 | 10 | 10.5 |
| | ✗ | ✗ | Temporal | None | MLP | **0.1408** | **2** | **0.2677** | **2** | **2.0** |
| | ✓ | ✗ | Feature | Patch | RNN | 0.1429 | 3 | **0.2661** | **1** | **2.0** |
| | ✗ | ✗ | Temporal | Token | RNN | **0.1349** | **1** | 0.2744 | 6 | 3.5 |
| | ✗ | ✗ | Temporal | Invert | MLP | 0.1449 | 4 | 0.2716 | 4 | 4.0 |
| PEMS07_48_S | ✗ | ✗ | Feature | Patch | RNN | 0.1501 | 9 | 0.2682 | 3 | 6.0 |
| | ✗ | ✗ | Feature | Patch | Trans. | 0.1466 | 6 | 0.2775 | 10 | 8.0 |
| | ✗ | ✗ | Feature | Invert | RNN | 0.1462 | 5 | 0.2800 | 11 | 8.0 |
| | ✗ | ✗ | Feature | None | Trans. | 0.1502 | 10 | 0.2748 | 8 | 9.0 |
| | ✗ | ✗ | Temporal | Token | Trans. | 0.1490 | 8 | 0.2803 | 12 | 10.0 |
| | ✓ | ✗ | Feature | Invert | Trans. | 0.1529 | 15 | 0.2741 | 5 | 10.0 |

Table 29: Top-10 configurations for the PEMS07 dataset univariate forecasting. IN: Instance Norm, SD: Series Decomposition. ✓ indicates module used, ✗ indicates not used. Red/blue highlights indicate best and second-best performances.

| Setup | IN | SD | Fusion | Embed | FF-Type | MSE | Rank | MAE | Rank | Total |
|---|---|---|---|---|---|---|---|---|---|---|
| | ✗ | ✗ | Temporal | Freq | MLP | **0.1575** | **1** | **0.2768** | **1** | **1.0** |
| | ✗ | ✗ | Temporal | Token | RNN | **0.1597** | **2** | 0.2812 | 3 | 2.5 |
| | ✗ | ✗ | Feature | Patch | RNN | 0.1608 | 3 | **0.2805** | **2** | 2.5 |
| | ✗ | ✗ | Feature | None | Trans. | 0.1652 | 5 | 0.2847 | 6 | 5.5 |
| PEMS08_12_S | ✗ | ✗ | Temporal | Token | Trans. | 0.1659 | 7 | 0.2828 | 5 | 6.0 |
| | ✗ | ✗ | Feature | Token | RNN | 0.1668 | 8 | 0.2822 | 4 | 6.0 |
| | ✗ | ✗ | Feature | Patch | Trans. | 0.1653 | 6 | 0.2869 | 8 | 7.0 |
| | ✓ | ✗ | Feature | Patch | RNN | 0.1671 | 9 | 0.2863 | 7 | 8.0 |
| | ✗ | ✗ | Temporal | Patch | MLP | 0.1623 | 4 | 0.2902 | 15 | 9.5 |
| | ✗ | ✗ | Feature | Invert | Trans. | 0.1696 | 11 | 0.2883 | 10 | 10.5 |
| | ✗ | ✗ | Feature | Patch | RNN | **0.1796** | **1** | **0.2985** | **1** | **1.0** |
| | ✗ | ✗ | Temporal | Freq | MLP | **0.1802** | **2** | **0.3012** | **2** | 2.0 |
| | ✗ | ✗ | Feature | Token | RNN | 0.1859 | 4 | 0.3029 | 4 | 4.0 |
| | ✗ | ✗ | Feature | Patch | Trans. | 0.1851 | 3 | 0.3059 | 7 | 5.0 |
| PEMS08_24_S | ✗ | ✗ | Temporal | Token | MLP | 0.1866 | 6 | 0.3030 | 5 | 5.5 |
| | ✗ | ✗ | Temporal | Invert | MLP | 0.1880 | 9 | 0.3028 | 3 | 6.0 |
| | ✗ | ✗ | Temporal | Token | Trans. | 0.1880 | 8 | 0.3046 | 6 | 7.0 |
| | ✗ | ✗ | Feature | Invert | Trans. | 0.1879 | 7 | 0.3070 | 11 | 9.0 |
| | ✓ | ✗ | Feature | Patch | RNN | 0.1864 | 5 | 0.3078 | 13 | 9.0 |
| | ✗ | ✗ | Temporal | None | MLP | 0.1911 | 11 | 0.3065 | 9 | 10.0 |
| | ✗ | ✗ | Temporal | Invert | MLP | **0.1980** | **2** | **0.3148** | **1** | **1.5** |
| | ✗ | ✗ | Feature | Invert | Trans. | **0.1979** | **1** | 0.3163 | 3 | 2.0 |
| | ✗ | ✗ | Feature | Token | RNN | 0.1994 | 3 | **0.3160** | **2** | 2.5 |
| | ✗ | ✗ | Feature | Patch | RNN | 0.1994 | 4 | 0.3191 | 7 | 5.5 |
| PEMS08_36_S | ✗ | ✗ | Feature | Invert | RNN | 0.2018 | 7 | 0.3167 | 5 | 6.0 |
| | ✗ | ✗ | Temporal | Token | MLP | 0.2036 | 10 | 0.3163 | 4 | 7.0 |
| | ✗ | ✗ | Feature | Patch | Trans. | 0.2006 | 5 | 0.3198 | 9 | 7.0 |
| | ✗ | ✗ | Temporal | None | MLP | 0.2028 | 9 | 0.3182 | 6 | 7.5 |
| | ✓ | ✗ | Feature | Patch | RNN | 0.2012 | 6 | 0.3204 | 10 | 8.0 |
| | ✗ | ✗ | Temporal | Freq | MLP | 0.2023 | 8 | 0.3206 | 11 | 9.5 |
| | ✗ | ✗ | Temporal | Token | MLP | 0.2059 | 3 | **0.3204** | **1** | **2.0** |
| | ✗ | ✗ | Temporal | Invert | MLP | **0.2039** | **1** | 0.3236 | 5 | 3.0 |
| | ✗ | ✗ | Temporal | None | MLP | **0.2055** | **2** | 0.3234 | 4 | 3.0 |
| | ✗ | ✗ | Feature | Invert | RNN | 0.2073 | 5 | **0.3221** | **2** | 3.5 |
| PEMS08_48_S | ✗ | ✗ | Feature | Patch | RNN | 0.2078 | 6 | 0.3227 | 3 | 4.5 |
| | ✗ | ✗ | Feature | Invert | Trans. | 0.2067 | 4 | 0.3241 | 6 | 5.0 |
| | ✗ | ✗ | Feature | Token | RNN | 0.2099 | 8 | 0.3247 | 7 | 7.5 |
| | ✗ | ✗ | Feature | Patch | Trans. | 0.2097 | 7 | 0.3283 | 8 | 7.5 |
| | ✓ | ✗ | Temporal | Invert | MLP | 0.2129 | 10 | 0.3315 | 11 | 10.5 |
| | ✗ | ✗ | Feature | None | RNN | 0.2155 | 12 | 0.3318 | 12 | 12.0 |

Table 30: Top-10 configurations for the PEMS08 dataset univariate forecasting. IN: Instance Norm, SD: Series Decomposition. ✓ indicates module used, ✗ indicates not used. Red/blue highlights indicate best and second-best performances.

