# OpenReview forum: "TimeRecipe: A Time-Series Forecasting Recipe via Benchmarking Module Level Effectiveness"
_ICLR.cc/2026/Conference — ICLR 2026 Poster_

### Official Review · Reviewer_W7W6 · 2025-10-26

**Soundness:** 4
**Presentation:** 3
**Contribution:** 3
**Rating:** 8
**Confidence:** 4

**Summary:**

TimeRecipe is a benchmarking framework designed to systematically evaluate time-series forecasting methods at the module level, rather than just assessing complete end-to-end models. This paper is well-written and easy to follow and reproduce, providing interesting view for designing/selecting TSF methods/modules.

**Strengths:**

1. It pioneers a systematic evaluation focused on the effectiveness of individual forecasting model components (modules), addressing a gap left by existing benchmarks that primarily assess entire models.
2. the study is built upon a large-scale experimental setup, encompassing over 10,000 experiments that cover numerous module combinations.
3. this benchmark successfully identifies module combinations that outperform established SOTA models in over 90% of evaluated scenarios , achieving an average error reduction. Crucially, it provides practical insights by correlating module effectiveness with data properties and offers a toolkit for model selection.

**Weaknesses:**

1. The connections drawn between module effectiveness and data characteristics are based on extensive empirical results and statistical tests, the paper provides limited theoretical analysis or causal explanation.
2. The benchmark focuses solely on predictive accuracy and completely omits analysis of computational efficiency. This oversight limits the benchmark's practical applicability, as users cannot assess the crucial accuracy-efficiency trade-offs needed for real-world deployments, especially in resource-constrained or latency-sensitive scenarios. Incorporating complexity analysis, like in LightCTS, would significantly enhance the framework's value.

**Questions:**

Please see the weaknesses.

---

> ### Author Response · Authors · 2025-11-14
> **Rebuttal to Reviewer W7W6**
>
> We appreciate the valuable comments from the reviewer, and we are willing to address the concerns.
>
> **Response to W1: Theoretical Analysis**
>
> We agree that our study does not include a full theoretical analysis. However, developing a rigorous theoretical understanding for even a single architectural design can itself be worth an entire line of research, which goes beyond our current scope of a benchmark-style paper that aims to provide empirical insights and systematic observations.
>
> Nevertheless, our empirical findings do reveal several non-trivial and theoretically intriguing phenomena that could inspire future analytical work. For instance, we observe that series decomposition performs better under lower distributional shift in multivariate setups, but under higher shift in univariate setups, which is a seemingly contradictory behavior that opens interesting directions for future theoretical exploration.
>
> **Response to W2: Computation Cost**
>
> We appreciate the reviewer’s valuable comment and fully agree that efficiency is an important aspect to analyze. In our current study, we primarily focused on predictive accuracy, as for most time-series forecasting module combinations, the differences in computational cost are relatively minor. However, we acknowledge that certain configurations, such as invert embedding + temporal fusion + transformer, can significantly increase computational complexity.
>
> For instance, this combination produces an attention tensor of shape [B,H,H,H], where H is the hidden dimension: Through invert embedding, the input becomes [B,H,D]; and when attention is performed over the second dimension, it expands to [B,H,H,H], leading to substantial memory usage and training overhead due to the tensor size and corresponding operations. We will include such computational analysis and discussions in the final version to better reflect the accuracy–efficiency trade-off.

---

### Official Review · Reviewer_qk8s · 2025-10-28

**Soundness:** 3
**Presentation:** 3
**Contribution:** 3
**Rating:** 6
**Confidence:** 4

**Summary:**

This paper introduces TimeRecipe, a modular benchmarking framework for time-series forecasting that evaluates the effectiveness of common architectural components (e.g., normalization, decomposition, embedding, fusion, feed-forward modeling) across a wide range of forecasting tasks, horizons, and datasets. By systematically exploring over 10,000 combinations, TimeRecipe identifies component-level contributions to performance, offers statistical correlations between data properties and module choices, and supplies a training-free recommendation toolkit for model selection. Extensive results demonstrate that exhaustive module-level exploration can yield architectures that outperform existing state-of-the-art (SOTA) methods and provide practical guidance for real-world forecasting.

**Strengths:**

1、The paper introduces a new paradigm for time-series forecasting benchmarking by breaking down models into five core modules and systematically benchmarking their combinations.

2、There are quite a few nice illustrations.

3、 This work focuses on an important problem that could have real-world applications.

4、 The figures and tables used in this work are clear and easy to read.

**Weaknesses:**

1、While the coverage of LTSF, PEMS, and M4 datasets is excellent, novel datasets introduced (e.g., unemployment forecasting from Time-MMD) are only briefly mentioned and lack rigorous description (see Section 4.2 and Appendix B). For maximal transparency, the properties, preprocessing, and evaluation setup should be as detailed for these new datasets as for the standard ones.

2、While Table 2 and Figure 1 are helpful, many of the empirical summaries require close reading to decipher key findings. It would be helpful if the text more directly guided the reader through takeaways from these visualizations; for example, highlighting trends or “rules” in Table 2, or explicit failure cases identified in the configuration tables.

3、A large combinatorial space is searched, but more discussion of robustness to random seed, hyperparameters, or the presence of multiple optima would be valuable. Are the best configurations stable across resamplings—or is performance sensitive to chance initialization? Furthermore, are error magnitudes meaningful under operational constraints for real-world forecasting tasks?

**Questions:**

1、How robust are the results to changes in hyperparameters, default training budgets, or random seeds? Are the observed improvements for best configurations consistent under different parameterizations?

2、For the model selection toolkit: How does the training-free strategy compare to stronger meta-learning or AutoML approaches? Is LightGBM sufficient, or do more advanced selectors close the gap further to the global best configurations?

3、Can the authors provide additional clarity or pseudocode for the inter-module dimension handling, especially for embeddings (patch vs. invert) and their compatibility with MLP/Transformer/RNN blocks?

4、Is there a sense in which the best module combinations generalize beyond the datasets tested, or are they highly scenario-dependent? Are any “failure” cases (where TimeRecipe-selected models underperform SOTA) documented, and why do they arise?

---

> ### Author Response · Authors · 2025-11-14
> **Rebuttal to Reviewer qk8s**
>
> We appreciate the valuable comments from the reviewer, and we are willing to address the concerns.
>
> **Response to W1: Dataset Explanation**
>
> We appreciate the reviewer’s comment. The Time-MMD dataset is a multimodal dataset containing univariate time-series data, used primarily to evaluate the generalization ability of our module-selection framework. We agree that a more detailed description would improve clarity. In the final version, we will include additional details on its properties, preprocessing procedures, and evaluation setup to ensure full transparency and reproducibility.
>
> **Response to W2: Visualization**
>
> We appreciate the reviewer’s valuable suggestion regarding visualization. In the initial submission, we primarily used table-based summaries (e.g., Table 2) due to strict space constraints. We agree that clearer visual guidance would make the key insights more accessible. In the final version, we plan to incorporate additional visualizations, such as trend plots on the win-rate of different embeddings across datasets and data-property conditions, to more directly highlight the main takeaways in our analysis.
>
> **Response to W3 & Q1: Training Setup**
>
> We appreciate the reviewer’s thoughtful comments on robustness and stability. To ensure robustness, all results for each configuration are averaged over four random seeds. For hyperparameter tuning, we tune key architectural parameters such as hidden dimension, number of layers, and method-specific settings (e.g., patch length). This helps each module combination remain representative without being under-representative or too overparameterized, which are expensive to train, enabling fair architectural comparisons.
>
> For training-related hyperparameters (e.g., learning rate), we adopt the standard tuning strategy from the Time-Series Library, following common practices in recent forecasting literature. Default initialization is used, which is generally robust across architectures in time-series forecasting tasks.
>
> Finally, to ensure that observed performance differences are meaningful, we conduct paired t-tests across runs, and only report conclusions with p-values < 0.05, confirming statistical significance. We will include a more detailed discussion of this evaluation protocol in the final version.
>
> **Response to Q2: AutoML**
>
> We thank the reviewer for this insightful question. Existing AutoML approaches for time-series forecasting mainly focus on hyperparameter optimization, rather than module selection. In contrast, TimeRecipe is designed as a complementary framework that enables architecture-level selection in a training-free manner. Our goal is therefore not to outperform current AutoML systems, but to inspire their extension toward incorporating model architecture selection. As shown in Section 4.3, integrating our module-selection mechanism into existing AutoML pipelines represents a promising future direction, which we leave for future work.
>
> **Response to Q3: Pseudo Code**
>
> We appreciate the reviewer’s request for additional implementation clarity. The full implementation is available at:
> https://anonymous.4open.science/r/timerecipe_iclr-608B/model/unitsf.py
>
> Specifically, we create a module dictionary (lines 51–96) that defines all compatible combinations capable of modifying intermediate hidden dimensions. The model then constructs the embedding and backbone accordingly (lines 173–184 and 187–209). The forward process proceeds through a consistent pipeline: instance normalization, series decomposition, embedding, permutation for temporal/feature fusion,  feed-forward model (MLP/Transformer/RNN) , and projection (lines 125–170). This structure ensures dimension consistency across different module types (e.g., patch vs. inverted embeddings) and supports flexible integration with MLP-, Transformer-, or RNN-based forecasting backbones by querying the dictionary.

---

> > ### Author Response · Authors · 2025-11-14
> > **Rebuttal to Reviewer qk8s Contd.**
> >
> > **Response to Q4: Failure Modes**
> >
> > We thank the reviewer for raising this question (we also address the second concern in Q2 here). The key finding of our work is that forecasting performance is indeed highly scenario-dependent. However, by quantizing diverse forecasting scenarios into several measurable data properties, we are able to uncover generalizable correlations between module selection and these quantized properties.
> >
> > Regarding generalization, our current implementation employs LightGBM as a simple proof-of-concept selector. While it effectively identifies reasonable architectures, it is not perfect due to its tendency to produce smoother decision boundaries, due to its algorithmic nature. For example, in Table 5 (Environment dataset), LightGBM recommends only RNN modules, although MLP can also perform competitively, or in Table 3, its recommendation includes an architecture with a low rank.
> >
> > As discussed in Section 4.3, a promising future direction is to treat these data properties as meta-features and jointly learn their relationships with forecasting performance using more expressive meta-learning strategies. This direction goes beyond the current scope but represents an exciting next step.

---

> ### Comment · Reviewer_qk8s · 2025-11-17
> **My question has been well resolved, so I have decided to increase my score！**
>
> Thank you for your reply. My question has been well resolved, so I have decided to increase my score and support the acceptance of my paper.

---

### Official Review · Reviewer_NBVJ · 2025-10-28

**Soundness:** 2
**Presentation:** 3
**Contribution:** 1
**Rating:** 2
**Confidence:** 4

**Summary:**

The paper advances a module-centric framework for time-series forecasting. Specifically, the authors decompose canonical architectures into pre-processing, embedding, backbone, fusion strategies, and projection/post-processing. Building on this decomposition, the authors conduct a large-scale benchmarking study that enumerates and evaluates numerous module combinations across diverse tasks and datasets. The study yields statistically grounded mappings from data characteristics, such as trend strength, seasonality, and cross-channel correlation, to appropriate module choices.

**Strengths:**

1. The paper is well written and easy to understand.
2. The mapping from measured properties to module choices is an interesting idea and worthy of investigation.
3. The training-free selector is a pragmatic contribution that can reduce exploration cost.

**Weaknesses:**

1. Regularization, optimization, schedulers, and data augmentation are not systematically modularized, though they often rival architecture in impact. The historical-window length is also unclear, despite its significant effect on performance.
2. The choice of datasets and prediction horizons (e.g., 720) has been criticized by researchers as impractical in real-world settings (https://cbergmeir.com/talks/bergmeir2024NeurIPSInvTalk.pdf), which weakens the reliability of the conclusions and suggests the findings may be primarily academic.
3. Some intertwined cross-channel mixers and hybrid designs are only partially represented within the studied design space.

**Questions:**

1. Do the authors plan to modularize regularization, optimizers, schedulers, and augmentation so that training choices can be benchmarked with the same rigor as architecture?
2. How would this framework integrate with pretrained time-series backbones, for example, by swapping embeddings/fusion while freezing a large backbone, or by using the selector to choose adaptation routes?

---

> ### Author Response · Authors · 2025-11-14
> **Rebuttal to Reviewer NBVJ**
>
> We appreciate the valuable comments from the reviewer, and we are willing to address the concerns.
>
> **Response to W1 & Q1: Additional Components**
>
> We appreciate the reviewer’s valuable suggestions regarding benchmarking additional components such as regularization, optimizers, schedulers, and data augmentation. We agree that these factors can be influential. However, we intentionally did not include them in this study for the following three reasons:
>
> First, although these components can affect performance, architectural choices typically have a more fundamental and substantial impact. This is supported by prior work, e.g., TimeMixer [1] reports consistent improvements over baselines both with and without hyperparameter fine-tuning. Such observations align with our motivation that architectural design is the primary determinant of forecasting performance, whereas optimization strategies mainly refine the results.
>
> Second, the time-series community already has many AutoML studies that contribute to hyperparameter tuning and optimization strategies (e.g., the survey [2]). In contrast, no existing work systematically studies module-level architectural effectiveness, nor how to select appropriate architectural components for forecasting. Our work fills this missing gap and complements existing AutoML efforts by focusing on architectural selection rather than re-addressing well-studied optimization aspects.
>
> Third, while model architecture in time-series forecasting remains highly debated, e.g., discussions about the usefulness of Transformers, the debates around optimization choices are comparatively mild. In practice, widely adopted defaults such as the Adam optimizer work robustly across diverse forecasting setups and rarely drive the core scientific disagreement in the field.
>
> **Response to W2: Reliability**
>
> We appreciate the reviewer’s concern regarding the reliability of conclusions drawn from datasets such as ETT. We acknowledge that these datasets, which are representative of the broader class of long-term forecasting benchmarks, have indeed received criticism for their relatively unrealistic forecasting horizons. However, including them in our benchmark does not undermine the validity of our findings, for three reasons:
>
> First, ETT datasets are part of our evaluations, where our benchmark also includes short-term forecasting settings such as M4 and PEMS, thereby ensuring a more balanced evaluation across forecasting horizons. Even with the well-studied ETT datasets, our study still reveals impressive and interesting intuitions, some of which are counterintuitive, as exemplified in lines 402-404.
>
> Second (also address the last concern in W1), although we use fixed lookback windows to predict target horizons, we do not assess performance solely based on the absolute horizon length. Instead, we adopt a data property named horizon-to-lookback (hl) ratio, which more fairly captures the difficulty level of forecasting tasks. For example, forecasting 720 future points with 96 historical points is indeed unrealistic; however, forecasting 720 points with 7200 historical points is analogous to forecasting 72 points with 720 lookback steps, which is a more realistic setup compared to the former one. This ratio-based view helps generalize the analysis beyond specific dataset settings.
>
> Third, we recognize that many practically relevant time-series datasets are private and thus inaccessible to the public (for example, rare finance companies would share their up-to-date datasets to ensure only they can make money rather than the public), public datasets such as ETT remain valuable for reproducibility and comparability. Although imperfect, they serve as widely accepted benchmarks that allow researchers to validate whether observed results are consistent with other studies. More importantly, our results in Section 4.2 demonstrate that the knowledge learned from these public datasets, through our module selection process via LightGBM, can generalize effectively to unseen datasets like Time-MMD, a short-term forecasting dataset. This suggests that even though our analysis leverages public datasets such as ETT, the insights gained remain applicable to practical, unseen (and potentially private) real-world datasets.

---

> > ### Author Response · Authors · 2025-11-14
> > **Rebuttal to Reviewer NBVJ Contd.**
> >
> > **Response to W3: Additional Architectural Design**
> >
> > We appreciate the reviewer’s insightful comment. Indeed, it is infeasible to include every possible architectural variant within a single benchmark. As noted in Remark 2, some hybrid or cross-channel designs were intentionally omitted because they are not yet as widely adopted or standardized as other module types. The examples mentioned by the reviewer, such as temporal plus feature fusion architectures (e.g., Crossformer and TSMixer), represent notable advances, but their adoption remains relatively limited compared to more established module families like Transformers. Therefore, we prioritize covering widely recognized architectural modules that reflect common design practices in current time-series forecasting research.
> >
> > **Response to Q2: Pre-Trained Backbone**
> >
> > TimeRecipe can be integrated with pre-trained backbones in principle. For example, one could replace the Transformer FF-model module with a stack of pre-trained Transformer blocks. However, doing so typically requires additional fine-tuning steps to adapt the pretrained backbone to the forecasting task, which goes beyond the conventional supervised setting we study in this work. Exploring such integration is an interesting direction for future research, but is currently outside the scope of this paper.
> >
> > **Rebuttal to Contribution Assessment**
> >
> > We would like to respectfully argue that this work makes a meaningful and necessary contribution to the time-series forecasting community.
> >
> > As discussed in our responses to the reviewer’s specific concerns (e.g., the unrealistic setup), the current research landscape in time-series forecasting is characterized by numerous debates, inconsistencies, and fragmented empirical findings. Many studies tend to pursue marginal “SOTA” gains on a limited set of benchmark datasets (e.g., ETT), often relying on tuning tricks or hyperparameter adjustments rather than a true understanding of why a particular architectural design works. For instance, applying instance normalization alone can dominate performance on the ETT datasets, making it difficult to disentangle whether the reported improvements truly stem from the proposed architectural innovations they aim to sell or from auxiliary tricks and parameter tuning. As an example, iTransformer outperforms PatchTST on ETT, yet patch embedding remains the more widely accepted design for later foundation models. Consequently, conclusions drawn under such conditions may overstate the general effectiveness of certain designs.
> >
> > In contrast, our work aims to provide principled intuition and an in-depth understanding of module-level effectiveness across diverse datasets and forecasting scenarios, which has never been studied by existing research or benchmark studies. For example, instance normalization tends to be beneficial in long-term forecasting but less so for short-term forecasting or when series are highly stationary. We believe this benchmark provides valuable, actionable insights for both researchers and practitioners in selecting or developing architectures that best fit their own forecasting tasks and data conditions.
> >
> > [1] Timemixer: Decomposable multiscale mixing for time series forecasting
> >
> > [2] Review of ML and AutoML solutions to forecast time-series data

---

> > > ### Comment · Reviewer_NBVJ · 2025-11-26
> > >
> > > Thank you very much for the rebuttal. The authors mention that TimeMixer reports consistent improvements over baselines both with and without hyperparameter fine-tuning. However, using a single example (TimeMixer) to argue that factors such as regularization, optimizers, schedulers, and data augmentation are less important than architecture is not convincing. As a researcher in this field, I often find these factors at least as important as the architectural design itself. The authors also seem to partly agree with this point in their later statements: “Many studies tend to pursue marginal ‘SOTA’ gains on a limited set of benchmark datasets (e.g., ETT), often relying on tuning tricks or hyperparameter adjustments rather than a true understanding of why a particular architectural design works.”
> > >
> > > I acknowledge that the conclusions drawn from this study may provide useful insights, but these insights are restricted to the datasets evaluated in this work, and their generalization to a broader range of time series data (e.g., the recent built benchmark fev-bench and gift-eval) is questionable. Using Time-MMD alone to demonstrate generalization is not sufficient.

---

> > > > ### Author Response · Authors · 2025-11-29
> > > > **Response to Reviewer NBVJ**
> > > >
> > > > Thank you for the additional comments. We would like to clarify our position on the key points raised.
> > > >
> > > > **First**, as stated in the title and throughout the paper, the focus of this work is **module-level architectural analysis within the conventional supervised forecasting setup**. For hyperparameters related to these modules, we do perform tuning over key architectural parameters such as the hidden dimension, number of layers, and method-specific settings (e.g., patch length). This ensures that each module combination is neither too small to be under-representative nor too large to cause excessive training cost, while maintaining fair architectural comparisons across module combinations.
> > > >
> > > > For training-related hyperparameters (e.g., learning rate), while we acknowledge that optimization strategies (regularization, schedulers, etc.) are important in practice, they fall outside the scope of architectural modularization. Our intention is not to argue that these factors are unimportant, but that including them would confound the measurement of architectural inductive biases, which are the primary subject of this study. This follows the common practice of architecture-centric benchmarks in NAS, where optimizations are deliberately fixed to isolate architectural effects [1,2].
> > > >
> > > > **Second**, regarding fev-bench and gift-eval:
> > > > **fev-bench was first released on September 30, 2025, after the ICLR submission deadline (September 24)**. More importantly, **both fev-bench and gift-eval explicitly target pretrained foundation models, whereas our study focuses on supervised forecasting**. Their evaluation protocols differ significantly from the supervised setting, and incorporating them would not constitute a fair comparison. However, we agree that these benchmarks are valuable and are willing to bring them into discussion in the final version of this paper.
> > > >
> > > > **Third**, regarding the reviewer’s statement that “these factors are at least as important as the architectural design itself,” we fully agree that hyperparameter tuning and optimization strategies matter in any machine learning pipeline. However, these aspects are widely recognized as **basic practices rather than conceptual contributions**.
> > > > In contrast, our work aims to provide intuition on what fundamentally contributes to forecasting performance, namely, identifying which architectural designs are actually useful, rather than re-emphasizing well-known optimization routines. As evidence, time-series literature frequently debates whether MLPs outperform Transformers, but to our best knowledge, there is no single work that particularly studies just arguing that “SGD is better than Adam” in time-series forecasting, nor rarely do benchmark studies aim to benchmark “the effectiveness of Adam over SGD” even in broader machine learning topics. These choices are important in practice but are not considered research insights.
> > > >
> > > > We also acknowledge the practical value of hyperparameter optimization and indeed recommend HPO tools such as Optuna. However, such tools still cannot replace a principled understanding of what structural choices matter for forecasting, which is precisely the novel scope our work investigates. Maybe one interesting future direction would be to integrate TimeRecipe with established AutoML pipelines so that hyperparameter and architectural choices can be jointly optimized; however, such an integrated system goes beyond the scope of our current work.
> > > >
> > > > [1] Nas-bench-101: Towards reproducible neural architecture search
> > > >
> > > > [2] NAS-Bench-Graph: Benchmarking Graph Neural Architecture Search

---

### Official Review · Reviewer_Vm6s · 2025-11-01

**Soundness:** 2
**Presentation:** 2
**Contribution:** 2
**Rating:** 4
**Confidence:** 5

**Summary:**

This paper focuses on time series forecasting applications. Building TSFM requires many modules, and testing each module is an important task. The introduction section is written clearly - the 4 outcomes of the study.

**Strengths:**

The paper has conducted an extensive survey of reusable components and then performed a systematic study. The Table 2 is an output of such extensive study.

**Weaknesses:**

- Given a paper submitted on learning time series and dynamical systems, I feel the paper is more suitable for the benchmark and dataset track. Thus, I have started looking at the paper from a benchmarking and experimental task perspective. Why are the foundation models not part of this work?

- Motivation. If I am a developer, how can I consume the outcome of your study? For example, can you provide a case study on how tord- get a leaderboatopping agent on GiftEval? GiftEval is a time series leaderboard, so it shows how much gain I have made using Table 2.

- There is an alternate thought: I would go to Leaderboard, compile all the models listed there along with their performance, and then generate Table 2. How accurate can it be? In summary, can I generate the same table 2 but without going into the entire time recipe framework? Can you compare and contrast this alternate, more dynamic approach?

- Hyperparameter tuning, I did not see how you capture the effect of parameter tuning?

- What are the new datasets that studies bring and then demonstrate the value of learning?

**Questions:**

Please address all the weak points.

---

> ### Author Response · Authors · 2025-11-14
> **Rebuttal to Reviewer Vm6s**
>
> We appreciate the valuable comments from the reviewer, and we are willing to address the concerns.
>
> **Response to W1:**
>
> **Primary Area Choice:** Our work covers both topics on time series and benchmark. We selected “learning on time series and dynamical systems” as the primary area because the contributions and insights of this paper are most relevant to the time-series community. We believe reviewers with expertise in time series could give fairer and deeper assessments of our work.
>
> **Foundation Models:** We appreciate the reviewer’s suggestion, and we had included the explanation in Remark 3 (page 5) for clarification. (1) Our work focuses on supervised forecasting models, aiming to provide actionable insights on the effectiveness at the module-level design. (2) Current time-series foundation models remain at an early and fragmented stage, lacking a common module design beyond Transformer variants. (3) Foundation models are typically pre-trained and end-to-end; altering a single module can easily disrupt dependencies among other pre-trained components, making systematic module analysis infeasible. (4) GiftEval has already provided comprehensive benchmarks for foundation models, and we intentionally avoid duplicating that scope.
>
> **Response to W2: Usage**
>
> We appreciate the reviewer’s question regarding the practical usage of our study. We want to clarify that the main purpose and outcome of our work is to **understand how different module designs perform under various forecasting scenarios, as summarized in Table 2, rather than create a new SOTA that beats a leaderboard**, for example, GiftEval. We provide in-depth intuitions of module-level effectiveness in Table 2, and to make these findings actionable, we propose a lightweight LightGBM-based recommender that predicts suitable model architectures for a new dataset (e.g., a private or unseen time-series dataset), leveraging the benchmarking results from TimeRecipe. This approach demonstrates simple yet effective performance, as shown in Tables 3 and 5.
>
> Beyond our simple LightGBM, practitioners can also directly benefit from the empirical insights. For example, given a new time-series forecasting task, one can examine its data properties and task setup without training, refer to the corresponding patterns in Table 2, and choose modules that are more likely to succeed: for instance, adopting instance normalization for long-term forecasting scenarios.
>
> **Response to W3: Go GiftEval Leaderboard**
>
> We acknowledge that GiftEval is a well-established and influential leaderboard in the time-series community. However, GiftEval is designed specifically for evaluating foundation models, primarily in zero-shot settings. As a result, strong supervised forecasting models, such as PatchTST, rank low on GiftEval simply because the evaluation protocol is not aligned with their training paradigm. Our work studies the module-level effectiveness of supervised forecasting architectures, which lies outside the scope of GiftEval’s objectives. Using zero-shot–oriented leaderboards to assess supervised module designs would therefore be inappropriate and potentially misleading.
>
> In contrast, Table 2 indeed summarizes the relative effectiveness of different module designs, which itself shares a similar idea to a leaderboard. For instance, we identify that using module X outperforms not using it (or using module Z) when the data property Y holds. The key difference is that Table 2 does not provide an explicit global ranking, as both the data properties and module designs are highly diverse.
>
> Regarding evaluating a new design, a comprehensive assessment indeed requires running the full TimeRecipe framework, since a single module cannot yield predictions independently and must be retrained in combination with others under a supervised setup. Nonetheless, a more efficient partial evaluation is possible: developers may start from the top-k best-performing module combinations identified by TimeRecipe, test their new design within those configurations, and thus substantially reduce computational cost while still obtaining informative insights.
>
> **Response to W4: Hyperparameter Tuning**
>
> We appreciate the reviewer’s comment. Many module designs (e.g., series decomposition) are non-hyperparametric by nature. For components involving hyperparameters, we do perform tuning over key architectural parameters such as the hidden dimension, number of layers, and method-specific settings (e.g., patch length). This ensures that each module combination is neither too small to be under-representative nor too large to cause excessive training cost, while maintaining fair architectural comparisons across module combinations. For training-related hyperparameters (e.g., learning rate), we follow the standard tuning strategy provided by the Time-Series Library. We will include a detailed discussion of this procedure in the final version.

---

> > ### Author Response · Authors · 2025-11-14
> > **Rebuttal to Reviewer Vm6s Contd.**
> >
> > **Response to W5: New Dataset**
> >
> > We acknowledge that our work does not introduce new datasets. However, benchmark studies do not necessarily require new datasets to be impactful; for example, the TFB or Tsfm-bench benchmark [1,2] is built on existing data. Our goal is to provide systematic insights into the effectiveness of commonly used module designs across a broad range of well-established datasets. We believe that such a comprehensive and principled understanding is itself a valuable contribution to the time-series forecasting community.
> >
> > [1] TFB: Towards Comprehensive and Fair Benchmarking of Time Series Forecasting Methods
> >
> > [2] Tsfm-bench: A comprehensive and unified benchmark of foundation models for time series forecasting

---

> ### Comment · Reviewer_Vm6s · 2025-11-21
> **Ack on Reading of Author Response**
>
> I ack that I looked at the response.

---

> > ### Author Response · Authors · 2025-11-21
> > **Response to Reviewer Vm6s**
> >
> > We appreciate Reviewer Vm6s’s acknowledgement. We would be happy to further discuss any remaining concerns or address any additional questions.

---

> > > ### Comment · Reviewer_Vm6s · 2025-11-26
> > > **Thank you**
> > >
> > > As per ICLR Request, I already acknowledged that I read the rebuttal. The given response does not meet several of my asks, and as the paper mainly focuses on the discovery aspect but does not show how it can be applied in a real-life design or adopting the output of the TimeRecipe, the performance of my model gets improved, or we take a model listed in some paper, and due to our study, the performance increases?

---

> ### Author Response · Authors · 2025-11-29
> **Response to Reviewer Vm6s**
>
> Thank you to the Reviewer for reiterating the concern. We provide a clearer explanation regarding the practical usage of our framework.
>
> **First**, we would like to clarify **why creating a leaderboard, such as GiftEval for foundation models, is not directly applicable or equally informative in supervised forecasting**. In the foundation-model setting, a leaderboard is meaningful because these models are explicitly designed for broad zero-shot generalization. Strong performance on a diverse collection of unseen evaluation sets reliably indicates that the same model can be applied to a practitioner’s private dataset without fine-tuning. In such a paradigm, a leaderboard directly reflects real-world utility: one can simply choose a top-ranked model and run zero-shot forecasting on their own data.
>
> Supervised time-series forecasting differs fundamentally. These models do not transfer in a zero-shot manner, and their relative performance is highly sensitive to dataset-specific characteristics. A model that performs best on one public dataset (e.g., PatchTST on Electricity as reported in TFB [1]) may be far from optimal on another dataset with different temporal patterns, domain characteristics, or horizon configurations. Consequently, **even a comprehensive leaderboard would not offer actionable guidance for a completely new dataset that has never appeared in existing benchmarks.**
>
> **Second**, given this inherent limitation of supervised forecasting, our work intentionally focuses on addressing the missing gap: instead of identifying a universal “best supervised model,” **we uncover data-property-driven principles that help practitioners select architectural modules suited for their dataset, even when the dataset is unseen and unbenchmarked**. Within the current scope of our study, we highlight two practical ways in which TimeRecipe can be used (while further and more advanced usage may be worth future works):
>
> **(1) Model architecture recommendation via the LightGBM selector.**
> Tables 3 and 5 demonstrate that even a simple LightGBM-based selector trained on benchmarked datasets can recommend model configurations that outperform the SOTA supervised architectures on unseen datasets, showing that our discovered mappings can be used directly for architectural design.
>
> **(2) Direct lookup guidance from Table 2.**
> Practitioners may compute basic statistical properties of their dataset and refer to Table 2 to quickly understand which architectural modules are likely to be effective. For example, datasets with low shifting tend to benefit from combinations of series decomposition, invertible embeddings, and RNN feed-forward modules. This bridges the previous gap by linking module effectiveness with data characteristics, offering a principled way to design or adapt models for new forecasting tasks.
>
> We hope this clarifies the practical value and intended usage of our framework, as well as why the supervised setting calls for a different type of generalization guidance compared to foundation-model leaderboards.
>
> [1] TFB: Towards Comprehensive and Fair Benchmarking of Time Series Forecasting Methods

---

### Meta-Review · Area_Chair_Wcuy · 2026-01-05

**Summary:**

The primary concerns raised by the reviewers centered on the scope and practical utility of the proposed "TimeRecipe" framework. Several reviewers (Vm6s, NBVJ) questioned why the study focused exclusively on architectural modules while omitting training-related factors like regularization, optimizers, and data augmentation, which are often as impactful as architecture. There were also some discussions regarding the choice of datasets.

However, the paper's pioneering systematic evaluation of over 10,000 module combinations. It provides a "map" of architectural inductive biases in supervised time-series forecasting. I recommend an accept.

**Reviewer Concerns:**

The authors clarified their use of random seeds and statistical significance testing (paired t-tests), which satisfied Reviewer qk8s. The provision of pseudocode and repository links addressed concerns regarding inter-module dimension handling.

Reviewer NBVJ remained unconvinced that a modular study is complete without including optimizers and schedulers. While the authors argued that architectural bias is the primary subject of study in NAS-like benchmarks. The distinction between supervised forecasting and zero-shot foundation models was well-argued by the authors, but some reviewers still felt the lack of "Foundation Model" integration limits the paper's potential.

**Reviewer Scores:**

Reviewer qk8s: 6 to 8

Others remain their scores.

---

### Decision · Program_Chairs · 2026-01-26

Accept (Poster)